# Gradient Heterogeneity Complements Hessian Heterogeneity in Transformer Optimization

## Abstract

Transformers are difficult to optimize with stochastic gradient descent (SGD) and largely rely on adaptive optimizers such as Adam. Despite extensive efforts, the mechanisms behind Adam's advantage over SGD in Transformer optimization are still not fully understood. In this study, we analyze the optimization of Transformer models in the fine-tuning setting through the lens of *gradient heterogeneity*, defined as the variation in gradient norms across parameter blocks. We provide a theoretical analysis showing that gradient heterogeneity, together with Hessian heterogeneity, degrades the convergence of gradient-based methods such as SGD, while sign-based methods are substantially less sensitive to this effect. Adam and SignSGD both perform coordinate-wise updates and are less sensitive to the scale of individual gradient coordinates than SGD. This motivates our use of SignSGD as an analytically tractable proxy for Adam-like behavior. Our analysis uses the fact that SGD and SignSGD follow steepest descent directions under different norms, and derives upper bounds on the iteration complexity with implications for learning rate scaling of SignSGD. We further investigate the origin of gradient heterogeneity in Transformer architectures and show that it is strongly influenced by the placement of layer normalization, with Post-LN architectures exhibiting particularly pronounced heterogeneity. Experimental results from fine-tuning Transformers in both NLP and vision domains validate our theoretical analysis.

## 1 Introduction

Transformers (Vaswani, 2017) have achieved significant success across a wide range of tasks, particularly in language models. In practice, training language models largely relies on adaptive optimization methods (Liu et al., 2024; Grattafiori et al., 2024) such as Adam (Kingma & Ba, 2015). In contrast, while stochastic gradient descent (SGD) has long been a standard optimizer in deep learning (Lecun et al., 1998; He et al., 2016), it often exhibits inferior optimization behavior in Transformer architectures (Schmidt et al., 2021; Choi, 2019; Zhang et al., 2020b; Kunstner et al., 2023; Zhang et al., 2024a; Kunstner et al., 2024).

However, the underlying reasons for the performance gap are not yet fully understood. In particular, Adam has been shown to outperform SGD even in full-batch settings, while SignSGD (Bernstein et al., 2018), which serves as a proxy for Adam-like behavior (Xie & Li, 2024; Li et al., 2025), achieves comparable performance under the same conditions (Kunstner et al., 2023). These observations suggest that the difference between Adam and SGD cannot be explained solely by stochastic gradient noise, but rather reflects fundamental differences between SGD and adaptive optimization methods. Other explanations, such as Adam's robustness to heavy-tailed label distributions (Kunstner et al., 2024), capture certain aspects of this gap but do not fully account for the behavior observed in fine-tuning regimes with a small amount of labeled data. More recently, Zhang et al. (2024a) associated the Adam–SGD gap with *Hessian heterogeneity* in Transformers, defined as differences in block-wise Hessian spectra, although the underlying mechanism remains unclear.

In this study, we take a step toward a better understanding of the difference between Adam and SGD through a theoretical analysis. Specifically, we compare their optimization behaviors by analyzing their *iteration complexity*, defined as the number of optimization steps required for the gradient norm to become sufficiently small. Our analysis reveals that *gradient heterogeneity* and Hessian heterogeneity (Zhang et al., 2024a) jointly

Table 1: Comparison with prior studies on Transformer optimization. $\checkmark$: Supported; –: Not supported.

| Paper | Transformer | Theoretical complexity | Heterogeneity | Layer normalization |
|---|---|---|---|---|
| Zhang et al. (2020b) | $\checkmark$ | $\checkmark$ | – | – |
| Crawshaw et al. (2022) | $\checkmark$ | $\checkmark$ | – | – |
| Kunstner et al. (2023) | $\checkmark$ | – | – | – |
| Pan & Li (2022) | $\checkmark$ | – | – | – |
| Kunstner et al. (2024) | $\checkmark$ | – | – | – |
| Zhang et al. (2024a) | $\checkmark$ | – | $\checkmark$ (Hessian) | – |
| **Ours** | $\checkmark$ | $\checkmark$ | $\checkmark$ (Gradient & Hessian) | $\checkmark$ |

play an important role in shaping these differences. Gradient heterogeneity is defined as the variation in gradient norms across parameter blocks and is amenable to empirical analysis.

We begin by deriving upper bounds on the iteration complexity of gradient-based and sign-based optimization methods in both deterministic and stochastic settings. Since Adam shares coordinate-wise update characteristics with SignSGD, we use SignSGD as an analytically tractable proxy for Adam-like behavior. Our analysis uses the fact that SGD and SignSGD correspond to steepest descent directions under different norms. Our results suggest that gradient-based methods are more sensitive to gradient and Hessian heterogeneity than sign-based methods, and also provide implications for the learning rate of SignSGD. To further investigate the origin of heterogeneity, we analyze gradient heterogeneity in Transformers and examine how it relates to architectural design choices. In particular, we find that applying layer normalization after residual connections amplifies gradient heterogeneity.

Our contributions are summarized as follows. Table 1 compares prior studies with ours.

- We derive upper bounds on iteration complexity in deterministic and stochastic settings. The bounds show that SGD is controlled by the gradient-weighted quantity $\Lambda_G$, whereas sign-based (Adam-like) methods depend on $\Lambda_P$, explaining their different sensitivity to heterogeneity (Theorems 4.6 and 4.9).

- We characterize gradient heterogeneity in Transformers and relate it to architectural choices, suggesting that normalization placement (Post-LN vs. Pre-LN) can amplify heterogeneity through the Jacobian structure (Section 4.7).

- Overall, we argue that the sign-based nature of Adam-like methods mitigates optimization difficulties induced by heterogeneity across parameter blocks, a property particularly relevant to Transformer architectures.

## 2 Related work

**Adam in deep learning.** Adam (Kingma & Ba, 2015) is a widely used optimization algorithm in deep learning with convergence properties (Zhang et al., 2022). However, the reasons for its superior performance are not yet fully understood. Jiang et al. (2024) empirically observed that Adam tends to converge to parameter regions with uniform diagonal elements in the Hessian, supported by theoretical analysis based on two-layer linear models. Rosenfeld & Risteski (2024) argued that the ability of Adam to handle outliers in features is a critical factor in its effectiveness. Additionally, Kunstner et al. (2024) attributed the performance of Adam in language models to its ability to manage heavy-tailed class imbalance. Orvieto & Gower (2025) showed that setting $\beta_1 = \beta_2$ preserves Adam's performance and enables a mean-field variational interpretation. In this study, we focus on a sign-based proxy for Adam-like behavior and analyze how heterogeneity across parameter blocks affects gradient-based and sign-based optimization differently.

**Sign-based optimization and variants.** SignSGD, also known as sign descent, is an optimization method that is computationally efficient and memory-efficient, making it suitable for distributed training (Bernstein et al., 2018). Adam can be interpreted as a variance-adapted variant of SignSGD (Balles & Hennig, 2018). For example, Xie & Li (2024) analyzed the convergence properties of Adam from this perspective. Consistent with this interpretation, Zhao et al. (2025) found that sign-based optimizers restore the stability and performance

of Adam and proposed using adaptive learning rates for each layer. Several variants of sign-based optimization have been proposed, such as block-wise adaptive learning rates (Zhang et al., 2024b) and error-feedback schemes that mitigate bias and improve convergence (Karimireddy et al., 2019). Through program search, a sign-based optimization algorithm called Lion (evolved sign momentum) was discovered (Chen et al., 2024b), and its effectiveness was shown by Chen et al. (2024a). Our analysis theoretically clarifies why sign-based methods are less sensitive to gradient and Hessian heterogeneity than gradient-based methods.

**Optimization challenges in Transformers.**  A key aspect of Transformer optimization is the notable superiority of Adam over SGD. Zhang et al. (2020b) attributed this to the heavy-tailed gradient noise, but Kunstner et al. (2023) later challenged this, arguing that the superior performance of Adam can be attributed to sign-based characteristics rather than gradient noise, supported by full-batch experiments. Li et al. (2025) demonstrated the similarity between Adam and SignSGD in the optimization and generalization of two-layer transformers. Pan & Li (2022) showed that, in Transformers, Adam updates exhibit lower directional sharpness than SGD. Ahn et al. (2024) demonstrated that linear Transformers exhibit optimization behaviors similar to standard Transformers. Zhang et al. (2024a) revealed that the Hessian spectrum of the loss function in Transformers is heterogeneous and suggested that this heterogeneity contributes to the Adam–SGD performance gap. This observation was later supported by Ormaniec et al. (2025), who explicitly derived the Hessian of Transformers. Our work complements these studies by introducing gradient heterogeneity and theoretically explaining how it interacts with Hessian heterogeneity to affect optimization.

## 3 Preliminaries

This section introduces the notation and outlines the optimization methods relevant to our study.

### 3.1 Notation and setup

**Vectors and matrices.**  The $k$-th element of a vector $\boldsymbol{a}$ is denoted by $\boldsymbol{a}_k$, and for a matrix $\boldsymbol{A}$, we use $\boldsymbol{A}_{k,:}$, $\boldsymbol{A}_{:,l}$, and $A_{k,l}$ to denote the $k$-th row, $l$-th column, and element at $(k,l)$, respectively. When a vector or matrix is split into blocks, $[\cdot]_b$ denotes the $b$-th block. The $\ell_q$ norm is denoted by $\|\cdot\|_q$ for vectors and represents the operator norm for matrices. The all-ones vector and identity matrix of size $a$ are denoted by $\mathbf{1}_a$ and $\boldsymbol{I}_a$, respectively. The operator blockdiag($\cdot$) constructs block diagonal matrices. Derivatives are computed using the numerator layout.

**Model and training.**  We consider a classification task with $C$ classes and sample space $\mathcal{X}$. The model $\boldsymbol{f}(\cdot;\boldsymbol{\theta}) : \mathcal{X} \to \mathbb{R}^C$ is parameterized by $\boldsymbol{\theta} \in \mathbb{R}^P$, which is divided into $B$ blocks, denoted as $[\boldsymbol{\theta}]_b \in \mathbb{R}^{P_b}$, with $\sum_{b=1}^{B} P_b = P$. The training dataset $\{(\boldsymbol{x}^{(i)}, y^{(i)})\}_{i=1}^{N}$ consists of $N$ samples $\boldsymbol{x}^{(i)} \in \mathcal{X}$ and the corresponding labels $y^{(i)} \in \{1, \ldots, C\}$. The training objective is to minimize the training loss $L(\boldsymbol{\theta}) := \frac{1}{N} \sum_{i=1}^{N} \ell(\boldsymbol{f}(\boldsymbol{x}^{(i)}; \boldsymbol{\theta}), y^{(i)})$. Here, $\ell : \mathbb{R}^C \times \{1, \ldots, C\} \to \mathbb{R}$ denotes the loss function. The element-wise sign function is denoted by sign($\cdot$). The mini-batch loss is denoted by $\widehat{L}(\boldsymbol{\theta})$, and the learning rate at step $t$ is represented by $\eta_t$.

### 3.2 Optimization algorithms

**Adam.**  Adam (Kingma & Ba, 2015) is widely used in deep learning. It uses the first and second moment estimates of the gradient $\nabla \widehat{L}(\boldsymbol{\theta}_t)$, denoted as $\boldsymbol{m}_t$ and $\boldsymbol{v}_t$, computed using an exponential moving average to reduce mini-batch noise. The update is performed coordinate-wise as:

$$\boldsymbol{\theta}_{t+1} = \boldsymbol{\theta}_t - \eta_t \frac{\widehat{\boldsymbol{m}_t}}{\sqrt{\widehat{\boldsymbol{v}_t}} + \epsilon},$$

where $\widehat{\bullet}$ denotes bias correction and $\epsilon$ is a small constant for numerical stability.

**Adaptive learning rate and SignSGD.**  A key feature of Adam is its *adaptive learning rate*, which is computed in a coordinate-wise manner. When the hyperparameter $\epsilon$, which is typically set close to zero, is

ignored and the ratio $|\widehat{\boldsymbol{m}}_t/\sqrt{\widehat{\boldsymbol{v}}_t}|$ is close to 1, Adam behaves similarly to SignSGD (Balles & Hennig, 2018; Bernstein et al., 2018). SignSGD updates the parameters with momentum $\boldsymbol{m}_t$ as:

$$\boldsymbol{\theta}_{t+1} = \boldsymbol{\theta}_t - \eta_t \, \mathrm{sign}(\boldsymbol{m}_t).$$

This method has the property that the updates are invariant to the scale of the gradient. In this sense, Adam can be seen as a soft version of SignSGD. Additionally, the optimizer RMSProp (Tieleman & Hinton, 2017), which inspired Adam, was originally motivated by the idea of using the sign of the gradient in a mini-batch setting. RMSProp is similar to Adam but without the momentum term.

**SGD and gradient clipping.** SGD can also be modified to achieve scale invariance. A simple way to introduce scale invariance is to normalize the learning rate by the gradient norm, a technique known as normalized gradient descent. This method has been shown to be equivalent to gradient clipping up to a constant factor in the learning rate (Zhang et al., 2020a). Gradient clipping is commonly used to stabilize training, particularly in cases where large gradient magnitudes cause instability, and is often applied alongside other optimizers. However, a key difference between Adam and SGD is that SGD does not adapt the learning rate in a coordinate-wise manner.

**Steepest descent direction.** SGD and SignSGD can be interpreted as updating in the direction of *steepest descent* (Boyd & Vandenberghe, 2004; Xie & Li, 2024; Bernstein & Newhouse, 2024):

$$\Delta_t \in \underset{\|\Delta\| \le 1}{\arg\min} \, \nabla \widehat{L}(\boldsymbol{\theta}_t)^\top \Delta.$$

The steepest descent direction associated with the norms $\|\cdot\|_2$ and $\|\cdot\|_\infty$ corresponds to the updates of SGD and SignSGD, respectively.

The steepest descent direction satisfies

$$\nabla \widehat{L}(\boldsymbol{\theta}_t)^\top \Delta = -\|\nabla \widehat{L}(\boldsymbol{\theta}_t)\|_*,$$

where $\|\cdot\|_*$ denotes the dual norm of $\|\cdot\|$. Thus, evaluating the gradient using the dual norm is natural for analyzing SGD and SignSGD, as it quantifies the steepest decrease in a given descent direction under a unit-norm constraint.

## 4 Main results

In this section, we theoretically analyze optimization methods. We first introduce the setting, assumptions (Section 4.1), and complexity measures (Section 4.2), and then examine gradient–Hessian correlations (Section 4.3). Next, we derive upper bounds on the iteration complexity in deterministic (Section 4.4) and stochastic (Section 4.5) settings, together with implications for the learning rate of SignSGD. Finally, we investigate gradient heterogeneity in Transformers (Section 4.7). Overall, our analysis suggests that heterogeneity across parameter blocks, a characteristic of Transformers, contributes to the Adam–SGD performance gap.

### 4.1 Setting and assumptions

**Gradient-based and sign-based sequences.** Kunstner et al. (2023) showed that in full-batch settings without gradient noise, SignSGD performs similarly to Adam and outperforms SGD. This suggests that the performance gap between Adam and SGD arises from differences between SignSGD and SGD. Other studies have also used SignSGD as a proxy for Adam in their analyses (Balles & Hennig, 2018; Li et al., 2025; Kunstner et al., 2024).

On the basis of these insights, we analyze the difference between parameter sequences $\{\boldsymbol{\theta}_t^{\mathrm{Grad}}\}_{t=0}^\infty$ and $\{\boldsymbol{\theta}_t^{\mathrm{Sign}}\}_{t=0}^\infty$, referred to as the gradient-based and sign-based sequences, respectively. These sequences

correspond to updates performed by gradient-based and sign-based optimization. In deterministic settings, these updates are defined as follows:

$$\boldsymbol{\theta}_{t+1}^{\mathrm{Grad}} = \boldsymbol{\theta}_t^{\mathrm{Grad}} - \eta_t \nabla L(\boldsymbol{\theta}_t^{\mathrm{Grad}}),$$
$$\boldsymbol{\theta}_{t+1}^{\mathrm{Sign}} = \boldsymbol{\theta}_t^{\mathrm{Sign}} - \eta_t \operatorname{sign}(\nabla L(\boldsymbol{\theta}_t^{\mathrm{Sign}})).$$

In the stochastic setting, the loss $L$ is replaced with the mini-batch loss $\widehat{L}$.

We consider fine-tuning settings, in which the parameter $\boldsymbol{\theta}$ can typically be assumed to remain within a region $\mathcal{R}_{\mathrm{FT}}$ throughout training. This assumption restricts $\boldsymbol{\theta}$ to the localized region $\mathcal{R}_{\mathrm{FT}}$, allowing further assumptions to be applied within this region.

**Assumption 4.1** (Fine-tuning). The parameter $\boldsymbol{\theta}$ remains within the region $\mathcal{R}_{\mathrm{FT}}$ throughout training and there exists $\boldsymbol{\theta}_* \in \mathcal{R}_{\mathrm{FT}}$ such that $L_* := L(\boldsymbol{\theta}_*) = \min_{\boldsymbol{\theta} \in \mathcal{R}_{\mathrm{FT}}} L(\boldsymbol{\theta})$.

We assume Lipschitz continuity of the Hessian matrix, a standard assumption in nonconvex optimization analysis (Nesterov, 2013). Although the algorithms studied in this study are first-order methods, this assumption is used only in the analysis to obtain a second-order descent bound with a controlled third-order remainder. This enables us to capture the effect of Hessian heterogeneity in the analysis.

**Assumption 4.2** (Lipschitz continuity (Nesterov, 2013)). Within the region $\mathcal{R}_{\mathrm{FT}}$, the loss function $L$ is twice differentiable, and its Hessian matrix is $\rho_H$-Lipschitz continuous

$$\|\nabla^2 L(\boldsymbol{\theta}) - \nabla^2 L(\boldsymbol{\theta}')\|_2 \leq \rho_H \|\boldsymbol{\theta} - \boldsymbol{\theta}'\|_2.$$

Additionally, empirical studies have shown that Hessian matrices of deep learning models often exhibit a near-block-diagonal structure (Maes et al., 2024; Kunstner et al., 2024; Collobert, 2004; Zhang et al., 2024a; Zhao et al., 2025). The block-diagonal approximation is also used in optimization methods (Martens & Grosse, 2015; Zhang et al., 2017). Thus, we assume that the Hessian matrix of the loss function is close to block-diagonal.

**Assumption 4.3** (Near block-diagonal Hessian). Within the region $\mathcal{R}_{\mathrm{FT}}$, the Hessian matrix can be approximated by a block-diagonal matrix with an approximation error $\delta_D$:

$$\|\nabla^2 L(\boldsymbol{\theta}) - \nabla^2 L_D(\boldsymbol{\theta})\|_2 \leq \delta_D, \tag{1}$$

for all $\boldsymbol{\theta} \in \mathcal{R}_{\mathrm{FT}}$, where

$$\nabla^2 L_D(\boldsymbol{\theta}) := \operatorname{blockdiag}(\{[\nabla^2 L(\boldsymbol{\theta})]_b\}_{b=1}^B)$$

represents the block-diagonal approximation.

Note that in Eq. (1), the left-hand side is bounded above by the sum of squared elements in the non-diagonal blocks, following the relationship between $\|\cdot\|_2$ and the Frobenius norm.

## 4.2 Gradient heterogeneity and complexity measure

**Gradient heterogeneity.** We define *gradient heterogeneity* as the disparity in gradient norms across different parameter blocks, $\{\|[\nabla L(\boldsymbol{\theta})]_b\|_2\}_{b=1}^B$.

This concept complements *Hessian heterogeneity*, introduced by Zhang et al. (2024a) (referred to as "block heterogeneity" in their paper), which is defined in terms of differences in the Hessian spectrum and is generally more difficult to analyze empirically than gradient heterogeneity. We characterize gradient heterogeneity quantitatively through visualizations (Figures 3 and S.2) and Gini coefficients (Table S.7), offering concrete measures.

**Weighted Hessian norms.** To analyze the complexity of optimization, we define the following two measures.

**Definition 4.4** (Weighted Hessian norms). The gradient-weighted Hessian norm $\Lambda_G$ and parameter-weighted Hessian norm $\Lambda_P$ are defined as:

$$\Lambda_G := \sup_{\boldsymbol{\theta} \in \mathcal{R}_{\mathrm{FT}}^+} \sum_{b=1}^{B} \frac{\|[\nabla L(\boldsymbol{\theta})]_b\|_2^2}{\|\nabla L(\boldsymbol{\theta})\|_2^2} \|[\nabla^2 L(\boldsymbol{\theta})]_b\|_2,$$

$$\Lambda_P := \sup_{\boldsymbol{\theta} \in \mathcal{R}_{\mathrm{FT}}} \sum_{b=1}^{B} \frac{P_b}{P} \|[\nabla^2 L(\boldsymbol{\theta})]_b\|_2.$$

Here, we define $\mathcal{R}_{\mathrm{FT}}^+ := \{\boldsymbol{\theta} \in \mathcal{R}_{\mathrm{FT}} : \|\nabla L(\boldsymbol{\theta})\|_2 > 0\}$.

We define $\Lambda_G$ over $\mathcal{R}_{\mathrm{FT}}^+$ to avoid the degenerate stationary case; this does not affect our iteration-complexity analysis.

In this definition, $\Lambda_G$ weights the operator norm of each Hessian block by the squared gradient norm of the corresponding block, while $\Lambda_P$ weights it by the parameter dimension. The definitions ensure that the weights of all Hessian blocks sum to 1, as shown by the equalities: $\sum_{b=1}^{B} \|[\nabla L(\boldsymbol{\theta})]_b\|_2^2 / \|\nabla L(\boldsymbol{\theta})\|_2^2 = \sum_{b=1}^{B} P_b/P = 1$. Therefore, $\Lambda_G$ and $\Lambda_P$ can be interpreted as weighted averages of the block-wise Hessian norms under two different weighting schemes.

These quantities are introduced to bound the local curvature term in the one-step descent bound, $\Delta_t^\top \nabla^2 L(\theta_t) \Delta_t$, where $\Delta_t$ denotes the update direction. The effective curvature depends not only on the Hessian blocks but also on how the update direction distributes its magnitude across blocks. For the gradient-based sequence, $\Delta_t = \nabla L(\theta_t)$, and the local curvature term is bounded by $\Lambda_G$. For the sign-based sequence, $\Delta_t = \mathrm{sign}(\nabla L(\theta_t))$, whose magnitude is independent of the gradient norm, and the corresponding curvature term is bounded by $\Lambda_P$. Thus, $\Lambda_G$ and $\Lambda_P$ characterize the effective curvature encountered by gradient-based and sign-based updates, respectively. A more detailed explanation and illustration are provided in Appendix A and Figure S.1.

### 4.3 Gradient-Hessian correlation

As shown in Figure 1, large Hessian operator norms $\|[\nabla^2 L(\boldsymbol{\theta})]_b\|_2$ are often associated with large gradient magnitudes $\|[\nabla L(\boldsymbol{\theta})]_b\|_2$. In contrast, no such correlation is observed between the Hessian operator norm $\|[\nabla^2 L(\boldsymbol{\theta})]_b\|_2$ and the parameter dimension $P_b$, as detailed in Appendix D.4. Under gradient–Hessian correlation, large gradient heterogeneity leads to an increase in $\Lambda_G$, whereas $\Lambda_P$ remains relatively small.

**Approximate explanation.** If the loss function $L$ is approximated in the region $\mathcal{R}_{\mathrm{FT}}$ by a second-order Taylor expansion around the optimum $\boldsymbol{\theta}_* \in \mathcal{R}_{\mathrm{FT}}$, where $\nabla L(\boldsymbol{\theta}_*)$ is close to $\mathbf{0}$, and the Hessian matrix is assumed to be block-diagonal, the following inequality approximately holds:

$$\|[\nabla L(\boldsymbol{\theta})]_b\|_2 \le \|[\nabla^2 L(\boldsymbol{\theta}_*)]_b\|_2 \|\delta_{\boldsymbol{\theta}}\|_2,$$

where $\delta_{\boldsymbol{\theta}} = \boldsymbol{\theta} - \boldsymbol{\theta}_*$. This inequality suggests a positive correlation between the gradient norm and the Hessian norm.

**Support from prior studies.** This gradient–Hessian correlation was observed or assumed in previous studies. For instance, Zhang et al. (2024a); Jiang et al. (2024) demonstrated the relationship between $|\nabla L(\boldsymbol{\theta})_i|$ and $|\nabla^2 L(\boldsymbol{\theta})_{i,i}|$. Additionally, the $(L_0, L_1)$-smoothness assumption (Zhang et al., 2020a) and its coordinate-wise generalization (Crawshaw et al., 2022) reflect this correlation.

### 4.4 Complexity bound

To analyze optimization algorithms, we define a complexity measure inspired by Carmon et al. (2020); Zhang et al. (2020a); Crawshaw et al. (2022). This measure reflects the number of parameter updates needed to achieve a sufficiently small gradient norm, with higher complexity indicating slower convergence.

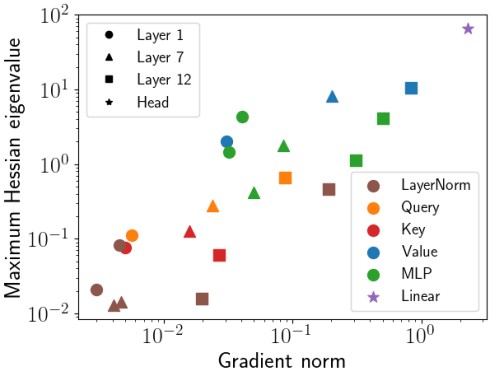 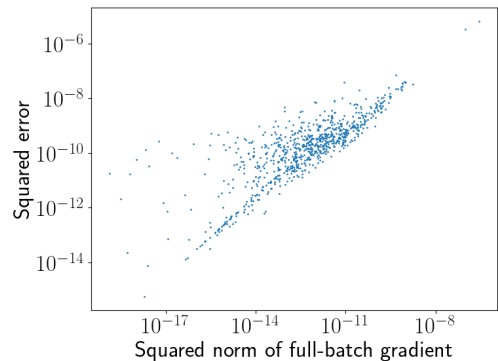

Figure 1: **Correlation between gradient norm and maximum Hessian eigenvalue.** Each point denotes the mean value computed over a parameter block (pre-trained RoBERTa on RTE).

Figure 2: **Correlation between the full-batch gradient and gradient error.** Each point represents the absolute values of a coordinate (pre-trained RoBERTa on RTE).

**Definition 4.5** (Iteration complexity)**.** We define the iteration complexity of a parameter sequence $\{\boldsymbol{\theta}_t\}_{t=0}^{\infty}$ for $\boldsymbol{\theta}_t \in \mathbb{R}^P$ with the loss function $L$ and the norm $\|\cdot\|_q$:

$$\mathcal{T}_{\varepsilon}(\{\boldsymbol{\theta}_t\}_{t=0}^{\infty}, L, \|\cdot\|_q) := \inf\{t \in \mathbb{N} \mid \mathcal{C}_{\varepsilon}(t)\},$$

where the condition $\mathcal{C}_{\varepsilon}(t)$ is defined as follows.

In the deterministic setting, $\mathcal{C}_{\varepsilon}(t)$ is defined as:

$$\|\nabla L(\boldsymbol{\theta}_t)\|_q \le P^{\frac{1}{q}} \varepsilon.$$

In the stochastic setting, $\mathcal{C}_{\varepsilon}(t)$ is defined as:

$$\mathbb{P}\big(\forall s \le t, \|\nabla L(\boldsymbol{\theta}_s)\|_q \ge P^{\frac{1}{q}} \varepsilon\big) \le \frac{1}{2}.$$

Compared with the complexity definitions in previous studies, we introduce a distinction in the choice of norms and a normalization term $P^{\frac{1}{q}}$ to ensure dimensional consistency across different norms.

Using this measure, we derive complexity bounds in deterministic (i.e., full-batch) settings. The parameter $\zeta_0 \in (0, 1)$ controls the range of learning rates.

**Theorem 4.6** (Deterministic setting)**.** *Assume $\delta_D < \min(\Lambda_G, \Lambda_P)/3$. Then, the iteration complexities in the deterministic setting are bounded as follows.*

*For the gradient-based sequence, suppose that $\varepsilon < \frac{\Lambda_G^2}{\rho_H \sqrt{P}}$ holds and that the learning rate at time $t$ satisfies $\eta_t = \zeta_t \min(\frac{1}{\Lambda_G}, \frac{1}{\sqrt{\rho_H \|\nabla L(\boldsymbol{\theta}_t^{Grad})\|_2}})$, where $\zeta_t \in [\zeta_0, 1]$, we have*

$$\mathcal{T}_{\varepsilon}(\{\boldsymbol{\theta}_t^{Grad}\}_{t=0}^{\infty}, L, \|\cdot\|_2) \le \frac{6(L(\boldsymbol{\theta}_0) - L_*)}{P \varepsilon^2 \zeta_0} \Lambda_G.$$

*For the sign-based sequence, suppose that $\varepsilon < \frac{\Lambda_P^2}{\rho_H \sqrt{P}}$ holds and that the learning rate at time $t$ satisfies $\eta_t = \zeta_t \min(\frac{\|\nabla L(\boldsymbol{\theta}_t^{Sign})\|_1}{\Lambda_P P}, \sqrt{\frac{\|\nabla L(\boldsymbol{\theta}_t^{Sign})\|_1}{\rho_H P^{3/2}}})$, where $\zeta_t \in [\zeta_0, 1]$, we have*

$$\mathcal{T}_{\varepsilon}(\{\boldsymbol{\theta}_t^{Sign}\}_{t=0}^{\infty}, L, \|\cdot\|_1) \le \frac{6(L(\boldsymbol{\theta}_0) - L_*)}{P \varepsilon^2 \zeta_0} \Lambda_P.$$

We provide the proof and its intuition in Appendix A.

The iteration complexity of the gradient-based and sign-based sequences is evaluated using the norms $\|\cdot\|_2$ and $\|\cdot\|_1$, respectively. This choice of norms is justified because they correspond to the dual norms that determine the steepest descent direction, as discussed in Section 3.2. For completeness, we also provide a common-$\ell_2$ comparison for SignSGD in Appendix G; this comparison is less favorable to SignSGD because $\ell_\infty$-steepest descent is not naturally measured by the $\ell_2$ stationarity criterion. Block-wise gradient normalization, as used in layer-wise adaptive methods (Zhang et al., 2024b; Zhao et al., 2025; Glentis et al., 2025), can likewise mitigate gradient heterogeneity; we extend our deterministic analysis to this setting in Appendix H.

**Gradient heterogeneity can increase the iteration complexity of the gradient-based sequence.** The theorem indicates that the iteration complexity of the gradient-based and sign-based sequences is characterized by $\Lambda_G$ and $\Lambda_P$, respectively. As discussed earlier, under gradient–Hessian correlation, large gradient heterogeneity leads to a large $\Lambda_G$. Consequently, the iteration complexity of the gradient-based sequence can surpass that of the sign-based sequence under such conditions.

**Connection to Zhang et al. (2024a).** Zhang et al. (2024a) show that the Adam–SGD gap arises from Hessian heterogeneity. This finding is consistent with our theoretical results (Theorem 4.6), and our analysis further explains this gap by taking gradient heterogeneity into account.

### 4.5 Stochastic setting

In practice, optimization is performed in a stochastic setting, where the gradient is estimated using a mini-batch. In this setting, we add the assumptions about noise, defined as the difference between the full-batch and mini-batch gradient.

**Assumption 4.7** (Noise). For all $\boldsymbol{\theta} \in \mathcal{R}_{\text{FT}}$, there exist constants $\sigma_3, \sigma_2, \tau_3, \tau_2 \geq 0$ such that:

$$\mathbb{E}[\nabla \widehat{L}(\boldsymbol{\theta})] = \nabla L(\boldsymbol{\theta}), \tag{2}$$

$$\mathbb{E}[\|\nabla \widehat{L}(\boldsymbol{\theta}) - \nabla L(\boldsymbol{\theta})\|_2^3] \leq \sigma_3 \|\nabla L(\boldsymbol{\theta})\|_2^3 + \tau_3, \tag{3}$$

and for all $i \in \{1, \ldots, P\}$,

$$\mathbb{E}[|\nabla \widehat{L}(\boldsymbol{\theta})_i - \nabla L(\boldsymbol{\theta})_i|^2] \leq \sigma_2 |\nabla L(\boldsymbol{\theta})_i|^2 + \tau_2. \tag{4}$$

The assumption in Eq. (2) is standard in stochastic optimization (Bernstein et al., 2018). Eq. (3) is introduced because our analysis relies on a second-order descent bound under the Hessian-Lipschitz assumption (Assumption 4.2), which contains a third-order remainder term (Lemma A.1). To control this term in the stochastic setting, we require a bound on the third-order moment of the gradient noise norm. Eq. (4) models the coordinate-wise correlation between the gradient noise and the gradient, which is empirically supported by Figure 2. The additive constants $\tau_2$ and $\tau_3$ capture gradient-independent noise floors; setting $\tau_2 = \tau_3 = 0$ recovers the relative-noise model used in the original analysis.

The sign-based sequence requires the stochastic gradient signs to be reliable under mini-batch noise. We therefore introduce the following sign-reliability condition.

**Assumption 4.8** (Sign reliability). There exists a constant $q \in [0, 1/2)$ such that for all $\boldsymbol{\theta} \in \mathcal{R}_{\text{FT}}$ with $\|\nabla L(\boldsymbol{\theta})\|_1 > 0$,

$$\sum_{i=1}^{P} \frac{|\nabla L(\boldsymbol{\theta})_i|}{\|\nabla L(\boldsymbol{\theta})\|_1} \mathbb{P}\Big(\text{sign}(\nabla \widehat{L}(\boldsymbol{\theta})_i) \neq \text{sign}(\nabla L(\boldsymbol{\theta})_i) \mid \boldsymbol{\theta}\Big) \leq q.$$

Empirically, the weighted sign mismatch rates estimated from sampled coordinates are below $1/2$ on average in all analyzed settings, providing evidence consistent with this assumption; see Appendix D.2.

Using these assumptions, we establish the complexity bounds for the stochastic setting, where $\zeta_0 \in (0, 1)$ controls the range of learning rates as in the deterministic setting.

**Theorem 4.9** (Stochastic setting). *Assume $\delta_D < \min(\Lambda_G, \Lambda_P)/3$. Then, the iteration complexities in the stochastic setting are bounded as follows.*

*For the gradient-based sequence, define*

$$\varepsilon_{\text{noise}}^2 := \frac{8\tau_2}{\zeta_0(1+\sigma_2)} + \frac{8\rho_H\tau_3}{\zeta_0(1+\sigma_2)^2\Lambda_G^2 P}. \tag{5}$$

*Suppose that $\varepsilon_{\text{noise}}^2 < \varepsilon^2 < \frac{(1+\sigma_2)^2\Lambda_G^2}{4(1+\sigma_3)\rho_H\sqrt{P}}$ holds and that the learning rate at time $t$ satisfies $\eta_t = \zeta_t \min\left(\frac{1}{(1+\sigma_2)\Lambda_G}, \frac{1}{2\sqrt{(1+\sigma_3)\rho_H\|\nabla L(\boldsymbol{\theta}_t^{Grad})\|_2}}\right)$, where $\zeta_t \in [\zeta_0, 1]$, we have*

$$\mathcal{T}_\varepsilon(\{\boldsymbol{\theta}_t^{Grad}\}_{t=0}^\infty, L, \|\cdot\|_2) \leq \frac{12(1+\sigma_2)\Lambda_G(L(\boldsymbol{\theta}_0) - L_*)}{\zeta_0 P(\varepsilon^2 - \varepsilon_{\text{noise}}^2)}.$$

*For the sign-based sequence, suppose that Assumption 4.8 holds, $\varepsilon < \frac{5\Lambda_P^2}{3(1-2q)\rho_H\sqrt{P}}$, and that the learning rate at time $t$ satisfies $\eta_t = \zeta_t \min\left(\frac{3(1-2q)\|\nabla L(\boldsymbol{\theta}_t^{Sign})\|_1}{5\Lambda_P P}, \sqrt{\frac{3(1-2q)\|\nabla L(\boldsymbol{\theta}_t^{Sign})\|_1}{5\rho_H P^{3/2}}}\right)$, where $\zeta_t \in [\zeta_0, 1]$, we have*

$$\mathcal{T}_\varepsilon(\{\boldsymbol{\theta}_t^{Sign}\}_{t=0}^\infty, L, \|\cdot\|_1) \leq \frac{20(L(\boldsymbol{\theta}_0) - L_*)}{3(1-2q)^2 P\varepsilon^2\zeta_0}\Lambda_P.$$

The proof of this theorem is provided in Appendix A.4. This theorem shows that stochasticity affects both sequences through multiplicative factors, including the sign-reliability factor $q$ for the sign-based sequence, while the gradient-based sequence is additionally subject to a noise floor induced by the additive noise terms $\tau_2$ and $\tau_3$. These terms introduce a threshold $\varepsilon_{\text{noise}}$, below which the theorem does not guarantee reaching an $\varepsilon$-stationary point. Nevertheless, despite these additional stochastic effects, the distinction between $\Lambda_G$ and $\Lambda_P$ remains central to the comparison between the two sequences, as in the deterministic setting.

### 4.6 Consistency with existing views of SignSGD

The learning-rate condition in Theorem 4.6 is

$$\eta_t = \zeta_t \min\left(\frac{\|\nabla L(\boldsymbol{\theta}_t^{\text{Sign}})\|_1}{\Lambda_P P}, \sqrt{\frac{\|\nabla L(\boldsymbol{\theta}_t^{\text{Sign}})\|_1}{\rho_H P^{3/2}}}\right).$$

Both terms scale monotonically with $\|\nabla L(\boldsymbol{\theta}_t^{\text{Sign}})\|_1$, indicating that larger $\ell_1$-norm gradients permit larger step sizes under the sufficient conditions of the theorem.

In the fine-tuning regime, where gradients are typically small, the linear term dominates whenever

$$\|\nabla L(\boldsymbol{\theta}_t^{\text{Sign}})\|_1 \leq \frac{\Lambda_P^2}{\rho_H}\sqrt{P}.$$

In this regime, the sufficient condition reduces to a step size proportional to the $\ell_1$-norm of the gradient,

$$\eta_t \propto \|\nabla \hat{L}(\boldsymbol{\theta}_t^{\text{Sign}})\|_1.$$

This dependence is consistent with existing geometric interpretations of SignSGD. In particular, $\ell_1$-scaled step sizes correspond to steepest descent with respect to the $\ell_\infty$-norm (Appendix I.1) (Balles et al., 2020; Bernstein & Newhouse, 2024). The same scaling is also recovered as the optimal step size in our quadratic analysis (Appendix D.11).

### 4.7 Gradient heterogeneity in Transformers

Transformers exhibit substantially greater parameter heterogeneity than other architectures (Zhang et al., 2024a; Cui & Wang, 2024), as confirmed by our experiments (Figure 3). To better understand the origin of gradient heterogeneity in Transformers, we investigate the role of layer normalization. In particular, we compare Post-LN and Pre-LN architectures and examine how their normalization schemes affect the distribution of gradient magnitudes across parameter blocks.

**Post-LN and Pre-LN.** In Transformers, residual connections and layer normalizations are combined with multi-head attention and feed-forward networks. The two main Transformer architectures are post-layer normalization (Post-LN), where the residual connection is followed by the layer normalization, and pre-layer normalization (Pre-LN), where the layer normalization precedes the residual connection. Pre-LN is known for greater stability (Wang et al., 2019b; Xiong et al., 2020; Takase et al., 2022).

**Jacobian of Transformers.** The Jacobians of a Transformer layer with Pre-LN and Post-LN are expressed as:

$$\boldsymbol{J}_{\text{Pre-LN}} = (\boldsymbol{J}_{\text{FFN}}\boldsymbol{J}_{\text{LN}} + \boldsymbol{I}_{nd})(\boldsymbol{J}_{\text{ATT}}\boldsymbol{J}_{\text{LN}} + \boldsymbol{I}_{nd}) \tag{6}$$

$$\boldsymbol{J}_{\text{Post-LN}} = \boldsymbol{J}_{\text{LN}}(\boldsymbol{J}_{\text{FFN}} + \boldsymbol{I}_{nd})\boldsymbol{J}_{\text{LN}}(\boldsymbol{J}_{\text{ATT}} + \boldsymbol{I}_{nd}), \tag{7}$$

where $\boldsymbol{J}_{\text{ATT}}$ and $\boldsymbol{J}_{\text{FFN}}$ denote the Jacobians of the self-attention and feed-forward network modules, respectively. For simplicity, the evaluation points of the Jacobians are omitted. The Jacobian of the layer normalization is represented by $\boldsymbol{J}_{\text{LN}}$, calculated for an input $\boldsymbol{X} \in \mathbb{R}^{n \times d}$ as:

$$\boldsymbol{J}_{\text{LN}}(\boldsymbol{X}) = \text{blockdiag}(\{\boldsymbol{L}_i(\boldsymbol{X})\}_{i=1}^n), \tag{8}$$

where each block $\boldsymbol{L}_i \in \mathbb{R}^{d \times d}$ is defined as:

$$\boldsymbol{L}_i(\boldsymbol{X}) \coloneqq \frac{\sqrt{d}}{\|\widetilde{\boldsymbol{X}_{i,:}}\|_2}\left(\boldsymbol{I}_d - \frac{\widetilde{\boldsymbol{X}_{i,:}}\widetilde{\boldsymbol{X}_{i,:}}^\top}{\|\widetilde{\boldsymbol{X}_{i,:}}\|_2^2}\right)\left(\boldsymbol{I}_d - \frac{\boldsymbol{1}\boldsymbol{1}^\top}{d}\right),$$

and $\widetilde{\boldsymbol{X}_{i,:}} \coloneqq \boldsymbol{X}_{i,:}(\boldsymbol{I}_d - \frac{\boldsymbol{1}\boldsymbol{1}^\top}{d})$. These derivations are provided in Appendix B.

**Greater gradient heterogeneity in Post-LN.** Equation (8) shows that the Jacobian of layer normalization, $\boldsymbol{J}_{\text{LN}}$, depends on the input, causing variations in its scale across tokens and layers. From Eqs. (6) and (7), we observe that in Post-LN, $\boldsymbol{J}_{\text{LN}}$ appears outside the residual branch and is multiplied with broad components of the layer Jacobian. This multiplicative placement can compound input-dependent scale variation more directly than in Pre-LN, where the identity path bypasses the normalization Jacobian. We therefore expect Post-LN to exhibit greater gradient heterogeneity, consistent with Table 2. Further discussion of gradient heterogeneity in Transformers, particularly in the attention mechanism, is provided in Appendix F.

## 5 Numerical evaluation

We numerically evaluate the following claims.

- Gradient heterogeneity is pronounced in Transformers and is influenced by the position of layer normalization (Section 5.2).

- SGD exhibits slower train-loss convergence under gradient heterogeneity compared with adaptive optimizers such as Adam (Section 5.3).

We provide details of the experimental setup and figures in Appendix C and additional results in Appendix D.

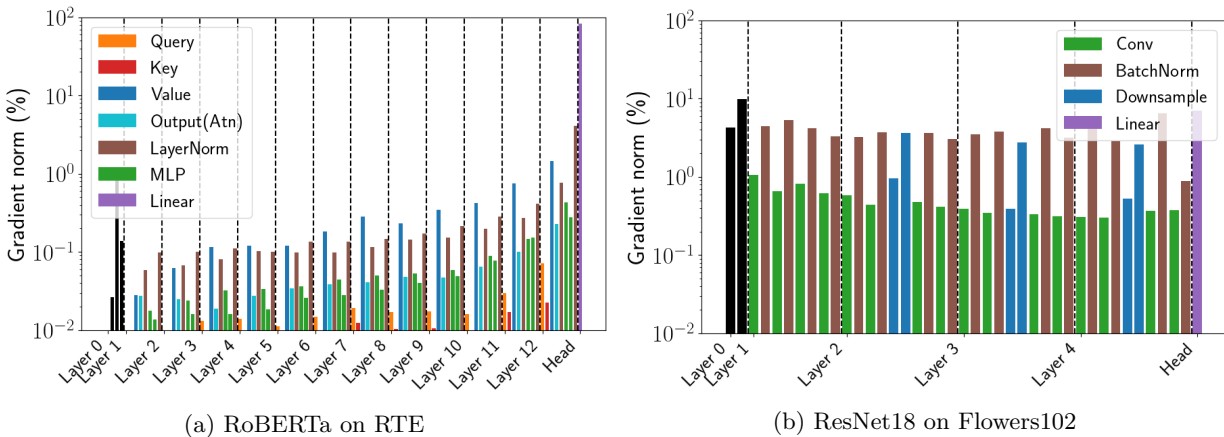

(a) RoBERTa on RTE  (b) ResNet18 on Flowers102

Figure 3: **Transformers exhibit large gradient heterogeneity.** Gradient norms of individual parameters in pre-trained models.

## 5.1 Experimental setup

**Datasets and models.** We used a total of nine datasets and three pre-trained models obtained from public sources. For NLP tasks, we used four datasets from SuperGLUE (Wang et al., 2019a) (BoolQ, CB, RTE, and WiC) and three datasets from GLUE (Wang et al., 2018) (CoLA, MRPC, and SST-2) with the RoBERTa-Base model (Liu et al., 2019). For vision tasks, we used the Flowers102 (Nilsback & Zisserman, 2008) and FGVC-Aircraft (Aircraft) (Maji et al., 2013) datasets with ViT-Base (Dosovitskiy et al., 2021) and ResNet18 (He et al., 2016).

**Training.** We compared optimizers with momentum (default) and without momentum (w/o M), as well as SignSGD with $\ell_1$-scaled learning rates (SignSGD (S); Section 4.6). The learning rates were tuned, gradient clipping was applied to SGD for stability, and the learning-rate schedule was fixed within each domain. For SGD, gradient clipping is equivalent to normalized gradient descent up to a constant factor (Zhang et al., 2020a), so it addresses global scale but not the coordinate-wise reweighting that distinguishes Adam-like and sign-based methods. The gradient-dependent factor in the learning-rate condition of Theorem 4.6 controls the global update scale as the gradient norm changes. In the experiments, tuned learning rates, schedules, and clipping play this scale-control role in a practical training setup. All models were fine-tuned from pre-trained weights.

## 5.2 Gradient heterogeneity

Figure 3 compares RoBERTa and ResNet18 and shows substantially higher gradient heterogeneity in the Transformer model. In RoBERTa, gradients are smaller near input layers compared to output layers, consistent with our analysis in Section 4.7.

**Effect of layer normalization.** Table 2 shows Gini coefficients for different normalization placements in RoBERTa on RTE. Post-LN shows higher heterogeneity than Pre-LN, consistent with our analysis (Section 4.7). Pre-trained weights are only available for Post-LN.

## 5.3 Training curves

**Limitations of SGD under gradient heterogeneity.** As shown in Figure 4, all optimizers successfully train ViT (and ResNet; see Figure S.7), whereas SGD exhibits substantially slower train-loss convergence on RoBERTa in high-heterogeneity settings such as RTE, even though it can reach similar final train losses on other RoBERTa tasks (e.g., CB; see Figure S.7). This pattern highlights the challenge posed by gradient heterogeneity. In contrast, SignSGD (S) reliably trains both ViT and RoBERTa. These observations are

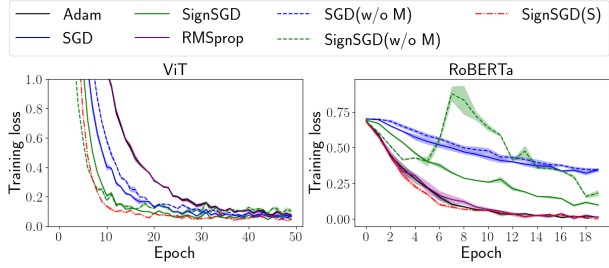

Figure 4: **RoBERTa is difficult to optimize with SGD.** Training loss curves for ViT on Flowers102 (left) and RoBERTa on RTE (right). Shaded regions denote interquartile ranges.

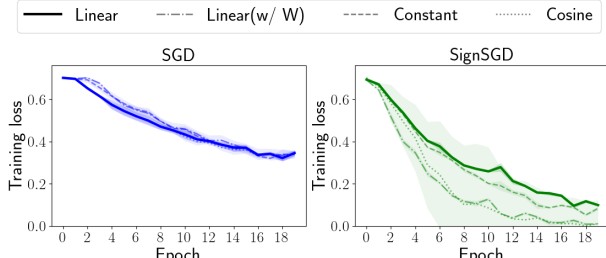

Figure 5: **Learning-rate schedulers improve SignSGD but have limited effect on SGD.** Training loss curves for RoBERTa on RTE. "w/ W" denotes with warmup.

Table 2: **Post-LN increases gradient heterogeneity.** Higher Gini coefficient indicates greater heterogeneity.

| Norm Type | Model state | Gini Coeff. |
|---|---|---|
| Pre-LN | Random init. | $0.880 \pm 0.004$ |
| Post-LN | Random init. | $0.941 \pm 0.012$ |
| Post-LN | Pre-trained | $0.944 \pm 0.005$ |

consistent with our theoretical analysis in Theorems 4.6 and 4.9. Additionally, the final training losses are similar between momentum and no-momentum variants of SGD and SignSGD, and Adam performs similarly to RMSProp, suggesting that adaptive learning rates, rather than momentum, are the primary cause of the performance gap (Kunstner et al., 2023). Note that the RTE dataset, which has two almost balanced classes, rules out the heavy-tailed class imbalance as an explanation for the Adam–SGD gap (Kunstner et al., 2024).

**Effectiveness of learning-rate schedules.** In NLP tasks, we use linear learning-rate scheduling by default. To assess whether SGD's slow convergence on RTE is due to its schedule, we train RoBERTa with various schedules. In Figure 5, learning-rate schedules do not improve SGD, while SignSGD benefits significantly from the appropriate schedules, achieving performance comparable to Adam.

## 6 Conclusion

We derive upper bounds on the iteration complexity of SGD and SignSGD to better understand the optimization gap between gradient-based and Adam-like methods. Since Adam shares coordinate-wise update characteristics with SignSGD, we use SignSGD as an analytically tractable proxy for Adam-like behavior. Our results suggest gradient and Hessian heterogeneity as key factors underlying the performance gap between Adam and SGD in Transformer models. Our analysis uses the fact that SGD and SignSGD correspond to steepest descent under different norms, yielding implications for learning rate scaling of SignSGD. We further show that gradient heterogeneity is particularly pronounced in Post-LN Transformers. Empirically, SGD converges more slowly under large gradient heterogeneity, whereas SignSGD, with appropriate learning-rate scheduling, achieves performance comparable to Adam.

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

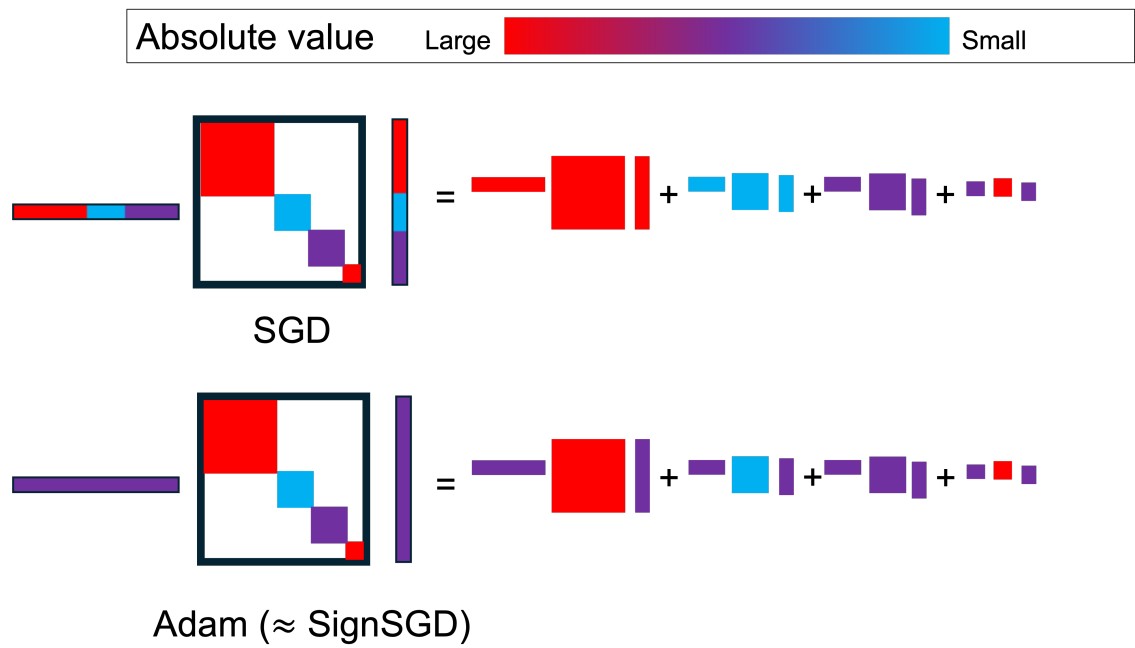

Figure S.1: Proof intuition for Theorem 4.6. This figure illustrates the key quantity $|\Delta_t^\top \nabla^2 L_D(\boldsymbol{\theta}_t)\Delta_t|$ discussed in Appendix A.1. Under gradient–Hessian correlation, gradient-based sequences (top) align large gradient norms with large Hessian operator norms, amplifying the block-diagonal quadratic form. Sign-based updates (bottom), which use unit-magnitude directions, mitigate this effect and yield more uniform contributions across blocks.

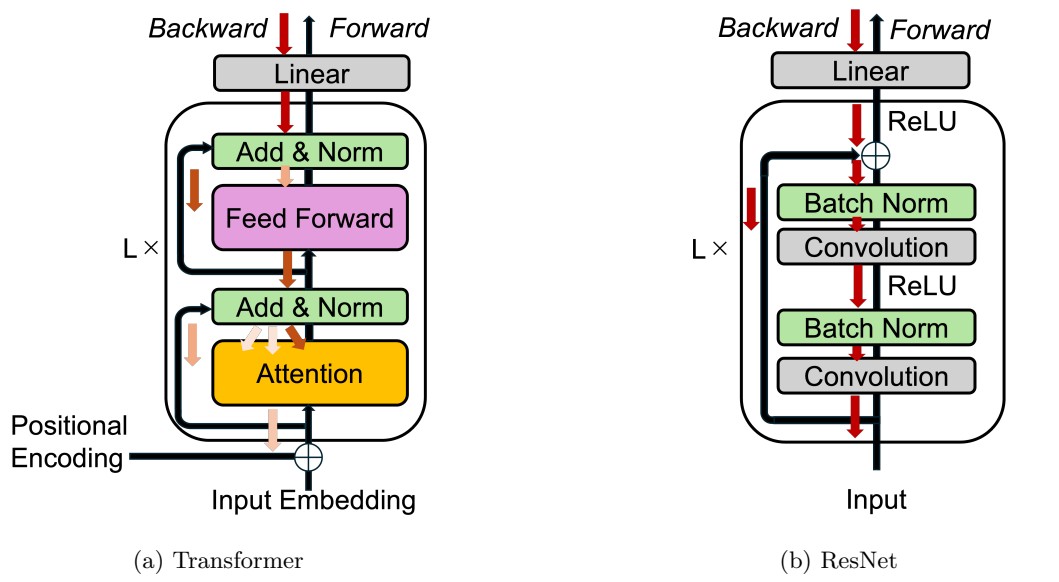

(a) Transformer          (b) ResNet

Figure S.2: Architecture and gradient heterogeneity across models. ResNets, as a representative CNN architecture, are constructed by the repetitive stacking of homogeneous parameter blocks (convolutional layers), which promotes relatively uniform gradient propagation. In contrast, Transformers involve stacking of heterogeneous parameter blocks, such as the Query, Key, Value, and output projection layers in attention as well as MLP layers, leading to uneven gradient propagation across modules and pronounced gradient heterogeneity.

# A    Proof

## A.1    Proof intuition

Here, we outline the core analysis underlying Theorem 4.6. We apply the descent lemma (Bertsekas, 1999) in our setting and construct an upper bound for the term

$$L(\boldsymbol{\theta}_{t+1}) - L(\boldsymbol{\theta}_t).$$

Under Assumption 4.2, the key quantity to be controlled is

$$|\Delta_t^\top \nabla^2 L(\boldsymbol{\theta}_t)\Delta_t|,$$

where

$$\Delta_t := \begin{cases} \nabla L(\boldsymbol{\theta}_t^{\mathrm{Grad}}) & \text{(gradient-based sequence)} \\ \mathrm{sign}(\nabla L(\boldsymbol{\theta}_t^{\mathrm{Sign}})) & \text{(sign-based sequence)} \end{cases}$$

is the update term without learning rate.

With Assumption 4.3, the dominant contribution to this quadratic form can be characterized by the block-diagonal component

$$|\Delta_t^\top \nabla^2 L_D(\boldsymbol{\theta}_t)\Delta_t|,$$

while the contribution of the off-diagonal blocks is controlled by the approximation error $\delta_D$.

For the gradient-based sequence, $\Delta_t$ coincides with the gradient itself, so parameter blocks with large gradient norms contribute more to the local curvature term. When such blocks also have large Hessian operator norms, the quadratic form can become large, leading to the gradient-weighted curvature quantity $\Lambda_G$. In contrast, for the sign-based sequence, each coordinate of $\Delta_t$ has unit magnitude, so the contribution of each block depends only on its parameter dimension rather than the gradient magnitude. This suppresses the alignment between large gradients and high-curvature blocks, resulting in the dimension-weighted quantity $\Lambda_P$. These effects are illustrated in Figure S.1. The complete proof is provided in Appendix A.3.

## A.2    Technical lemma

The following lemma is the starting point of our convergence analysis. Under the Hessian-Lipschitz assumption, it provides a second-order descent bound with a controlled third-order remainder. This form keeps the second-order term explicit, allowing us to relate the convergence behavior to Hessian heterogeneity.

**Lemma A.1.** *Under Assumption 4.2, for any $\boldsymbol{\theta}, \boldsymbol{\theta}' \in \mathbb{R}^P$, the following inequality holds:*

$$L(\boldsymbol{\theta}') - L(\boldsymbol{\theta}) \le \nabla L(\boldsymbol{\theta})^\top (\boldsymbol{\theta}' - \boldsymbol{\theta}) + \frac{1}{2}(\boldsymbol{\theta}' - \boldsymbol{\theta})^\top \nabla^2 L(\boldsymbol{\theta})(\boldsymbol{\theta}' - \boldsymbol{\theta}) + \frac{\rho_H}{6}\|\boldsymbol{\theta}' - \boldsymbol{\theta}\|_2^3.$$

*Proof.* Define $\boldsymbol{\nu}(\alpha) := \boldsymbol{\theta} + \alpha(\boldsymbol{\theta}' - \boldsymbol{\theta})$. By the $\rho_H$-Lipschitz continuity of the Hessian (Assumption 4.2), we have:

$$(\nabla L(\boldsymbol{\theta}') - \nabla L(\boldsymbol{\theta}))^\top (\boldsymbol{\theta}' - \boldsymbol{\theta})$$

$$= \int_0^1 (\boldsymbol{\theta}' - \boldsymbol{\theta})^\top \nabla^2 L(\boldsymbol{\nu}(\alpha))(\boldsymbol{\theta}' - \boldsymbol{\theta})d\alpha$$

$$= (\boldsymbol{\theta}' - \boldsymbol{\theta})^\top \nabla^2 L(\boldsymbol{\theta})(\boldsymbol{\theta}' - \boldsymbol{\theta}) + \int_0^1 (\boldsymbol{\theta}' - \boldsymbol{\theta})^\top (\nabla^2 L(\boldsymbol{\nu}(\alpha)) - \nabla^2 L(\boldsymbol{\theta}))(\boldsymbol{\theta}' - \boldsymbol{\theta})d\alpha$$

$$\le (\boldsymbol{\theta}' - \boldsymbol{\theta})^\top \nabla^2 L(\boldsymbol{\theta})(\boldsymbol{\theta}' - \boldsymbol{\theta}) + \int_0^1 \|\nabla^2 L(\boldsymbol{\nu}(\alpha)) - \nabla^2 L(\boldsymbol{\theta})\|_2 \|\boldsymbol{\theta}' - \boldsymbol{\theta}\|_2^2 d\alpha$$

$$\le (\boldsymbol{\theta}' - \boldsymbol{\theta})^\top \nabla^2 L(\boldsymbol{\theta})(\boldsymbol{\theta}' - \boldsymbol{\theta}) + \int_0^1 \rho_H \alpha \|\boldsymbol{\theta}' - \boldsymbol{\theta}\|_2^3 d\alpha$$

$$= (\boldsymbol{\theta}' - \boldsymbol{\theta})^\top \nabla^2 L(\boldsymbol{\theta})(\boldsymbol{\theta}' - \boldsymbol{\theta}) + \frac{\rho_H}{2} \|\boldsymbol{\theta}' - \boldsymbol{\theta}\|_2^3.$$

Using this inequality, we obtain:

$$
\begin{aligned}
&L(\boldsymbol{\theta}') - L(\boldsymbol{\theta}) \\
&= \int_0^1 \nabla L(\boldsymbol{\nu}(\alpha))^\top (\boldsymbol{\theta}' - \boldsymbol{\theta}) d\alpha \\
&= \nabla L(\boldsymbol{\theta})^\top (\boldsymbol{\theta}' - \boldsymbol{\theta}) + \int_0^1 (\nabla L(\boldsymbol{\nu}(\alpha)) - \nabla L(\boldsymbol{\theta}))^\top (\boldsymbol{\theta}' - \boldsymbol{\theta}) d\alpha \\
&= \nabla L(\boldsymbol{\theta})^\top (\boldsymbol{\theta}' - \boldsymbol{\theta}) + \int_0^1 (\nabla L(\boldsymbol{\nu}(\alpha)) - \nabla L(\boldsymbol{\theta}))^\top \frac{1}{\alpha}(\boldsymbol{\nu}(\alpha) - \boldsymbol{\theta}) d\alpha \\
&\leq \nabla L(\boldsymbol{\theta})^\top (\boldsymbol{\theta}' - \boldsymbol{\theta}) + \int_0^1 \frac{1}{\alpha} \left( (\boldsymbol{\nu}(\alpha) - \boldsymbol{\theta})^\top \nabla^2 L(\boldsymbol{\theta})(\boldsymbol{\nu}(\alpha) - \boldsymbol{\theta}) + \frac{\rho_H}{2} \|\boldsymbol{\nu}(\alpha) - \boldsymbol{\theta}\|_2^3 \right) d\alpha \\
&= \nabla L(\boldsymbol{\theta})^\top (\boldsymbol{\theta}' - \boldsymbol{\theta}) + \int_0^1 \left( (\boldsymbol{\theta}' - \boldsymbol{\theta})^\top \nabla^2 L(\boldsymbol{\theta})(\boldsymbol{\theta}' - \boldsymbol{\theta})\alpha + \frac{\rho_H}{2} \|\boldsymbol{\theta}' - \boldsymbol{\theta}\|_2^3 \alpha^2 \right) d\alpha \\
&= \nabla L(\boldsymbol{\theta})^\top (\boldsymbol{\theta}' - \boldsymbol{\theta}) + \frac{1}{2}(\boldsymbol{\theta}' - \boldsymbol{\theta})^\top \nabla^2 L(\boldsymbol{\theta})(\boldsymbol{\theta}' - \boldsymbol{\theta}) + \frac{\rho_H}{6} \|\boldsymbol{\theta}' - \boldsymbol{\theta}\|_2^3.
\end{aligned}
$$

$\square$

**Lemma A.2.** *For any $a, b \geq 0$, the following inequality holds:*

$$(a + b)^3 \leq 4(a^3 + b^3).$$

*Proof.* Calculating the difference between the right-hand and left-hand side, we obtain:

$$
\begin{aligned}
4(a^3 + b^3) - (a + b)^3 &= 4(a^3 + b^3) - (a^3 + 3a^2 b + 3ab^2 + b^3) \\
&= 3(a^3 + b^3) - 3a^2 b - 3ab^2 \\
&= 3(a + b)(a - b)^2 \geq 0.
\end{aligned}
$$

$\square$

### A.3 Proof of Theorem 4.6

**Theorem 4.6 is restated.** *Assume $\delta_D < \min(\Lambda_G, \Lambda_P)/3$. Then, the iteration complexities in the deterministic setting are bounded as follows.*

*For the gradient-based sequence, suppose that $\varepsilon < \frac{\Lambda_G^2}{\rho_H \sqrt{P}}$ holds and that the learning rate at time $t$ satisfies $\eta_t = \zeta_t \min(\frac{1}{\Lambda_G}, \frac{1}{\sqrt{\rho_H \|\nabla L(\boldsymbol{\theta}_t^{Grad})\|_2}})$, where $\zeta_t \in [\zeta_0, 1]$, we have*

$$\mathcal{T}_\varepsilon(\{\boldsymbol{\theta}_t^{Grad}\}_{t=0}^\infty, L, \|\cdot\|_2) \leq \frac{6(L(\boldsymbol{\theta}_0) - L_*)}{P\varepsilon^2 \zeta_0} \Lambda_G.$$

*For the sign-based sequence, suppose that $\varepsilon < \frac{\Lambda_P^2}{\rho_H \sqrt{P}}$ holds and that the learning rate at time $t$ satisfies $\eta_t = \zeta_t \min(\frac{\|\nabla L(\boldsymbol{\theta}_t^{Sign})\|_1}{\Lambda_P P}, \sqrt{\frac{\|\nabla L(\boldsymbol{\theta}_t^{Sign})\|_1}{\rho_H P^{3/2}}})$, where $\zeta_t \in [\zeta_0, 1]$, we have*

$$\mathcal{T}_\varepsilon(\{\boldsymbol{\theta}_t^{Sign}\}_{t=0}^\infty, L, \|\cdot\|_1) \leq \frac{6(L(\boldsymbol{\theta}_0) - L_*)}{P\varepsilon^2 \zeta_0} \Lambda_P.$$

*Proof of gradient-based sequence.* The update rule of the gradient-based sequence in deterministic setting is $\boldsymbol{\theta}_{t+1}^{\mathrm{Grad}} = \boldsymbol{\theta}_t^{\mathrm{Grad}} - \eta_t \nabla L(\boldsymbol{\theta}_t^{\mathrm{Grad}})$. Thus, using Lemma A.1, we obtain:

$$L(\boldsymbol{\theta}_{t+1}^{\mathrm{Grad}}) - L(\boldsymbol{\theta}_t^{\mathrm{Grad}})$$

$$\leq \nabla L(\boldsymbol{\theta}_t^{\text{Grad}})^\top (\boldsymbol{\theta}_{t+1}^{\text{Grad}} - \boldsymbol{\theta}_t^{\text{Grad}}) + \frac{1}{2}(\boldsymbol{\theta}_{t+1}^{\text{Grad}} - \boldsymbol{\theta}_t^{\text{Grad}})^\top \nabla^2 L(\boldsymbol{\theta}_t^{\text{Grad}})(\boldsymbol{\theta}_{t+1}^{\text{Grad}} - \boldsymbol{\theta}_t^{\text{Grad}}) + \frac{\rho_H}{6}\|\boldsymbol{\theta}_{t+1}^{\text{Grad}} - \boldsymbol{\theta}_t^{\text{Grad}}\|_2^3$$

$$= -\eta_t\|\nabla L(\boldsymbol{\theta}_t^{\text{Grad}})\|_2^2 + \frac{\eta_t^2}{2}\nabla L(\boldsymbol{\theta}_t^{\text{Grad}})^\top \nabla^2 L(\boldsymbol{\theta}_t^{\text{Grad}})\nabla L(\boldsymbol{\theta}_t^{\text{Grad}}) + \eta_t^3\frac{\rho_H}{6}\|\nabla L(\boldsymbol{\theta}_t^{\text{Grad}})\|_2^3$$

$$= -\eta_t\|\nabla L(\boldsymbol{\theta}_t^{\text{Grad}})\|_2^2 + \frac{\eta_t^2}{2}\nabla L(\boldsymbol{\theta}_t^{\text{Grad}})^\top \nabla^2 L_D(\boldsymbol{\theta}_t^{\text{Grad}})\nabla L(\boldsymbol{\theta}_t^{\text{Grad}})$$

$$+ \frac{\eta_t^2}{2}\nabla L(\boldsymbol{\theta}_t^{\text{Grad}})^\top (\nabla^2 L(\boldsymbol{\theta}_t^{\text{Grad}}) - \nabla^2 L_D(\boldsymbol{\theta}_t^{\text{Grad}}))\nabla L(\boldsymbol{\theta}_t^{\text{Grad}}) + \eta_t^3\frac{\rho_H}{6}\|\nabla L(\boldsymbol{\theta}_t^{\text{Grad}})\|_2^3$$

$$= -\eta_t\|\nabla L(\boldsymbol{\theta}_t^{\text{Grad}})\|_2^2 + \frac{\eta_t^2}{2}\sum_b [\nabla L(\boldsymbol{\theta}_t^{\text{Grad}})]_b^\top [\nabla^2 L(\boldsymbol{\theta}_t^{\text{Grad}})]_b [\nabla L(\boldsymbol{\theta}_t^{\text{Grad}})]_b$$

$$+ \frac{\eta_t^2}{2}\nabla L(\boldsymbol{\theta}_t^{\text{Grad}})^\top (\nabla^2 L(\boldsymbol{\theta}_t^{\text{Grad}}) - \nabla^2 L_D(\boldsymbol{\theta}_t^{\text{Grad}}))\nabla L(\boldsymbol{\theta}_t^{\text{Grad}}) + \eta_t^3\frac{\rho_H}{6}\|\nabla L(\boldsymbol{\theta}_t^{\text{Grad}})\|_2^3$$

$$\leq -\eta_t\|\nabla L(\boldsymbol{\theta}_t^{\text{Grad}})\|_2^2 + \frac{\eta_t^2}{2}\sum_b \|[\nabla^2 L(\boldsymbol{\theta}_t^{\text{Grad}})]_b\|_2 \|[\nabla L(\boldsymbol{\theta}_t^{\text{Grad}})]_b\|_2^2$$

$$+ \frac{\eta_t^2}{2}\|\nabla^2 L(\boldsymbol{\theta}_t^{\text{Grad}}) - \nabla^2 L_D(\boldsymbol{\theta}_t^{\text{Grad}})\|_2 \|\nabla L(\boldsymbol{\theta}_t^{\text{Grad}})\|_2^2 + \eta_t^3\frac{\rho_H}{6}\|\nabla L(\boldsymbol{\theta}_t^{\text{Grad}})\|_2^3$$

$$\leq -\eta_t\|\nabla L(\boldsymbol{\theta}_t^{\text{Grad}})\|_2^2 + \frac{\eta_t^2}{2}\Lambda_G\|\nabla L(\boldsymbol{\theta}_t^{\text{Grad}})\|_2^2 + \frac{\eta_t^2}{2}\delta_D\|\nabla L(\boldsymbol{\theta}_t^{\text{Grad}})\|_2^2 + \eta_t^3\frac{\rho_H}{6}\|\nabla L(\boldsymbol{\theta}_t^{\text{Grad}})\|_2^3 \qquad (9)$$

$$\leq -\eta_t\|\nabla L(\boldsymbol{\theta}_t^{\text{Grad}})\|_2^2 + \frac{\eta_t}{2}\|\nabla L(\boldsymbol{\theta}_t^{\text{Grad}})\|_2^2 + \frac{\eta_t}{6}\|\nabla L(\boldsymbol{\theta}_t^{\text{Grad}})\|_2^2 + \frac{\eta_t}{6}\|\nabla L(\boldsymbol{\theta}_t^{\text{Grad}})\|_2^2$$

$$= -\frac{\eta_t}{6}\|\nabla L(\boldsymbol{\theta}_t^{\text{Grad}})\|_2^2,$$

where we use $\eta_t \leq \min\left(\frac{1}{\Lambda_G}, \frac{1}{\sqrt{\rho_H\|\nabla L(\boldsymbol{\theta}_t^{\text{Grad}})\|_2}}\right)$ and $\delta_D < \Lambda_G/3$.

Taking a telescoping sum and noting that $\boldsymbol{\theta}_0 = \boldsymbol{\theta}_0^{\text{Grad}}$ and $\eta_t \geq \zeta_0 \min\left(\frac{1}{\Lambda_G}, \frac{1}{\sqrt{\rho_H\|\nabla L(\boldsymbol{\theta}_t^{\text{Grad}})\|_2}}\right)$, we have:

$$L(\boldsymbol{\theta}_T^{\text{Grad}}) - L(\boldsymbol{\theta}_0) \leq -\frac{1}{6}\sum_{t=0}^{T-1}\eta_t\|\nabla L(\boldsymbol{\theta}_t^{\text{Grad}})\|_2^2$$

$$\leq -\frac{\zeta_0}{6}\sum_{t=0}^{T-1}\min\left(\frac{\|\nabla L(\boldsymbol{\theta}_t^{\text{Grad}})\|_2^2}{\Lambda_G}, \frac{\|\nabla L(\boldsymbol{\theta}_t^{\text{Grad}})\|_2^{3/2}}{\sqrt{\rho_H}}\right).$$

Assume that $\|\nabla L(\boldsymbol{\theta}_t^{\text{Grad}})\|_2 \geq \sqrt{P}\varepsilon$ holds for all $0 \leq t < T$. Then, using $\varepsilon < \frac{\Lambda_G^2}{\rho_H\sqrt{P}}$, we have:

$$L(\boldsymbol{\theta}_T^{\text{Grad}}) - L(\boldsymbol{\theta}_0) \leq -\frac{T\zeta_0}{6}\min\left(\frac{P\varepsilon^2}{\Lambda_G}, \frac{P^{3/4}\varepsilon^{3/2}}{\sqrt{\rho_H}}\right)$$

$$= -\frac{TP\varepsilon^2\zeta_0}{6\Lambda_G}.$$

Therefore, we have

$$T \leq \frac{6(L(\boldsymbol{\theta}_0) - L(\boldsymbol{\theta}_T^{\text{Grad}}))}{P\varepsilon^2\zeta_0}\Lambda_G$$

$$\leq \frac{6(L(\boldsymbol{\theta}_0) - L_*)}{P\varepsilon^2\zeta_0}\Lambda_G.$$

This means

$$\mathcal{T}_\varepsilon(\{\boldsymbol{\theta}_t^{\text{Grad}}\}_{t=0}^\infty, L, \|\cdot\|_2) \leq \frac{6(L(\boldsymbol{\theta}_0) - L_*)}{P\varepsilon^2\zeta_0}\Lambda_G.$$

$\square$

*Proof of sign-based sequence.* The update rule of the sign-based sequence in deterministic setting is $\boldsymbol{\theta}_{t+1}^{\mathrm{Sign}} = \boldsymbol{\theta}_t^{\mathrm{Sign}} - \eta_t \operatorname{sign}(\nabla L(\boldsymbol{\theta}_t^{\mathrm{Sign}}))$. Thus, using Lemma A.1, we obtain:

$$L(\boldsymbol{\theta}_{t+1}^{\mathrm{Sign}}) - L(\boldsymbol{\theta}_t^{\mathrm{Sign}})$$

$$\leq \nabla L(\boldsymbol{\theta}_t^{\mathrm{Sign}})^\top (\boldsymbol{\theta}_{t+1}^{\mathrm{Sign}} - \boldsymbol{\theta}_t^{\mathrm{Sign}}) + \frac{1}{2}(\boldsymbol{\theta}_{t+1}^{\mathrm{Sign}} - \boldsymbol{\theta}_t^{\mathrm{Sign}})^\top \nabla^2 L(\boldsymbol{\theta}_t^{\mathrm{Sign}})(\boldsymbol{\theta}_{t+1}^{\mathrm{Sign}} - \boldsymbol{\theta}_t^{\mathrm{Sign}}) + \frac{\rho_H}{6}\|\boldsymbol{\theta}_{t+1}^{\mathrm{Sign}} - \boldsymbol{\theta}_t^{\mathrm{Sign}}\|_2^3$$

$$= -\eta_t \|\nabla L(\boldsymbol{\theta}_t^{\mathrm{Sign}})\|_1 + \frac{\eta_t^2}{2}\operatorname{sign}(\nabla L(\boldsymbol{\theta}_t^{\mathrm{Sign}}))^\top \nabla^2 L(\boldsymbol{\theta}_t^{\mathrm{Sign}})\operatorname{sign}(\nabla L(\boldsymbol{\theta}_t^{\mathrm{Sign}})) + \eta_t^3 \frac{\rho_H}{6}\|\operatorname{sign}(\nabla L(\boldsymbol{\theta}_t^{\mathrm{Sign}}))\|_2^3$$

$$= -\eta_t \|\nabla L(\boldsymbol{\theta}_t^{\mathrm{Sign}})\|_1 + \frac{\eta_t^2}{2}\operatorname{sign}(\nabla L(\boldsymbol{\theta}_t^{\mathrm{Sign}}))^\top \nabla^2 L_D(\boldsymbol{\theta}_t^{\mathrm{Sign}})\operatorname{sign}(\nabla L(\boldsymbol{\theta}_t^{\mathrm{Sign}}))$$

$$+ \frac{\eta_t^2}{2}\operatorname{sign}(\nabla L(\boldsymbol{\theta}_t^{\mathrm{Sign}}))^\top (\nabla^2 L(\boldsymbol{\theta}_t^{\mathrm{Sign}}) - \nabla^2 L_D(\boldsymbol{\theta}_t^{\mathrm{Sign}}))\operatorname{sign}(\nabla L(\boldsymbol{\theta}_t^{\mathrm{Sign}})) + \eta_t^3 \frac{\rho_H}{6}P^{3/2}$$

$$= -\eta_t \|\nabla L(\boldsymbol{\theta}_t^{\mathrm{Sign}})\|_1 + \frac{\eta_t^2}{2}\sum_b [\operatorname{sign}(\nabla L(\boldsymbol{\theta}_t^{\mathrm{Sign}}))]_b^\top [\nabla^2 L(\boldsymbol{\theta}_t^{\mathrm{Sign}})]_b [\operatorname{sign}(\nabla L(\boldsymbol{\theta}_t^{\mathrm{Sign}}))]_b$$

$$+ \frac{\eta_t^2}{2}\operatorname{sign}(\nabla L(\boldsymbol{\theta}_t^{\mathrm{Sign}}))^\top (\nabla^2 L(\boldsymbol{\theta}_t^{\mathrm{Sign}}) - \nabla^2 L_D(\boldsymbol{\theta}_t^{\mathrm{Sign}}))\operatorname{sign}(\nabla L(\boldsymbol{\theta}_t^{\mathrm{Sign}})) + \eta_t^3 \frac{\rho_H}{6}P^{3/2}$$

$$\leq -\eta_t \|\nabla L(\boldsymbol{\theta}_t^{\mathrm{Sign}})\|_1 + \frac{\eta_t^2}{2}\sum_b \|[\nabla^2 L(\boldsymbol{\theta}_t^{\mathrm{Sign}})]_b\|_2 P_b + \frac{\eta_t^2}{2}\|\nabla^2 L(\boldsymbol{\theta}_t^{\mathrm{Sign}}) - \nabla^2 L_D(\boldsymbol{\theta}_t^{\mathrm{Sign}})\|_2 P + \eta_t^3 \frac{\rho_H}{6}P^{3/2}$$

$$\leq -\eta_t \|\nabla L(\boldsymbol{\theta}_t^{\mathrm{Sign}})\|_1 + \frac{\eta_t^2}{2}\Lambda_P P + \frac{\eta_t^2}{2}\delta_D P + \eta_t^3 \frac{\rho_H}{6}P^{3/2} \qquad (10)$$

$$\leq -\eta_t \|\nabla L(\boldsymbol{\theta}_t^{\mathrm{Sign}})\|_1 + \frac{\eta_t}{2}\|\nabla L(\boldsymbol{\theta}_t^{\mathrm{Sign}})\|_1 + \frac{\eta_t}{6}\|\nabla L(\boldsymbol{\theta}_t^{\mathrm{Sign}})\|_1 + \frac{\eta_t}{6}\|\nabla L(\boldsymbol{\theta}_t^{\mathrm{Sign}})\|_1$$

$$= -\frac{\eta_t}{6}\|\nabla L(\boldsymbol{\theta}_t^{\mathrm{Sign}})\|_1,$$

where we used $\eta_t \leq \min(\frac{\|\nabla L(\boldsymbol{\theta}_t^{\mathrm{Sign}})\|_1}{\Lambda_P P}, \sqrt{\frac{\|\nabla L(\boldsymbol{\theta}_t^{\mathrm{Sign}})\|_1}{\rho_H P^{3/2}}})$ and $\delta_D < \Lambda_P/3$.

Taking the telescoping sum, and noting that $\boldsymbol{\theta}_0 = \boldsymbol{\theta}_0^{\mathrm{Sign}}$ and $\eta_t \geq \zeta_0 \min(\frac{\|\nabla L(\boldsymbol{\theta}_t^{\mathrm{Sign}})\|_1}{\Lambda_P P}, \sqrt{\frac{\|\nabla L(\boldsymbol{\theta}_t^{\mathrm{Sign}})\|_1}{\rho_H P^{3/2}}})$, we have:

$$L(\boldsymbol{\theta}_T^{\mathrm{Sign}}) - L(\boldsymbol{\theta}_0) \leq -\frac{1}{6}\sum_{t=0}^{T-1} \eta_t \|\nabla L(\boldsymbol{\theta}_t^{\mathrm{Sign}})\|_1$$

$$\leq -\frac{\zeta_0}{6}\sum_{t=0}^{T-1} \min(\frac{\|\nabla L(\boldsymbol{\theta}_t^{\mathrm{Sign}})\|_1}{P\Lambda_P}, \sqrt{\frac{\|\nabla L(\boldsymbol{\theta}_t^{\mathrm{Sign}})\|_1}{\rho_H P^{3/2}}})\|\nabla L(\boldsymbol{\theta}_t^{\mathrm{Sign}})\|_1. \qquad (11)$$

Assume that $\|\nabla L(\boldsymbol{\theta}_t^{\mathrm{Sign}})\|_1 \geq P\varepsilon$ holds for all $0 \leq t < T$. Then, using $\varepsilon < \frac{\Lambda_P^2}{\rho_H \sqrt{P}}$, we have

$$L(\boldsymbol{\theta}_T^{\mathrm{Sign}}) - L(\boldsymbol{\theta}_0) \leq -\frac{TP\varepsilon\zeta_0}{6}\min(\frac{\varepsilon}{\Lambda_P}, \sqrt{\frac{\varepsilon}{\rho_H P^{1/2}}})$$

$$= -\frac{TP\varepsilon^2\zeta_0}{6\Lambda_P}.$$

Therefore, we have:

$$T \leq \frac{6(L(\boldsymbol{\theta}_0) - L(\boldsymbol{\theta}_T^{\mathrm{Sign}}))}{P\varepsilon^2\zeta_0}\Lambda_P$$

$$\leq \frac{6(L(\boldsymbol{\theta}_0) - L_*)}{P\varepsilon^2\zeta_0}\Lambda_P.$$

This means:

$$\mathcal{T}_\varepsilon(\{\boldsymbol{\theta}_t^{\mathrm{Sign}}\}_{t=0}^\infty, L, \|\cdot\|_1) \leq \frac{6(L(\boldsymbol{\theta}_0) - L_*)}{P\varepsilon^2\zeta_0}\Lambda_P.$$

$\square$

### A.4 Proof of Theorem 4.9

**Theorem 4.9 is restated.** *Assume $\delta_D < \min(\Lambda_G, \Lambda_P)/3$. Then, the iteration complexities in the stochastic setting are bounded as follows.*

*For the gradient-based sequence, define $\varepsilon_{\text{noise}}^2$ as in Eq. (5). Suppose that $\varepsilon_{\text{noise}}^2 < \varepsilon^2 < \frac{(1+\sigma_2)^2 \Lambda_G^2}{4(1+\sigma_3)\rho_H \sqrt{P}}$ holds and that the learning rate at time $t$ satisfies $\eta_t = \zeta_t \min(\frac{1}{(1+\sigma_2)\Lambda_G}, \frac{1}{2\sqrt{(1+\sigma_3)\rho_H \|\nabla L(\boldsymbol{\theta}_t^{Grad})\|_2}})$, where $\zeta_t \in [\zeta_0, 1]$, we have*

$$\mathcal{T}_\varepsilon(\{\boldsymbol{\theta}_t^{Grad}\}_{t=0}^\infty, L, \|\cdot\|_2) \le \frac{12(1+\sigma_2)\Lambda_G(L(\boldsymbol{\theta}_0) - L_*)}{\zeta_0 P(\varepsilon^2 - \varepsilon_{\text{noise}}^2)}.$$

*For the sign-based sequence, suppose that Assumption 4.8 holds, $\varepsilon < \frac{5\Lambda_P^2}{3(1-2q)\rho_H \sqrt{P}}$, and that the learning rate at time $t$ satisfies $\eta_t = \zeta_t \min(\frac{3(1-2q)\|\nabla L(\boldsymbol{\theta}_t^{Sign})\|_1}{5\Lambda_P P}, \sqrt{\frac{3(1-2q)\|\nabla L(\boldsymbol{\theta}_t^{Sign})\|_1}{5\rho_H P^{3/2}}})$, where $\zeta_t \in [\zeta_0, 1]$, we have*

$$\mathcal{T}_\varepsilon(\{\boldsymbol{\theta}_t^{Sign}\}_{t=0}^\infty, L, \|\cdot\|_1) \le \frac{20(L(\boldsymbol{\theta}_0) - L_*)}{3(1-2q)^2 P \varepsilon^2 \zeta_0} \Lambda_P.$$

*Proof of gradient-based sequence.* The update rule of the gradient-based sequence in stochastic setting is $\boldsymbol{\theta}_{t+1}^{\text{Grad}} = \boldsymbol{\theta}_t^{\text{Grad}} - \eta_t \nabla \widehat{L}(\boldsymbol{\theta}_t^{\text{Grad}})$. Thus, using Lemma A.1, we obtain:

$$\mathbb{E}\left[L(\boldsymbol{\theta}_{t+1}^{\text{Grad}}) - L(\boldsymbol{\theta}_t^{\text{Grad}}) \mid \boldsymbol{\theta}_t^{\text{Grad}}\right]$$

$$\le \mathbb{E}\left[\nabla L(\boldsymbol{\theta}_t^{\text{Grad}})^\top(\boldsymbol{\theta}_{t+1}^{\text{Grad}} - \boldsymbol{\theta}_t^{\text{Grad}}) + \frac{1}{2}(\boldsymbol{\theta}_{t+1}^{\text{Grad}} - \boldsymbol{\theta}_t^{\text{Grad}})^\top \nabla^2 L(\boldsymbol{\theta}_t^{\text{Grad}})(\boldsymbol{\theta}_{t+1}^{\text{Grad}} - \boldsymbol{\theta}_t^{\text{Grad}}) + \frac{\rho_H}{6}\|\boldsymbol{\theta}_{t+1}^{\text{Grad}} - \boldsymbol{\theta}_t^{\text{Grad}}\|_2^3 \mid \boldsymbol{\theta}_t^{\text{Grad}}\right]$$

$$= -\eta_t\|\nabla L(\boldsymbol{\theta}_t^{\text{Grad}})\|_2^2 + \mathbb{E}\left[\frac{\eta_t^2}{2}\nabla\widehat{L}(\boldsymbol{\theta}_t^{\text{Grad}})^\top \nabla^2 L(\boldsymbol{\theta}_t^{\text{Grad}})\nabla\widehat{L}(\boldsymbol{\theta}_t^{\text{Grad}}) + \eta_t^3\frac{\rho_H}{6}\|\nabla\widehat{L}(\boldsymbol{\theta}_t^{\text{Grad}})\|_2^3 \mid \boldsymbol{\theta}_t^{\text{Grad}}\right]$$

$$= -\eta_t\|\nabla L(\boldsymbol{\theta}_t^{\text{Grad}})\|_2^2 + \mathbb{E}\left[\frac{\eta_t^2}{2}\nabla\widehat{L}(\boldsymbol{\theta}_t^{\text{Grad}})^\top \nabla^2 L_D(\boldsymbol{\theta}_t^{\text{Grad}})\nabla\widehat{L}(\boldsymbol{\theta}_t^{\text{Grad}}) \mid \boldsymbol{\theta}_t^{\text{Grad}}\right]$$

$$+ \mathbb{E}\left[\frac{\eta_t^2}{2}\nabla\widehat{L}(\boldsymbol{\theta}_t^{\text{Grad}})^\top(\nabla^2 L(\boldsymbol{\theta}_t^{\text{Grad}}) - \nabla^2 L_D(\boldsymbol{\theta}_t^{\text{Grad}}))\nabla\widehat{L}(\boldsymbol{\theta}_t^{\text{Grad}}) + \eta_t^3\frac{\rho_H}{6}\|\nabla\widehat{L}(\boldsymbol{\theta}_t^{\text{Grad}})\|_2^3 \mid \boldsymbol{\theta}_t^{\text{Grad}}\right]$$

$$= -\eta_t\|\nabla L(\boldsymbol{\theta}_t^{\text{Grad}})\|_2^2 + \mathbb{E}\left[\frac{\eta_t^2}{2}\sum_b[\nabla\widehat{L}(\boldsymbol{\theta}_t^{\text{Grad}})]_b^\top[\nabla^2 L(\boldsymbol{\theta}_t^{\text{Grad}})]_b[\nabla\widehat{L}(\boldsymbol{\theta}_t^{\text{Grad}})]_b \mid \boldsymbol{\theta}_t^{\text{Grad}}\right]$$

$$+ \mathbb{E}\left[\frac{\eta_t^2}{2}\nabla\widehat{L}(\boldsymbol{\theta}_t^{\text{Grad}})^\top(\nabla^2 L(\boldsymbol{\theta}_t^{\text{Grad}}) - \nabla^2 L_D(\boldsymbol{\theta}_t^{\text{Grad}}))\nabla\widehat{L}(\boldsymbol{\theta}_t^{\text{Grad}}) + \eta_t^3\frac{\rho_H}{6}\|\nabla\widehat{L}(\boldsymbol{\theta}_t^{\text{Grad}})\|_2^3 \mid \boldsymbol{\theta}_t^{\text{Grad}}\right]$$

$$\le -\eta_t\|\nabla L(\boldsymbol{\theta}_t^{\text{Grad}})\|_2^2 + \mathbb{E}\left[\frac{\eta_t^2}{2}\sum_b\|[\nabla^2 L(\boldsymbol{\theta}_t^{\text{Grad}})]_b\|_2\|[\nabla\widehat{L}(\boldsymbol{\theta}_t^{\text{Grad}})]_b\|_2^2 \mid \boldsymbol{\theta}_t^{\text{Grad}}\right]$$

$$+ \mathbb{E}\left[\frac{\eta_t^2}{2}\|\nabla^2 L(\boldsymbol{\theta}_t^{\text{Grad}}) - \nabla^2 L_D(\boldsymbol{\theta}_t^{\text{Grad}})\|_2\|\nabla\widehat{L}(\boldsymbol{\theta}_t^{\text{Grad}})\|_2^2 \mid \boldsymbol{\theta}_t^{\text{Grad}}\right] + \mathbb{E}\left[\eta_t^3\frac{\rho_H}{6}\|\nabla\widehat{L}(\boldsymbol{\theta}_t^{\text{Grad}})\|_2^3 \mid \boldsymbol{\theta}_t^{\text{Grad}}\right]. \quad (12)$$

For the second and third term, using Eqs.(2)(4), and $\delta_D < \Lambda_G/3$, we can derive an upper bound as follows:

$$\mathbb{E}\left[\frac{\eta_t^2}{2}\sum_b\|[\nabla^2 L(\boldsymbol{\theta}_t^{\text{Grad}})]_b\|_2\|[\nabla\widehat{L}(\boldsymbol{\theta}_t^{\text{Grad}})]_b\|_2^2 \mid \boldsymbol{\theta}_t^{\text{Grad}}\right] + \mathbb{E}\left[\frac{\eta_t^2}{2}\|\nabla^2 L(\boldsymbol{\theta}_t^{\text{Grad}}) - \nabla^2 L_D(\boldsymbol{\theta}_t^{\text{Grad}})\|_2\|\nabla\widehat{L}(\boldsymbol{\theta}_t^{\text{Grad}})\|_2^2 \mid \boldsymbol{\theta}_t^{\text{Grad}}\right]$$

$$\le \mathbb{E}\left[\frac{\eta_t^2}{2}\sum_b\|[\nabla^2 L(\boldsymbol{\theta}_t^{\text{Grad}})]_b\|_2\|[\nabla\widehat{L}(\boldsymbol{\theta}_t^{\text{Grad}})]_b\|_2^2 \mid \boldsymbol{\theta}_t^{\text{Grad}}\right] + \mathbb{E}\left[\frac{\eta_t^2}{2}\delta_D\|\nabla\widehat{L}(\boldsymbol{\theta}_t^{\text{Grad}})\|_2^2 \mid \boldsymbol{\theta}_t^{\text{Grad}}\right]$$

$$= \frac{\eta_t^2}{2}\sum_b\|[\nabla^2 L(\boldsymbol{\theta}_t^{\text{Grad}})]_b\|_2\sum_i\mathbb{E}\left[((([\nabla L(\boldsymbol{\theta}_t^{\text{Grad}})]_b)_i + ([\nabla\widehat{L}(\boldsymbol{\theta}_t^{\text{Grad}})]_b)_i - ([\nabla L(\boldsymbol{\theta}_t^{\text{Grad}})]_b)_i)^2 \mid \boldsymbol{\theta}_t^{\text{Grad}}\right]$$

$$+ \frac{\eta_t^2}{2}\delta_D \sum_i \mathbb{E}\left[(\nabla L(\boldsymbol{\theta}_t^{\mathrm{Grad}})_i + \nabla\widehat{L}(\boldsymbol{\theta}_t^{\mathrm{Grad}})_i - \nabla L(\boldsymbol{\theta}_t^{\mathrm{Grad}})_i)^2 \mid \boldsymbol{\theta}_t^{\mathrm{Grad}}\right]$$

$$\leq \frac{\eta_t^2}{2}\sum_b \|[\nabla^2 L(\boldsymbol{\theta}_t^{\mathrm{Grad}})]_b\|_2 \left((1+\sigma_2)\|[\nabla L(\boldsymbol{\theta}_t^{\mathrm{Grad}})]_b\|_2^2 + P_b\tau_2\right) + \frac{\eta_t^2}{2}\delta_D\left((1+\sigma_2)\|\nabla L(\boldsymbol{\theta}_t^{\mathrm{Grad}})\|_2^2 + P\tau_2\right)$$

$$\leq \frac{\eta_t^2}{2}(1+\sigma_2)\Lambda_G\|\nabla L(\boldsymbol{\theta}_t^{\mathrm{Grad}})\|_2^2 + \frac{\eta_t^2}{2}(1+\sigma_2)\delta_D\|\nabla L(\boldsymbol{\theta}_t^{\mathrm{Grad}})\|_2^2 + \frac{\eta_t^2}{2}(\Lambda_G + \delta_D)P\tau_2$$

$$\leq \frac{2\eta_t^2}{3}(1+\sigma_2)\Lambda_G\|\nabla L(\boldsymbol{\theta}_t^{\mathrm{Grad}})\|_2^2 + \frac{2\eta_t^2}{3}\Lambda_G P\tau_2 \quad . \tag{13}$$

For the fourth term, using Lemma A.2 and Eq.(3), we can derive an upper bound as follows:

$$\mathbb{E}\left[\eta_t^3 \frac{\rho_H}{6}\|\nabla\widehat{L}(\boldsymbol{\theta}_t^{\mathrm{Grad}})\|_2^3 \mid \boldsymbol{\theta}_t^{\mathrm{Grad}}\right]$$

$$\leq \eta_t^3 \frac{\rho_H}{6}\mathbb{E}\left[(\|\nabla L(\boldsymbol{\theta}_t^{\mathrm{Grad}})\|_2 + \|\nabla\widehat{L}(\boldsymbol{\theta}_t^{\mathrm{Grad}}) - \nabla L(\boldsymbol{\theta}_t^{\mathrm{Grad}})\|_2)^3 \mid \boldsymbol{\theta}_t^{\mathrm{Grad}}\right]$$

$$\leq \frac{2\eta_t^3\rho_H}{3}\mathbb{E}\left[\|\nabla L(\boldsymbol{\theta}_t^{\mathrm{Grad}})\|_2^3 + \|\nabla\widehat{L}(\boldsymbol{\theta}_t^{\mathrm{Grad}}) - \nabla L(\boldsymbol{\theta}_t^{\mathrm{Grad}})\|_2^3 \mid \boldsymbol{\theta}_t^{\mathrm{Grad}}\right]$$

$$\leq \frac{2\eta_t^3\rho_H}{3}\left((1+\sigma_3)\|\nabla L(\boldsymbol{\theta}_t^{\mathrm{Grad}})\|_2^3 + \tau_3\right). \tag{14}$$

Combining Eqs.(12)(13)(14) and $\eta_t \leq \min(\frac{1}{(1+\sigma_2)\Lambda_G}, \frac{1}{2\sqrt{(1+\sigma_3)\rho_H\|\nabla L(\boldsymbol{\theta}_t^{\mathrm{Grad}})\|_2}})$, we have:

$$\mathbb{E}\left[L(\boldsymbol{\theta}_{t+1}^{\mathrm{Grad}}) - L(\boldsymbol{\theta}_t^{\mathrm{Grad}}) \mid \boldsymbol{\theta}_t^{\mathrm{Grad}}\right]$$

$$\leq -\eta_t\|\nabla L(\boldsymbol{\theta}_t^{\mathrm{Grad}})\|_2^2 + \frac{2\eta_t^2}{3}(1+\sigma_2)\Lambda_G\|\nabla L(\boldsymbol{\theta}_t^{\mathrm{Grad}})\|_2^2 + \frac{2\eta_t^3\rho_H}{3}(1+\sigma_3)\|\nabla L(\boldsymbol{\theta}_t^{\mathrm{Grad}})\|_2^3$$

$$+ \frac{2\eta_t^2}{3}\Lambda_G P\tau_2 + \frac{2\eta_t^3\rho_H}{3}\tau_3$$

$$\leq -\frac{\eta_t}{6}\|\nabla L(\boldsymbol{\theta}_t^{\mathrm{Grad}})\|_2^2 + \frac{2\eta_t^2}{3}\Lambda_G P\tau_2 + \frac{2\eta_t^3\rho_H}{3}\tau_3. \tag{15}$$

Since $\eta_t \leq \frac{1}{(1+\sigma_2)\Lambda_G}$, the additive noise terms satisfy

$$\frac{2\eta_t^2}{3}\Lambda_G P\tau_2 + \frac{2\eta_t^3\rho_H}{3}\tau_3 \leq \frac{2P\tau_2}{3(1+\sigma_2)^2\Lambda_G} + \frac{2\rho_H\tau_3}{3(1+\sigma_2)^3\Lambda_G^3} \leq \frac{\zeta_0 P}{12(1+\sigma_2)\Lambda_G}\varepsilon_{\mathrm{noise}}^2,$$

where the last inequality uses the definition of $\varepsilon_{\mathrm{noise}}^2$ in Eq. (5). Assume that the probability of the event $\mathcal{E}(T) = \{\forall s \leq T, \|\nabla L(\boldsymbol{\theta}_s^{\mathrm{Grad}})\|_2 \geq \sqrt{P}\varepsilon\}$ satisfies $\mathbb{P}(\mathcal{E}(T)) \geq \frac{1}{2}$. Applying the telescoping sum to Eq. (15) and taking expectations, and noting that $\boldsymbol{\theta}_0 = \boldsymbol{\theta}_0^{\mathrm{Grad}}$, $\eta_t \geq \zeta_0 \min(\frac{1}{(1+\sigma_2)\Lambda_G}, \frac{1}{2\sqrt{(1+\sigma_3)\rho_H\|\nabla L(\boldsymbol{\theta}_t^{\mathrm{Grad}})\|_2}})$, and $\varepsilon < \frac{(1+\sigma_2)^2\Lambda_G^2}{4(1+\sigma_3)\rho_H\sqrt{P}}$, we have:

$$\mathbb{E}\left[L(\boldsymbol{\theta}_T^{\mathrm{Grad}})\right] - L(\boldsymbol{\theta}_0)$$

$$\leq -\frac{1}{6}\sum_{t=0}^{T-1}\mathbb{E}\left[\eta_t\|\nabla L(\boldsymbol{\theta}_t^{\mathrm{Grad}})\|_2^2\right] + \frac{T\zeta_0 P}{12(1+\sigma_2)\Lambda_G}\varepsilon_{\mathrm{noise}}^2$$

$$= -\frac{1}{6}\sum_{t=0}^{T-1}\left(\mathbb{E}\left[\eta_t\|\nabla L(\boldsymbol{\theta}_t^{\mathrm{Grad}})\|_2^2 \mid \mathcal{E}(T)\right]\mathbb{P}(\mathcal{E}(T)) + \mathbb{E}\left[\eta_t\|\nabla L(\boldsymbol{\theta}_t^{\mathrm{Grad}})\|_2^2 \mid \overline{\mathcal{E}(T)}\right]\mathbb{P}\left(\overline{\mathcal{E}(T)}\right)\right) + \frac{T\zeta_0 P}{12(1+\sigma_2)\Lambda_G}\varepsilon_{\mathrm{noise}}^2$$

$$\leq -\frac{1}{6}\sum_{t=0}^{T-1}\mathbb{E}\left[\eta_t\|\nabla L(\boldsymbol{\theta}_t^{\mathrm{Grad}})\|_2^2 \mid \mathcal{E}(T)\right]\mathbb{P}(\mathcal{E}(T)) + \frac{T\zeta_0 P}{12(1+\sigma_2)\Lambda_G}\varepsilon_{\mathrm{noise}}^2$$

$$\leq -\frac{1}{12}\sum_{t=0}^{T-1}\mathbb{E}\left[\eta_t\|\nabla L(\boldsymbol{\theta}_t^{\mathrm{Grad}})\|_2^2 \mid \mathcal{E}(T)\right] + \frac{T\zeta_0 P}{12(1+\sigma_2)\Lambda_G}\varepsilon_{\mathrm{noise}}^2$$

$$
\begin{aligned}
&\leq -\frac{\zeta_0}{12} \sum_{t=0}^{T-1} \mathbb{E}\left[\min\left(\frac{\|\nabla L(\boldsymbol{\theta}_t^{\mathrm{Grad}})\|_2^2}{(1+\sigma_2)\Lambda_G}, \frac{\|\nabla L(\boldsymbol{\theta}_t^{\mathrm{Grad}})\|_2^{3/2}}{2\sqrt{(1+\sigma_3)\rho_H}}\right) \mid \mathcal{E}(T)\right] + \frac{T\zeta_0 P}{12(1+\sigma_2)\Lambda_G}\varepsilon_{\mathrm{noise}}^2 \\
&\leq -\frac{T\zeta_0}{12}\min\left(\frac{P\varepsilon^2}{(1+\sigma_2)\Lambda_G}, \frac{P^{3/4}\varepsilon^{3/2}}{2\sqrt{(1+\sigma_3)\rho_H}}\right) + \frac{T\zeta_0 P}{12(1+\sigma_2)\Lambda_G}\varepsilon_{\mathrm{noise}}^2 \\
&= -\frac{TP\zeta_0(\varepsilon^2 - \varepsilon_{\mathrm{noise}}^2)}{12(1+\sigma_2)\Lambda_G}.
\end{aligned}
$$

Therefore, we have

$$
\begin{aligned}
T &\leq \frac{12(1+\sigma_2)\Lambda_G(L(\boldsymbol{\theta}_0) - \mathbb{E}\left[L(\boldsymbol{\theta}_T^{\mathrm{Grad}})\right])}{\zeta_0 P(\varepsilon^2 - \varepsilon_{\mathrm{noise}}^2)} \\
&\leq \frac{12(1+\sigma_2)\Lambda_G(L(\boldsymbol{\theta}_0) - L_*)}{\zeta_0 P(\varepsilon^2 - \varepsilon_{\mathrm{noise}}^2)}.
\end{aligned}
$$

This means that when we take $T > \frac{12(1+\sigma_2)\Lambda_G(L(\boldsymbol{\theta}_0) - L_*)}{\zeta_0 P(\varepsilon^2 - \varepsilon_{\mathrm{noise}}^2)}$, we have $\mathbb{P}\left(\mathcal{E}(T)\right) < \frac{1}{2}$. Therefore, we have

$$
\mathcal{T}_\varepsilon(\{\boldsymbol{\theta}_t^{\mathrm{Grad}}\}_{t=0}^\infty, L, \|\cdot\|_2) \leq \frac{12(1+\sigma_2)\Lambda_G(L(\boldsymbol{\theta}_0) - L_*)}{\zeta_0 P(\varepsilon^2 - \varepsilon_{\mathrm{noise}}^2)}.
$$

$\square$

*Proof of sign-based sequence.* The update rule of the sign-based sequence in stochastic setting is $\boldsymbol{\theta}_{t+1}^{\mathrm{Sign}} = \boldsymbol{\theta}_t^{\mathrm{Sign}} - \eta_t \operatorname{sign}(\nabla\widehat{L}(\boldsymbol{\theta}_t^{\mathrm{Sign}}))$. Thus, using Lemma A.1, we obtain:

$$
\begin{aligned}
&\mathbb{E}\left[L(\boldsymbol{\theta}_{t+1}^{\mathrm{Sign}}) - L(\boldsymbol{\theta}_t^{\mathrm{Sign}}) \mid \boldsymbol{\theta}_t^{\mathrm{Sign}}\right] \\
&\leq \mathbb{E}\left[\nabla L(\boldsymbol{\theta}_t^{\mathrm{Sign}})^\top(\boldsymbol{\theta}_{t+1}^{\mathrm{Sign}} - \boldsymbol{\theta}_t^{\mathrm{Sign}}) + \frac{1}{2}(\boldsymbol{\theta}_{t+1}^{\mathrm{Sign}} - \boldsymbol{\theta}_t^{\mathrm{Sign}})^\top \nabla^2 L(\boldsymbol{\theta}_t^{\mathrm{Sign}})(\boldsymbol{\theta}_{t+1}^{\mathrm{Sign}} - \boldsymbol{\theta}_t^{\mathrm{Sign}}) + \frac{\rho_H}{6}\|\boldsymbol{\theta}_{t+1}^{\mathrm{Sign}} - \boldsymbol{\theta}_t^{\mathrm{Sign}}\|_2^3 \mid \boldsymbol{\theta}_t^{\mathrm{Sign}}\right] \\
&= -\eta_t\|\nabla L(\boldsymbol{\theta}_t^{\mathrm{Sign}})\|_1 + \mathbb{E}\left[\frac{\eta_t^2}{2}\operatorname{sign}(\nabla\widehat{L}(\boldsymbol{\theta}_t^{\mathrm{Sign}}))^\top \nabla^2 L(\boldsymbol{\theta}_t^{\mathrm{Sign}})\operatorname{sign}(\nabla\widehat{L}(\boldsymbol{\theta}_t^{\mathrm{Sign}})) + \eta_t^3\frac{\rho_H}{6}\|\operatorname{sign}(\nabla\widehat{L}(\boldsymbol{\theta}_t^{\mathrm{Sign}}))\|_2^3 \mid \boldsymbol{\theta}_t^{\mathrm{Sign}}\right] \\
&\quad + \mathbb{E}\left[-\eta_t\nabla L(\boldsymbol{\theta}_t^{\mathrm{Sign}})^\top(\operatorname{sign}(\nabla\widehat{L}(\boldsymbol{\theta}_t^{\mathrm{Sign}})) - \operatorname{sign}(\nabla L(\boldsymbol{\theta}_t^{\mathrm{Sign}}))) \mid \boldsymbol{\theta}_t^{\mathrm{Sign}}\right]. \quad (16)
\end{aligned}
$$

For the second term, we can derive an upper bound in the same way as in the deterministic case:

$$
\begin{aligned}
&\mathbb{E}\left[\frac{\eta_t^2}{2}\operatorname{sign}(\nabla\widehat{L}(\boldsymbol{\theta}_t^{\mathrm{Sign}}))^\top \nabla^2 L(\boldsymbol{\theta}_t^{\mathrm{Sign}})\operatorname{sign}(\nabla\widehat{L}(\boldsymbol{\theta}_t^{\mathrm{Sign}})) + \eta_t^3\frac{\rho_H}{6}\|\operatorname{sign}(\nabla\widehat{L}(\boldsymbol{\theta}_t^{\mathrm{Sign}}))\|_2^3 \mid \boldsymbol{\theta}_t^{\mathrm{Sign}}\right] \\
&= \mathbb{E}\left[\frac{\eta_t^2}{2}\operatorname{sign}(\nabla\widehat{L}(\boldsymbol{\theta}_t^{\mathrm{Sign}}))^\top \nabla^2 L_D(\boldsymbol{\theta}_t^{\mathrm{Sign}})\operatorname{sign}(\nabla\widehat{L}(\boldsymbol{\theta}_t^{\mathrm{Sign}})) \mid \boldsymbol{\theta}_t^{\mathrm{Sign}}\right] \\
&\quad + \mathbb{E}\left[\frac{\eta_t^2}{2}\operatorname{sign}(\nabla\widehat{L}(\boldsymbol{\theta}_t^{\mathrm{Sign}}))^\top(\nabla^2 L(\boldsymbol{\theta}_t^{\mathrm{Sign}}) - \nabla^2 L_D(\boldsymbol{\theta}_t^{\mathrm{Sign}}))\operatorname{sign}(\nabla\widehat{L}(\boldsymbol{\theta}_t^{\mathrm{Sign}})) \mid \boldsymbol{\theta}_t^{\mathrm{Sign}}\right] + \eta_t^3\frac{\rho_H}{6}P^{3/2} \\
&= \mathbb{E}\left[\frac{\eta_t^2}{2}\sum_b [\operatorname{sign}(\nabla\widehat{L}(\boldsymbol{\theta}_t^{\mathrm{Sign}}))]_b^\top[\nabla^2 L(\boldsymbol{\theta}_t^{\mathrm{Sign}})]_b[\operatorname{sign}(\nabla\widehat{L}(\boldsymbol{\theta}_t^{\mathrm{Sign}}))]_b \mid \boldsymbol{\theta}_t^{\mathrm{Sign}}\right] \\
&\quad + \mathbb{E}\left[\frac{\eta_t^2}{2}\operatorname{sign}(\nabla\widehat{L}(\boldsymbol{\theta}_t^{\mathrm{Sign}}))^\top(\nabla^2 L(\boldsymbol{\theta}_t^{\mathrm{Sign}}) - \nabla^2 L_D(\boldsymbol{\theta}_t^{\mathrm{Sign}}))\operatorname{sign}(\nabla\widehat{L}(\boldsymbol{\theta}_t^{\mathrm{Sign}})) \mid \boldsymbol{\theta}_t^{\mathrm{Sign}}\right] + \eta_t^3\frac{\rho_H}{6}P^{3/2} \\
&\leq \frac{\eta_t^2}{2}\sum_b \|[\nabla^2 L(\boldsymbol{\theta}_t^{\mathrm{Sign}})]_b\|_2 P_b + \frac{\eta_t^2}{2}\|\nabla^2 L(\boldsymbol{\theta}_t^{\mathrm{Sign}}) - \nabla^2 L_D(\boldsymbol{\theta}_t^{\mathrm{Sign}})\|_2 P + \eta_t^3\frac{\rho_H}{6}P^{3/2} \\
&\leq \frac{\eta_t^2}{2}\Lambda_P P + \frac{\eta_t^2}{2}\delta_D P + \eta_t^3\frac{\rho_H}{6}P^{3/2}. \quad (17)
\end{aligned}
$$

For the third term, we can derive an upper bound using the sign-reliability condition:

$$
\mathbb{E}\left[-\eta_t \nabla L(\boldsymbol{\theta}_t^{\mathrm{Sign}})^\top (\mathrm{sign}(\nabla \widehat{L}(\boldsymbol{\theta}_t^{\mathrm{Sign}})) - \mathrm{sign}(\nabla L(\boldsymbol{\theta}_t^{\mathrm{Sign}}))) \mid \boldsymbol{\theta}_t^{\mathrm{Sign}}\right]
$$

$$
= \eta_t \sum_{i=1}^{P} \nabla L(\boldsymbol{\theta}_t^{\mathrm{Sign}})_i \mathbb{E}\left[\mathrm{sign}(\nabla L(\boldsymbol{\theta}_t^{\mathrm{Sign}}))_i - \mathrm{sign}(\nabla \widehat{L}(\boldsymbol{\theta}_t^{\mathrm{Sign}}))_i \mid \boldsymbol{\theta}_t^{\mathrm{Sign}}\right]
$$

$$
= \eta_t \sum_{i=1}^{P} |\nabla L(\boldsymbol{\theta}_t^{\mathrm{Sign}})_i| 2\mathbb{E}\left[\mathbb{1}[\mathrm{sign}(\nabla L(\boldsymbol{\theta}_t^{\mathrm{Sign}}))_i \neq \mathrm{sign}(\nabla \widehat{L}(\boldsymbol{\theta}_t^{\mathrm{Sign}}))_i] \mid \boldsymbol{\theta}_t^{\mathrm{Sign}}\right]
$$

$$
= \eta_t \sum_{i=1}^{P} |\nabla L(\boldsymbol{\theta}_t^{\mathrm{Sign}})_i| 2\mathbb{P}\left(\mathrm{sign}(\nabla L(\boldsymbol{\theta}_t^{\mathrm{Sign}}))_i \neq \mathrm{sign}(\nabla \widehat{L}(\boldsymbol{\theta}_t^{\mathrm{Sign}}))_i \mid \boldsymbol{\theta}_t^{\mathrm{Sign}}\right)
$$

$$
\leq 2q\eta_t \|\nabla L(\boldsymbol{\theta}_t^{\mathrm{Sign}})\|_1, \tag{18}
$$

where the last inequality uses Assumption 4.8.

Let $\alpha > \frac{5}{6(1-2q)}$ and suppose that $\eta_t \leq \min(\frac{\|\nabla L(\boldsymbol{\theta}_t^{\mathrm{Sign}})\|_1}{\alpha \Lambda_P P}, \sqrt{\frac{\|\nabla L(\boldsymbol{\theta}_t^{\mathrm{Sign}})\|_1}{\alpha \rho_H P^{3/2}}})$. Combining Eqs.(16)(17)(18) and $\delta_D < \Lambda_P/3$, we have:

$$
\mathbb{E}\left[L(\boldsymbol{\theta}_{t+1}^{\mathrm{Sign}}) - L(\boldsymbol{\theta}_t^{\mathrm{Sign}}) \mid \boldsymbol{\theta}_t^{\mathrm{Sign}}\right]
$$

$$
\leq -\eta_t \|\nabla L(\boldsymbol{\theta}_t^{\mathrm{Sign}})\|_1 + \frac{\eta_t^2}{2}\Lambda_P P + \frac{\eta_t^2}{2}\delta_D P + \eta_t^3 \frac{\rho_H}{6} P^{3/2} + 2q\eta_t \|\nabla L(\boldsymbol{\theta}_t^{\mathrm{Sign}})\|_1
$$

$$
\leq -\eta_t \|\nabla L(\boldsymbol{\theta}_t^{\mathrm{Sign}})\|_1 + \frac{\eta_t}{2\alpha}\|\nabla L(\boldsymbol{\theta}_t^{\mathrm{Sign}})\|_1 + \frac{\eta_t}{6\alpha}\|\nabla L(\boldsymbol{\theta}_t^{\mathrm{Sign}})\|_1 + \frac{\eta_t}{6\alpha}\|\nabla L(\boldsymbol{\theta}_t^{\mathrm{Sign}})\|_1 + 2q\eta_t \|\nabla L(\boldsymbol{\theta}_t^{\mathrm{Sign}})\|_1
$$

$$
= -\frac{(6\alpha(1-2q) - 5)\eta_t}{6\alpha}\|\nabla L(\boldsymbol{\theta}_t^{\mathrm{Sign}})\|_1.
$$

Assume that the probability of the event $\mathcal{E}(T) = \{\forall s \leq T, \|\nabla L(\boldsymbol{\theta}_s^{\mathrm{Sign}})\|_1 \geq P\varepsilon\}$ satisfies $\mathbb{P}(\mathcal{E}(T)) \geq \frac{1}{2}$. By applying the telescoping sum and taking expectations, and noting that $\boldsymbol{\theta}_0 = \boldsymbol{\theta}_0^{\mathrm{Sign}}$, $\eta_t \geq \zeta_0 \min(\frac{\|\nabla L(\boldsymbol{\theta}_t^{\mathrm{Sign}})\|_1}{\alpha \Lambda_P P}, \sqrt{\frac{\|\nabla L(\boldsymbol{\theta}_t^{\mathrm{Sign}})\|_1}{\alpha \rho_H P^{3/2}}})$, and $\varepsilon < \frac{\alpha \Lambda_P^2}{\rho_H \sqrt{P}}$ we have:

$$
\mathbb{E}\left[L(\boldsymbol{\theta}_T^{\mathrm{Sign}})\right] - L(\boldsymbol{\theta}_0)
$$

$$
\leq -\frac{6\alpha(1-2q) - 5}{6\alpha}\sum_{t=0}^{T-1}\mathbb{E}\left[\eta_t \|\nabla L(\boldsymbol{\theta}_t^{\mathrm{Sign}})\|_1\right]
$$

$$
\leq -\frac{6\alpha(1-2q) - 5}{12\alpha}\sum_{t=0}^{T-1}\mathbb{E}\left[\eta_t \|\nabla L(\boldsymbol{\theta}_t^{\mathrm{Sign}})\|_1 \mid \mathcal{E}(T)\right]
$$

$$
\leq -\frac{(6\alpha(1-2q) - 5)\zeta_0}{12\alpha}\sum_{t=0}^{T-1}\mathbb{E}\left[\min(\frac{\|\nabla L(\boldsymbol{\theta}_t^{\mathrm{Sign}})\|_1^2}{\alpha \Lambda_P P}, \frac{\|\nabla L(\boldsymbol{\theta}_t^{\mathrm{Sign}})\|_1^{3/2}}{\sqrt{\alpha \rho_H P^{3/2}}}) \mid \mathcal{E}(T)\right]
$$

$$
\leq -\frac{(6\alpha(1-2q) - 5)\zeta_0}{12\alpha}\sum_{t=0}^{T-1}\min(\frac{P\varepsilon^2}{\alpha \Lambda_P}, P\varepsilon\sqrt{\frac{\varepsilon}{\alpha \rho_H P^{1/2}}})
$$

$$
= -\frac{(6\alpha(1-2q) - 5)TP\varepsilon^2\zeta_0}{12\alpha^2 \Lambda_P}.
$$

Therefore, we have:

$$
T \leq \frac{12\alpha^2 (L(\boldsymbol{\theta}_0) - \mathbb{E}\left[L(\boldsymbol{\theta}_T^{\mathrm{Sign}})\right])}{(6\alpha(1-2q) - 5)P\varepsilon^2\zeta_0}\Lambda_P
$$

$$\leq \frac{12\alpha^2(L(\boldsymbol{\theta}_0) - L_*)}{(6\alpha(1 - 2q) - 5)P\varepsilon^2\zeta_0}\Lambda_P.$$

This means that when we take $T > \frac{12\alpha^2(L(\boldsymbol{\theta}_0) - L_*)}{(6\alpha(1-2q)-5)P\varepsilon^2\zeta_0}\Lambda_P$, we have $\mathbb{P}\left(\mathcal{E}(T)\right) < \frac{1}{2}$. Therefore, we have

$$\mathcal{T}_\varepsilon(\{\boldsymbol{\theta}_t^{\mathrm{Sign}}\}_{t=0}^\infty, L, \|\cdot\|_1) \leq \frac{12\alpha^2(L(\boldsymbol{\theta}_0) - L_*)}{(6\alpha(1 - 2q) - 5)P\varepsilon^2\zeta_0}\Lambda_P.$$

Setting $\alpha = \frac{5}{3(1-2q)}$ completes the proof. $\qquad\square$

When Eq. (4) holds with $\tau_2 = 0$, Chebyshev's inequality implies that Assumption 4.8 is satisfied with $q = \sigma_2$. The following corollary gives an explicit sign-based complexity bound in this relative-noise case, a specialization of Theorem 4.9.

**Corollary A.3** (Relative coordinate-wise variance implies sign reliability)**.** *Under the same structural assumptions as Theorem 4.9, suppose that Eq. (4) holds with $\tau_2 = 0$ and $\sigma_2 \leq 1/24$. For the sign-based sequence, if $\varepsilon < \frac{\Lambda_P^2}{\rho_H\sqrt{P}}$ and the learning rate satisfies $\eta_t = \zeta_t \min(\frac{\|\nabla L(\boldsymbol{\theta}_t^{Sign})\|_1}{\Lambda_P P}, \sqrt{\frac{\|\nabla L(\boldsymbol{\theta}_t^{Sign})\|_1}{\rho_H P^{3/2}}})$, where $\zeta_t \in [\zeta_0, 1]$, then*

$$\mathcal{T}_\varepsilon(\{\boldsymbol{\theta}_t^{Sign}\}_{t=0}^\infty, L, \|\cdot\|_1) \leq \frac{12(1 + 24\sigma_2)(L(\boldsymbol{\theta}_0) - L_*)}{P\varepsilon^2\zeta_0}\Lambda_P.$$

*Proof.* By Chebyshev's inequality and Eq. (4) with $\tau_2 = 0$,

$$\mathbb{P}\left(\mathrm{sign}(\nabla L(\boldsymbol{\theta})_i) \neq \mathrm{sign}(\nabla\widehat{L}(\boldsymbol{\theta})_i) \mid \boldsymbol{\theta}\right) \leq \frac{\mathbb{E}[|\nabla\widehat{L}(\boldsymbol{\theta})_i - \nabla L(\boldsymbol{\theta})_i|^2]}{|\nabla L(\boldsymbol{\theta})_i|^2} \leq \sigma_2.$$

Therefore, Assumption 4.8 holds with $q = \sigma_2$. Taking $\alpha = 1$ in the proof of Theorem 4.9 gives the descent coefficient $1 - 12\sigma_2$, which is positive when $\sigma_2 < 1/12$. When $\sigma_2 \leq 1/24$, we have $(1 - 12\sigma_2)^{-1} \leq 1 + 24\sigma_2$, which yields the stated bound. $\qquad\square$

# B  Derivation of Jacobian matrix in Section 4.7

## B.1  Jacobian of Transformer layer

The output of a Transformer layer for an input $\boldsymbol{X} \in \mathbb{R}^{n \times d}$ is given by $\mathcal{M}(\mathcal{A}(\boldsymbol{X}))$, where $\mathcal{A}(\cdot)$ is the attention layer and $\mathcal{M}(\cdot)$ is the feed-forward layer. In the following, we denote the Jacobian of the self-attention module, the feed-forward module, and the layer normalization as $\boldsymbol{J}_{\mathrm{ATT}}$, $\boldsymbol{J}_{\mathrm{FFN}}$, and $\boldsymbol{J}_{\mathrm{LN}}$, respectively.

**In Pre-LN.** The self-attention and feed-forward layers in the Pre-LN architecture are given by

$$\mathcal{A}(\boldsymbol{X}) = \mathrm{ATT}(\mathrm{LN}(\boldsymbol{X})) + \boldsymbol{X},$$
$$\mathcal{M}(\boldsymbol{Y}) = \mathrm{FFN}(\mathrm{LN}(\boldsymbol{Y})) + \boldsymbol{Y}.$$

The Jacobian of these modules are as follows:

$$\frac{\partial\mathcal{A}(\boldsymbol{X})}{\partial\boldsymbol{X}} = \left.\frac{\partial\,\mathrm{ATT}(\boldsymbol{Z})}{\partial\boldsymbol{Z}}\right|_{\boldsymbol{Z}=\mathrm{LN}(\boldsymbol{X})}\frac{\partial\,\mathrm{LN}(\boldsymbol{X})}{\partial\boldsymbol{X}} + \frac{\partial\boldsymbol{X}}{\partial\boldsymbol{X}}$$
$$= \boldsymbol{J}_{\mathrm{ATT}}(\mathrm{LN}(\boldsymbol{X}))\boldsymbol{J}_{\mathrm{LN}}(\boldsymbol{X}) + \boldsymbol{I}_{nd},$$
$$\frac{\partial\mathcal{M}(\boldsymbol{Y})}{\partial\boldsymbol{Y}} = \left.\frac{\partial\,\mathrm{FFN}(\boldsymbol{Y})}{\partial\boldsymbol{Y}}\right|_{\boldsymbol{Y}=\mathrm{LN}(\boldsymbol{Y})}\frac{\partial\,\mathrm{LN}(\boldsymbol{Y})}{\partial\boldsymbol{Y}} + \frac{\partial\boldsymbol{Y}}{\partial\boldsymbol{Y}}$$
$$= \boldsymbol{J}_{\mathrm{FFN}}(\mathrm{LN}(\boldsymbol{Y}))\boldsymbol{J}_{\mathrm{LN}}(\boldsymbol{Y}) + \boldsymbol{I}_{nd}.$$

Therefore, the Jacobian of the Pre-LN layer is given by

$$
\begin{aligned}
\boldsymbol{J}_{\text{Pre-LN}}(\boldsymbol{X}) &= \left.\frac{\partial \mathcal{M}(\boldsymbol{Y})}{\partial \boldsymbol{Y}}\right|_{\boldsymbol{Y}=\mathcal{A}(\boldsymbol{X})} \frac{\partial \mathcal{A}(\boldsymbol{X})}{\partial \boldsymbol{X}} \\
&= \left(\boldsymbol{J}_{\text{FFN}}(\text{LN}(\mathcal{A}(\boldsymbol{X})))\boldsymbol{J}_{\text{LN}}(\mathcal{A}(\boldsymbol{X})) + \boldsymbol{I}_{nd}\right)\left(\boldsymbol{J}_{\text{ATT}}(\text{LN}(\boldsymbol{X}))\boldsymbol{J}_{\text{LN}}(\boldsymbol{X}) + \boldsymbol{I}_{nd}\right)
\end{aligned}
$$

and with omitting the evaluation point, we can write the Jacobian as

$$
\boldsymbol{J}_{\text{Pre-LN}} = \left(\boldsymbol{J}_{\text{FFN}}\boldsymbol{J}_{\text{LN}} + \boldsymbol{I}_{nd}\right)\left(\boldsymbol{J}_{\text{ATT}}\boldsymbol{J}_{\text{LN}} + \boldsymbol{I}_{nd}\right).
$$

**In Post-LN.** The self-attention and feed-forward layers in the Post-LN architecture are given by

$$
\begin{aligned}
\mathcal{A}(\boldsymbol{X}) &= \text{LN}(\text{ATT}(\boldsymbol{X}) + \boldsymbol{X}), \\
\mathcal{M}(\boldsymbol{Y}) &= \text{LN}(\text{FFN}(\boldsymbol{Y}) + \boldsymbol{Y}).
\end{aligned}
$$

The Jacobian of these modules are as follows:

$$
\begin{aligned}
\frac{\partial \mathcal{A}(\boldsymbol{X})}{\partial \boldsymbol{X}} &= \left.\frac{\partial \text{LN}(\boldsymbol{Z})}{\partial \boldsymbol{Z}}\right|_{\boldsymbol{Z}=\text{ATT}(\boldsymbol{X})+\boldsymbol{X}} \left(\frac{\partial \text{ATT}(\boldsymbol{X})}{\partial \boldsymbol{X}} + \frac{\partial \boldsymbol{X}}{\partial \boldsymbol{X}}\right) \\
&= \boldsymbol{J}_{\text{LN}}(\text{ATT}(\boldsymbol{X}) + \boldsymbol{X})\left(\boldsymbol{J}_{\text{ATT}}(\boldsymbol{X}) + \boldsymbol{I}_{nd}\right), \\
\frac{\partial \mathcal{M}(\boldsymbol{Y})}{\partial \boldsymbol{Y}} &= \left.\frac{\partial \text{LN}(\boldsymbol{Z})}{\partial \boldsymbol{Z}}\right|_{\boldsymbol{Z}=\text{FFN}(\boldsymbol{Y})+\boldsymbol{Y}} \left(\frac{\partial \text{FFN}(\boldsymbol{Y})}{\partial \boldsymbol{Y}} + \frac{\partial \boldsymbol{Y}}{\partial \boldsymbol{Y}}\right) \\
&= \boldsymbol{J}_{\text{LN}}(\text{FFN}(\boldsymbol{Y}) + \boldsymbol{Y})\left(\boldsymbol{J}_{\text{FFN}}(\boldsymbol{Y}) + \boldsymbol{I}_{nd}\right).
\end{aligned}
$$

Therefore, the Jacobian of the Post-LN architecture is given by

$$
\begin{aligned}
\boldsymbol{J}_{\text{Post-LN}}(\boldsymbol{X}) &= \left.\frac{\partial \mathcal{M}(\boldsymbol{Y})}{\partial \boldsymbol{Y}}\right|_{\boldsymbol{Y}=\mathcal{A}(\boldsymbol{X})} \frac{\partial \mathcal{A}(\boldsymbol{X})}{\partial \boldsymbol{X}} \\
&= \boldsymbol{J}_{\text{LN}}(\text{FFN}(\mathcal{A}(\boldsymbol{X})) + \mathcal{A}(\boldsymbol{X}))\left(\boldsymbol{J}_{\text{FFN}}(\mathcal{A}(\boldsymbol{X})) + \boldsymbol{I}_{nd}\right)\boldsymbol{J}_{\text{LN}}(\text{ATT}(\boldsymbol{X}) + \boldsymbol{X})\left(\boldsymbol{J}_{\text{ATT}}(\boldsymbol{X}) + \boldsymbol{I}_{nd}\right)
\end{aligned}
$$

and with omitting the evaluation point, we can write the Jacobian as

$$
\boldsymbol{J}_{\text{Post-LN}} = \boldsymbol{J}_{\text{LN}}\left(\boldsymbol{J}_{\text{FFN}} + \boldsymbol{I}_{nd}\right)\boldsymbol{J}_{\text{LN}}\left(\boldsymbol{J}_{\text{ATT}} + \boldsymbol{I}_{nd}\right).
$$

## B.2 Jacobian of layer normalization

Since the layer normalization is a row-wise operation, the Jacobian of the layer normalization for the input matrix $\boldsymbol{X} \in \mathbb{R}^{n \times d}$ is given by

$$
\boldsymbol{J}_{\text{LN}}(X) = \text{blockdiag}(\{\frac{\partial \text{LN}(\boldsymbol{X})_{i,:}}{\partial \boldsymbol{X}_{i,:}}\}_{i=1}^{n}).
$$

where $\frac{\partial \text{LN}(\boldsymbol{X})_{i,:}}{\partial \boldsymbol{X}_{i,:}}$ is the Jacobian of the layer normalization for the $i$-th row of the input matrix $\boldsymbol{X}$. The layer normalization for the $i$-th row of the input matrix $\boldsymbol{X}$ is given by

$$
\text{LN}(\boldsymbol{X})_{i,:} = \frac{\sqrt{d}\widetilde{\boldsymbol{X}_{i,:}}}{\|\widetilde{\boldsymbol{X}_{i,:}}\|},
$$

where $\widetilde{\boldsymbol{X}_{i,:}} \coloneqq \boldsymbol{X}_{i,:}(\boldsymbol{I}_d - \frac{1}{d}\mathbf{1}\mathbf{1}^\top)$. Therefore, the $i$-th block of the Jacobian of the layer normalization is given by

$$
\begin{aligned}
\frac{\partial \text{LN}(\boldsymbol{X})_{i,:}}{\partial \boldsymbol{X}_{i,:}} &= \frac{\partial \text{LN}(\boldsymbol{X})_{i,:}}{\partial \widetilde{\boldsymbol{X}_{i,:}}} \frac{\partial \widetilde{\boldsymbol{X}_{i,:}}}{\partial \boldsymbol{X}_{i,:}} \\
&= \sqrt{d}\left(\frac{1}{\|\widetilde{\boldsymbol{X}_{i,:}}\|}\boldsymbol{I}_d - \widetilde{\boldsymbol{X}_{i,:}}\frac{\widetilde{\boldsymbol{X}_{i,:}}^\top}{\|\widetilde{\boldsymbol{X}_{i,:}}\|^3}\right)\left(\boldsymbol{I}_d - \frac{1}{d}\mathbf{1}\mathbf{1}^\top\right)
\end{aligned}
$$

$$= \frac{\sqrt{d}}{\|\widetilde{\boldsymbol{X}_{i,:}}\|_2}\left(\boldsymbol{I}_d - \frac{\widetilde{\boldsymbol{X}_{i,:}}\widetilde{\boldsymbol{X}_{i,:}}^\top}{\|\widetilde{\boldsymbol{X}_{i,:}}\|_2^2}\right)\left(\boldsymbol{I}_d - \frac{\boldsymbol{1}\boldsymbol{1}^\top}{d}\right).$$

Therefore, we can write the Jacobian of the layer normalization as

$$\boldsymbol{J}_{\mathrm{LN}}(\boldsymbol{X}) = \mathrm{blockdiag}(\{\boldsymbol{L}_i(\boldsymbol{X})\}_{i=1}^n),$$

where

$$\boldsymbol{L}_i(\boldsymbol{X}) = \frac{\sqrt{d}}{\|\widetilde{\boldsymbol{X}_{i,:}}\|_2}\left(\boldsymbol{I}_d - \frac{\widetilde{\boldsymbol{X}_{i,:}}\widetilde{\boldsymbol{X}_{i,:}}^\top}{\|\widetilde{\boldsymbol{X}_{i,:}}\|_2^2}\right)\left(\boldsymbol{I}_d - \frac{\boldsymbol{1}\boldsymbol{1}^\top}{d}\right).$$

### B.3 Jacobian of RMS normalization

Since the RMS normalization is a row-wise operation, the Jacobian of the RMS normalization for the input matrix $\boldsymbol{X} \in \mathbb{R}^{n \times d}$ is given by

$$\boldsymbol{J}_{\mathrm{RMS}}(\boldsymbol{X}) = \mathrm{blockdiag}\left(\left\{\frac{\partial\,\mathrm{RMS}(\boldsymbol{X})_{i,:}}{\partial\boldsymbol{X}_{i,:}}\right\}_{i=1}^n\right).$$

where $\frac{\partial\,\mathrm{RMS}(\boldsymbol{X})_{i,:}}{\partial\boldsymbol{X}_{i,:}}$ is the Jacobian of the RMS normalization for the $i$-th row of the input matrix $\boldsymbol{X}$. The RMS normalization for the $i$-th row of the input matrix $\boldsymbol{X}$ is given by

$$\mathrm{RMS}(\boldsymbol{X})_{i,:} = \mathrm{Diag}(\boldsymbol{\gamma})\frac{\boldsymbol{X}_{i,:}}{r_i}, \qquad r_i := \sqrt{\frac{1}{d}\|\boldsymbol{X}_{i,:}\|_2^2},$$

where $\boldsymbol{\gamma} \in \mathbb{R}^d$ is a learnable scale parameter. Let $\boldsymbol{x} := \boldsymbol{X}_{i,:} \in \mathbb{R}^{d \times 1}$ and $\boldsymbol{y} := \mathrm{RMS}(\boldsymbol{X})_{i,:} \in \mathbb{R}^{d \times 1}$. Then $\boldsymbol{y} = \mathrm{Diag}(\boldsymbol{\gamma})\boldsymbol{x}/r$ with $r = \sqrt{\frac{1}{d}\|\boldsymbol{x}\|_2^2}$.

First, we compute the derivative of $r$ with respect to $\boldsymbol{x}$. Since $r = \left(\frac{1}{d}\boldsymbol{x}^\top\boldsymbol{x}\right)^{1/2}$, we have

$$\frac{\partial r}{\partial \boldsymbol{x}} = \frac{1}{2}\left(\frac{1}{d}\boldsymbol{x}^\top\boldsymbol{x}\right)^{-1/2}\cdot\frac{2}{d}\boldsymbol{x} = \frac{1}{d\,r}\boldsymbol{x}.$$

Hence,

$$\frac{\partial(r^{-1})}{\partial\boldsymbol{x}} = -r^{-2}\frac{\partial r}{\partial\boldsymbol{x}} = -\frac{1}{d\,r^3}\boldsymbol{x}.$$

Therefore, the $i$-th block of the Jacobian of the RMS normalization is given by

$$\begin{aligned}
\frac{\partial\,\mathrm{RMS}(\boldsymbol{X})_{i,:}}{\partial\boldsymbol{X}_{i,:}} &= \mathrm{Diag}(\boldsymbol{\gamma})\frac{\partial(\boldsymbol{x}r^{-1})}{\partial\boldsymbol{x}} \\
&= \mathrm{Diag}(\boldsymbol{\gamma})\left(\frac{1}{r}\boldsymbol{I}_d + \boldsymbol{x}\frac{\partial(r^{-1})}{\partial\boldsymbol{x}^\top}\right) \\
&= \mathrm{Diag}(\boldsymbol{\gamma})\left(\frac{1}{r}\boldsymbol{I}_d - \frac{1}{d\,r^3}\boldsymbol{x}\boldsymbol{x}^\top\right) \\
&= \mathrm{Diag}(\boldsymbol{\gamma})\left(\frac{1}{r_i}\boldsymbol{I}_d - \frac{1}{d\,r_i^3}\boldsymbol{X}_{i,:}\boldsymbol{X}_{i,:}^\top\right) \\
&= \mathrm{Diag}(\boldsymbol{\gamma})\frac{\sqrt{d}}{\|\boldsymbol{X}_{i,:}\|_2}\left(\boldsymbol{I}_d - \frac{\boldsymbol{X}_{i,:}\boldsymbol{X}_{i,:}^\top}{\|\boldsymbol{X}_{i,:}\|_2^2}\right).
\end{aligned}$$

Therefore, we can write the Jacobian of the RMS normalization as

$$\boldsymbol{J}_{\mathrm{RMS}}(\boldsymbol{X}) = \mathrm{blockdiag}\left(\{\boldsymbol{R}_i(\boldsymbol{X})\}_{i=1}^n\right),$$

where

$$\boldsymbol{R}_i(\boldsymbol{X}) = \text{Diag}(\boldsymbol{\gamma}) \frac{\sqrt{d}}{\|\boldsymbol{X}_{i,:}\|_2} \left( \boldsymbol{I}_d - \frac{\boldsymbol{X}_{i,:} \boldsymbol{X}_{i,:}^\top}{\|\boldsymbol{X}_{i,:}\|_2^2} \right), \qquad r_i = \sqrt{\frac{1}{d} \|\boldsymbol{X}_{i,:}\|_2^2}.$$

We note that all the above derivations for Pre-LN and Post-LN architectures remain valid when layer normalization is replaced with RMS normalization, by simply substituting $\boldsymbol{J}_{\text{LN}}$ with $\boldsymbol{J}_{\text{RMS}}$.

# C Experimental details

## C.1 Implementation and training details

Our implementation, based on PyTorch (Paszke et al., 2019), uses the HuggingFace Transformers library (Wolf et al., 2020) for NLP tasks and primarily follows Tomihari & Sato (2024). All experiments were conducted on a single NVIDIA A100 GPU. The reported results are averages over five training seeds. We used the cross-entropy loss, defined as $\ell(\boldsymbol{f}(\boldsymbol{x}), y) \coloneqq -\log\left(\boldsymbol{\sigma}_{\mathrm{SM}}(\boldsymbol{f}(\boldsymbol{x}))_y\right)$, where the function $\boldsymbol{\sigma}_{\mathrm{SM}} : \mathbb{R}^C \to \mathbb{R}^C$ represents the softmax operation.

Following the methodology of Kunstner et al. (2023), we optimized the learning rate via grid search based on the training loss, while keeping other hyperparameters, such as batch size and the number of epochs, fixed. Momentum was set to 0.9 as the default configuration for both SGD and SignSGD. Gradient clipping with a threshold of 1.0 was applied to SGD, which is equivalent to normalized gradient descent up to a constant factor in the learning rate (Zhang et al., 2020a). For NLP tasks, we used linear learning rate scheduling, whereas for vision tasks, a warmup schedule was applied.

Other hyperparameters followed the default values provided by PyTorch, including Adam ($\beta_1 = 0.9$, $\beta_2 = 0.999$, $\epsilon = 1e - 8$) and RMSProp ($\alpha = 0.99$, $\epsilon = 1e - 8$). For NLP tasks, the original training set was split into a 9:1 training-to-validation ratio, with the original validation set used as the test set, following Chen et al. (2022); Tomihari & Sato (2024).

We provide dataset statistics and hyperparameter configurations in Table S.1 and Tables S.2–S.4, respectively.

## C.2 Details of each experiment and figure

**Correlation between Hessian and gradient.** In Figure 1, we show the correlation between the Hessian and the gradient. The maximum eigenvalue of the Hessian was computed using power iteration, as described in Park & Kim (2022), with the PyHessian implementation (Yao et al., 2020). To estimate the maximum eigenvectors of the block-diagonal elements of the Hessian, we calculated the product of the Hessian and a random vector for each parameter. The batch size used for these computations was the same as the training batch size. The maximum eigenvalue and the gradient were computed for each batch across all training data.

**Correlation between full-batch gradient and gradient error.** In Figure 2, we show the correlation between the full-batch gradient and the gradient error in a coordinate-wise manner. We randomly sampled $1,000$ coordinates from the parameters and computed the squared norm of the full-batch gradient and the gradient error for each coordinate. The gradient error is defined as the difference between the full-batch gradient and the gradient computed with a mini-batch. The batch size was the same as the training batch size. The gradient error was computed for each batch across all training data.

**Gradient heterogeneity.** In Figure 3, we show the ratio of the gradient norm for each parameter relative to the sum of the gradient norms. Specifically, we plot:

$$\frac{G_\theta / \sqrt{P_\theta}}{\sum_{\theta'} G_{\theta'} / \sqrt{P_{\theta'}}},$$

for each parameter $\boldsymbol{\theta}$, where $G_\theta$ is the full-batch gradient norm of parameter $\boldsymbol{\theta}$, and $P_\theta$ is its dimension. To compare gradient norms across different parameters, we normalize each gradient norm by the square root of its parameter dimension. Bias parameters are omitted in these plots.

**Effect of layer normalization.** In Tables 2 and S.10, all models share the same RoBERTa backbone and differ only in the placement of the normalization layer. To minimize the effect of initialization, we trained scratch-initialized models for 1000 iterations. Note that pre-trained weights are available only for the Post-LN variant.

**Training Curve.** In Figure 4, we show training runs with the median final loss value among the five training seeds. The shaded area represents the interquartile range across the five seeds. This approach is used to reduce the influence of outliers on the reported results.

Table S.1: Dataset statistics, including the number of classes and counts of training (Train), validation (Val), and test samples for each dataset.

| Domain | Dataset | Classes | Train | Val | Test |
|---|---|---|---|---|---|
| NLP | CB (De Marneffe et al., 2019) | 3 | 225 | 25 | 57 |
| | RTE (Wang et al., 2018) | 2 | 2,241 | 249 | 277 |
| | BoolQ (Clark et al., 2019) | 2 | 8,484 | 943 | 3,270 |
| | WiC (Pilehvar & Camacho-Collados, 2019) | 2 | 5,400 | 600 | 638 |
| | CoLA (Warstadt et al., 2019) | 2 | 7,695 | 855 | 1,040 |
| | SST-2 (Socher et al., 2013) | 2 | 60,614 | 6,735 | 872 |
| | MRPC (Dolan & Brockett, 2005) | 2 | 3,301 | 367 | 408 |
| Vision | Flowers102 (Nilsback & Zisserman, 2008) | 102 | 1,632 | 408 | 6,149 |
| | Aircraft (Maji et al., 2013) | 100 | 5,334 | 1,333 | 3,333 |

Table S.2: Hyperparameter configurations for RoBERTa-Base. The settings include batch size (bs), learning rate (lr), and the number of epochs (epochs). "w/o M" denotes optimizers without momentum and "Const", "Cos", and "Lin-W" denote constant, cosine, and linear with warm-up learning rate schedules, respectively.

| Optimizer | Param | CB | RTE | BoolQ | WiC | CoLA | SST-2 | MRPC |
|---|---|---|---|---|---|---|---|---|
| Common | bs | 8 | 8 | 32 | 32 | 32 | 32 | 32 |
| | epochs | 20 | 20 | 20 | 20 | 20 | 10 | 20 |
| Adam | lr | $1e-4$ | $1e-5$ | $1e-5$ | $1e-5$ | $1e-5$ | $1e-5$ | $1e-5$ |
| SGD | | $1e-2$ | $1e-3$ | $1e-2$ | $1e-3$ | $1e-3$ | $1e-2$ | $1e-2$ |
| SGD (w/o M) | | $1e-1$ | $1e-2$ | $1e-1$ | $1e-2$ | $1e-2$ | $1e-1$ | $1e-1$ |
| SignSGD | | $1e-5$ | $1e-6$ | $1e-5$ | $1e-5$ | $1e-5$ | $1e-5$ | $1e-5$ |
| SignSGD (w/o M) | | $1e-4$ | $1e-5$ | $1e-5$ | $1e-5$ | $1e-4$ | $1e-5$ | $1e-5$ |
| RMSProp | | $1e-5$ | $1e-5$ | $1e-5$ | $1e-5$ | $1e-5$ | $1e-5$ | $1e-5$ |
| SignSGD (S) | | $1e-4$ | $1e-4$ | $1e-4$ | $1e-4$ | $5e-4$ | $1e-4$ | $5e-4$ |
| SGD (Const) | | $1e-2$ | $1e-3$ | - | - | - | - | - |
| SGD (Cos) | | $1e-2$ | $1e-3$ | - | - | - | - | - |
| SGD (Lin-W) | | $1e-2$ | $1e-3$ | - | - | - | - | - |
| SignSGD (Const) | | $1e-6$ | $1e-6$ | - | - | - | - | - |
| SignSGD (Cos) | | $1e-5$ | $1e-5$ | - | - | - | - | - |
| SignSGD (Lin-W) | | $1e-5$ | $1e-5$ | - | - | - | - | - |

Table S.3: Hyperparameter configurations for ResNet18. The settings include batch size (bs), learning rate (lr), and the number of epochs (epochs). "w/o M" denotes optimizers without momentum.

| Optimizer | Param | Flowers102 | Aircraft |
|---|---|---|---|
| Common | bs | 32 | 32 |
| | epochs | 50 | 100 |
| Adam | lr | $1e-4$ | $1e-4$ |
| SGD | | $1e-2$ | $1e-2$ |
| SGD (w/o M) | | $1e-1$ | $1e-1$ |
| SignSGD | | $1e-5$ | $1e-5$ |
| SignSGD (w/o M) | | $1e-4$ | $1e-4$ |
| RMSProp | | $1e-4$ | $1e-4$ |
| SignSGD (S) | | $5e-4$ | $5e-4$ |

Table S.4: Hyperparameter configurations for ViT-Base. The settings include batch size (bs), learning rate (lr), and the number of epochs (epochs)."w/o M" denotes optimizers without momentum.

| Optimizer | Param | Flowers102 | Aircraft |
|---|---|---|---|
| Common | bs | 32 | 32 |
| | epochs | 50 | 100 |
| Adam | | $1e-5$ | $1e-5$ |
| SGD | | $1e-2$ | $1e-2$ |
| SGD (w/o M) | lr | $1e-1$ | $5e-1$ |
| SignSGD | | $1e-5$ | $1e-5$ |
| SignSGD (w/o M) | | $1e-4$ | $1e-5$ |
| RMSProp | | $1e-5$ | $1e-5$ |
| SignSGD (S) | | $5e-4$ | $1e-4$ |

# D    Additional experimental results

## D.1    Empirical estimates of weighted Hessian quantities

We also estimate the weighted Hessian quantities $\Lambda_G$ and $\Lambda_P$ from the same block-wise Hessian and gradient statistics. For each sampled batch, we compute $\widehat{\Lambda}_G = \sum_b \frac{\|[\nabla L(\boldsymbol{\theta})]_b\|_2^2}{\sum_j \|[\nabla L(\boldsymbol{\theta})]_j\|_2^2} \|[\nabla^2 L(\boldsymbol{\theta})]_b\|_2$ and $\widehat{\Lambda}_P = \sum_b \frac{P_b}{\sum_j P_j} \|[\nabla^2 L(\boldsymbol{\theta})]_b\|_2$, and report averages over batches in Table S.5. The results show that $\widehat{\Lambda}_G$ is substantially larger than $\widehat{\Lambda}_P$ for RoBERTa, especially in the pre-trained models, while the separation is smaller for ResNet18 and ViT.

| Model | Dataset | State | $\widehat{\Lambda}_G$ | $\widehat{\Lambda}_P$ | $\widehat{\Lambda}_G/\widehat{\Lambda}_P$ |
|---|---|---|---|---|---|
| RoBERTa-Base | CB | Adam | 0.128 | 0.0246 | 16.6 |
| RoBERTa-Base | CB | Pre-trained | 40.9 | 6.14 | 21.0 |
| RoBERTa-Base | CB | SGD w/ M | 42.1 | 18.3 | 3.57 |
| RoBERTa-Base | RTE | Adam | 0.0781 | 0.0106 | 7.00 |
| RoBERTa-Base | RTE | Pre-trained | 52.6 | 2.28 | 39.9 |
| RoBERTa-Base | RTE | SGD w/ M | $1.14\times10^3$ | 186 | 7.41 |
| ResNet18 | Aircraft | Adam | 52.3 | 30.4 | 1.74 |
| ResNet18 | Aircraft | Pre-trained | 10.1 | 3.93 | 2.58 |
| ResNet18 | Aircraft | SGD w/ M | 73.2 | 61.3 | 1.19 |
| ResNet18 | Flowers102 | Adam | 36.9 | 16.7 | 2.09 |
| ResNet18 | Flowers102 | Pre-trained | 6.96 | 3.44 | 2.02 |
| ResNet18 | Flowers102 | SGD w/ M | 36.7 | 27.0 | 1.34 |
| ViT-Base | Aircraft | Adam | 30.4 | 17.9 | 1.66 |
| ViT-Base | Aircraft | Pre-trained | 3.66 | 5.29 | 0.778 |
| ViT-Base | Aircraft | SGD w/ M | 11.3 | 9.74 | 1.15 |
| ViT-Base | Flowers102 | Adam | 7.52 | 6.53 | 1.09 |
| ViT-Base | Flowers102 | Pre-trained | 4.45 | 7.25 | 0.634 |
| ViT-Base | Flowers102 | SGD w/ M | 4.91 | 4.15 | 1.14 |

Table S.5: Empirical estimates of the weighted Hessian quantities $\widehat{\Lambda}_G$ and $\widehat{\Lambda}_P$ on real models. The Hessian and gradient are computed for the pre-trained model or the fine-tuned model corresponding to the median final loss among five training seeds. Values are averages over sampled batches.

## D.2    Empirical sign reliability

We also estimate the sign-reliability parameter in Assumption 4.8 from existing coordinate-wise full-batch and mini-batch gradient statistics. For each mini-batch, we compute the weighted sign mismatch rate

$$\widehat{q} = \sum_{i=1}^{P_s} \frac{|g_i|}{\sum_{j=1}^{P_s} |g_j|} \mathbb{1}[\text{sign}(\widehat{g}_i) \neq \text{sign}(g_i)],$$

where $g_i$ and $\widehat{g}_i$ denote the full-batch and mini-batch gradients on sampled coordinates. The empirical averages in Table S.6 are below $1/2$ in all analyzed settings, although some settings are close to this boundary.

| Model | Dataset | Mean $\widehat{q}$ | P90 $\widehat{q}$ | Mean reliability |
|-------|---------|--------|--------|------------------|
| RoBERTa-Base | CB | 0.145 | 0.209 | 0.855 |
| RoBERTa-Base | RTE | 0.429 | 0.867 | 0.571 |
| ResNet18 | Aircraft | 0.439 | 0.475 | 0.561 |
| ResNet18 | Flowers102 | 0.457 | 0.496 | 0.543 |
| ViT-Base | Aircraft | 0.484 | 0.529 | 0.516 |
| ViT-Base | Flowers102 | 0.476 | 0.530 | 0.524 |

Table S.6: Empirical estimates of the weighted sign mismatch rate $\widehat{q}$ from sampled coordinates. Mean reliability is $1 - \mathbb{E}[\widehat{q}]$.

### D.3 Correlation between Hessian and gradient

We show the correlation between the Hessian and the gradient in Figure S.3. The Hessian and gradient are computed using the pre-trained models or the trained models corresponding to the median final loss value among the five training seeds shown in Figures 4 and S.7 and Appendix D.8.

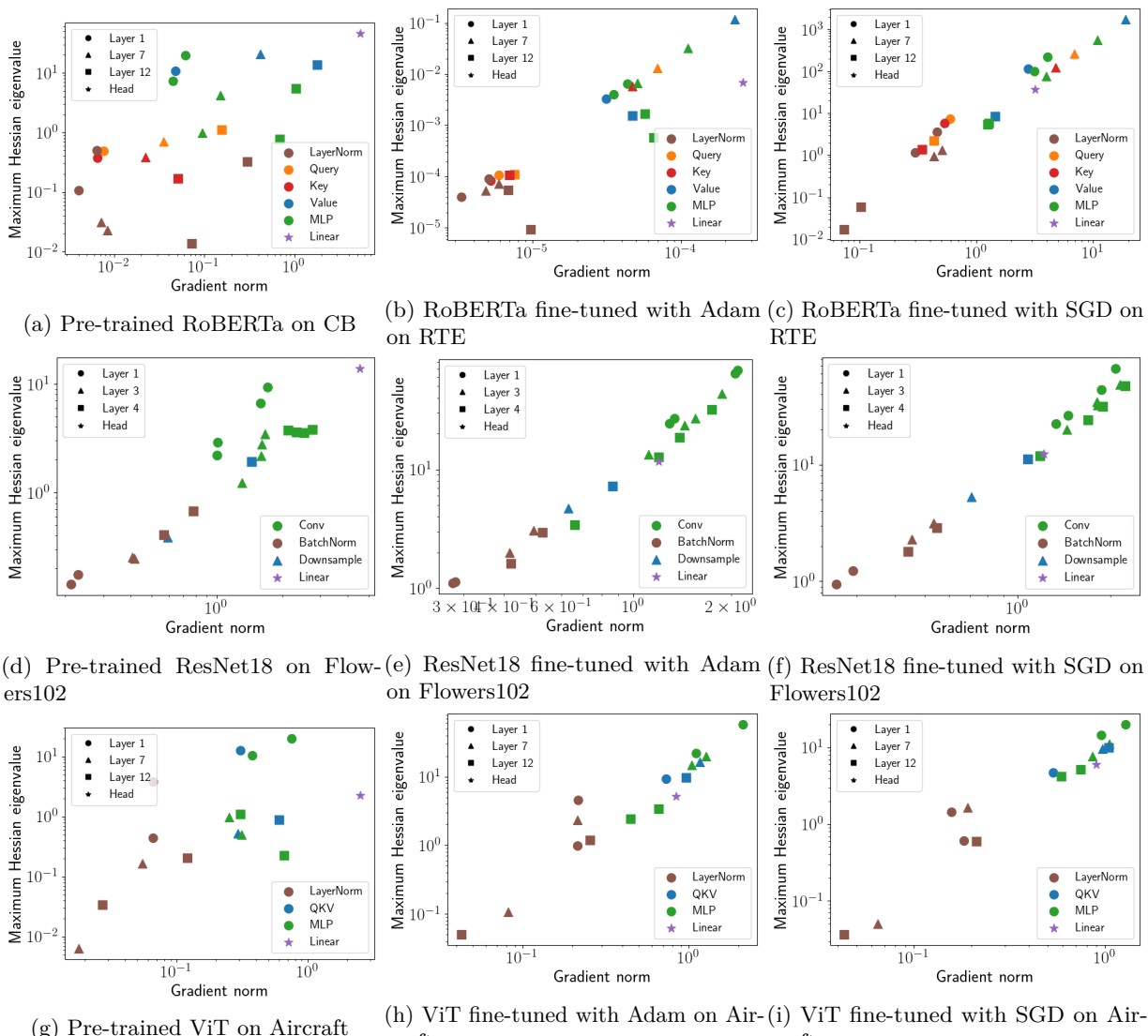

(a) Pre-trained RoBERTa on CB

(b) RoBERTa fine-tuned with Adam on RTE

(c) RoBERTa fine-tuned with SGD on RTE

(d) Pre-trained ResNet18 on Flowers102

(e) ResNet18 fine-tuned with Adam on Flowers102

(f) ResNet18 fine-tuned with SGD on Flowers102

(g) Pre-trained ViT on Aircraft

(h) ViT fine-tuned with Adam on Aircraft

(i) ViT fine-tuned with SGD on Aircraft

Figure S.3: Gradient vs. Hessian matrix.

### D.4 Correlation between Hessian and parameter dimension

We show the correlation between the Hessian and the parameter in Figure S.4. The Hessian and parameter dimension do not show a clear correlation.

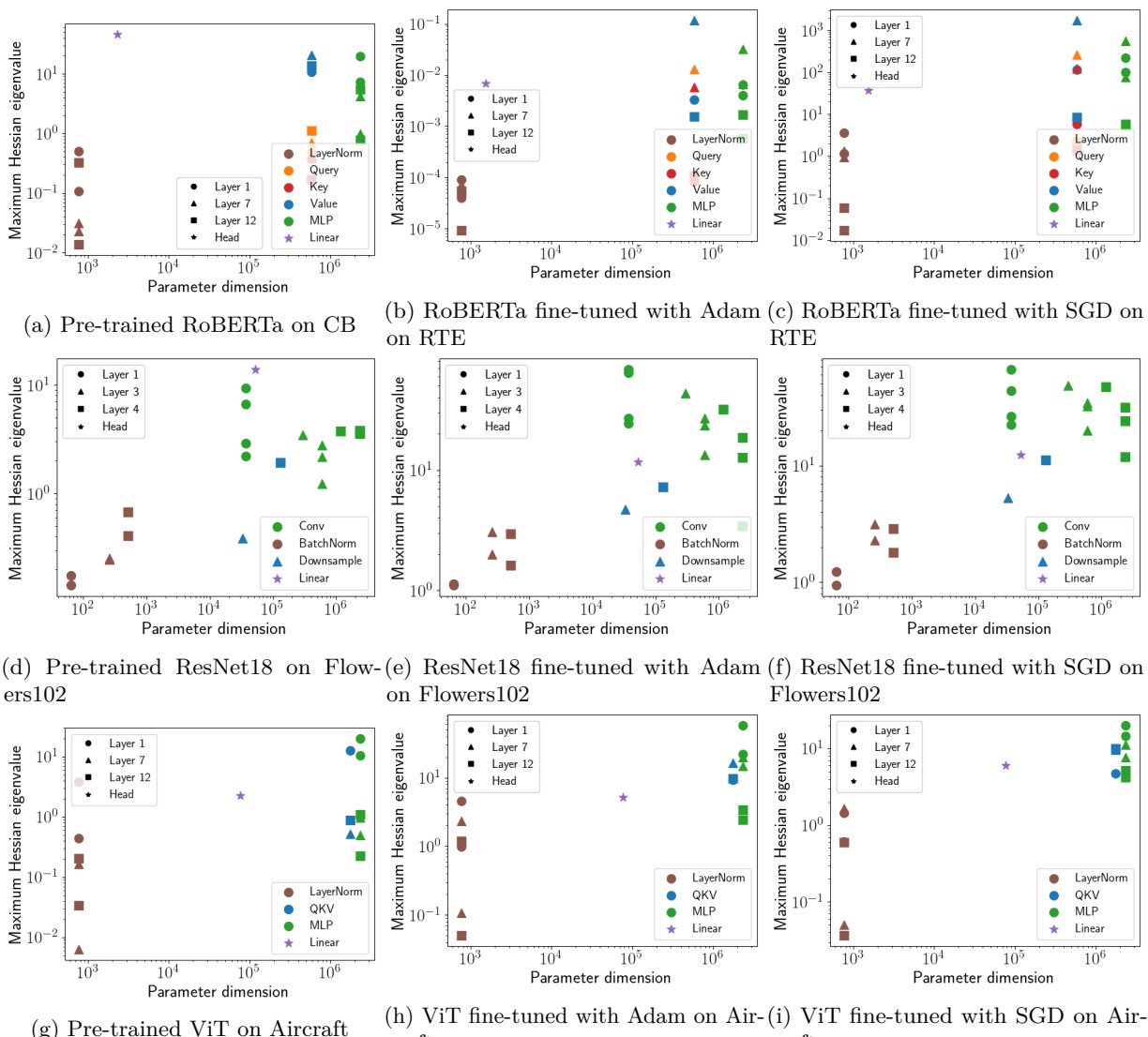

(a) Pre-trained RoBERTa on CB

(b) RoBERTa fine-tuned with Adam on RTE

(c) RoBERTa fine-tuned with SGD on RTE

(d) Pre-trained ResNet18 on Flowers102

(e) ResNet18 fine-tuned with Adam on Flowers102

(f) ResNet18 fine-tuned with SGD on Flowers102

(g) Pre-trained ViT on Aircraft

(h) ViT fine-tuned with Adam on Aircraft

(i) ViT fine-tuned with SGD on Aircraft

Figure S.4: Parameter dimension vs. Hessian matrix.

### D.5 Correlation between full-batch gradient and gradient error

We show the correlation between the full-batch gradient and the gradient error in Figure S.5.

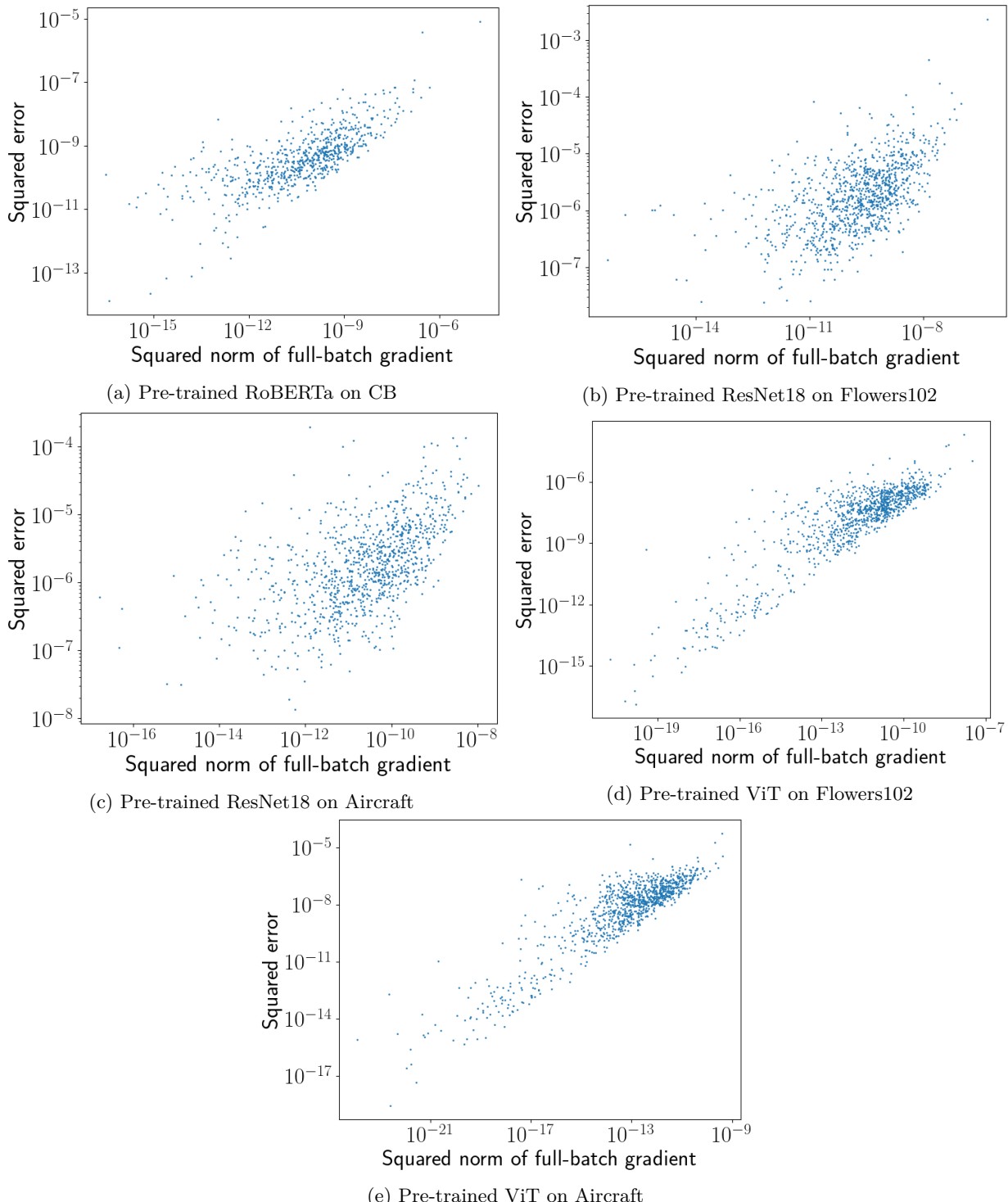

(a) Pre-trained RoBERTa on CB

(b) Pre-trained ResNet18 on Flowers102

(c) Pre-trained ResNet18 on Aircraft

(d) Pre-trained ViT on Flowers102

(e) Pre-trained ViT on Aircraft

Figure S.5: coordinate-wise full-batch gradient vs. gradient error.

## D.6 Gradient per parameter

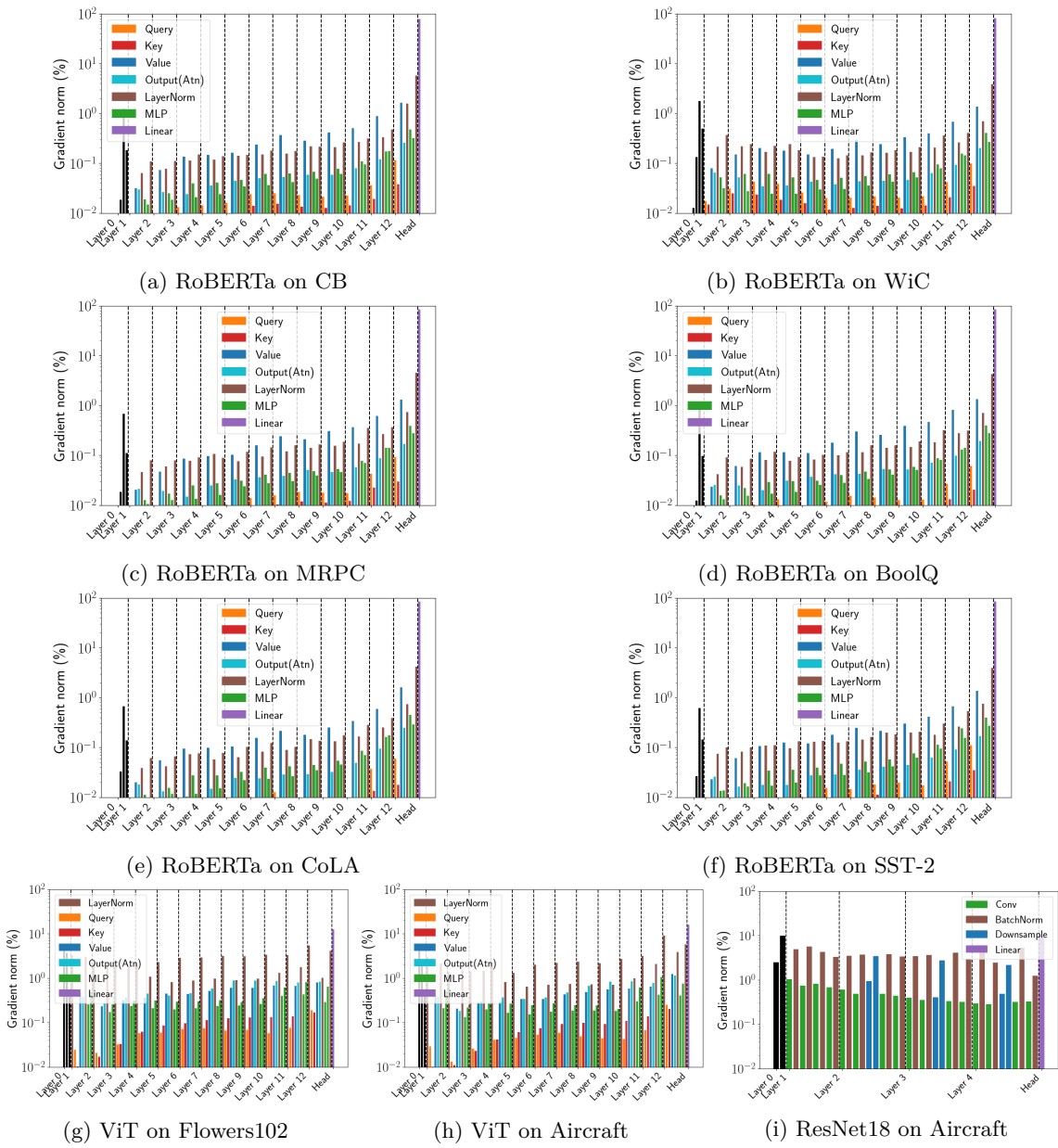

(a) RoBERTa on CB

(b) RoBERTa on WiC

(c) RoBERTa on MRPC

(d) RoBERTa on BoolQ

(e) RoBERTa on CoLA

(f) RoBERTa on SST-2

(g) ViT on Flowers102

(h) ViT on Aircraft

(i) ResNet18 on Aircraft

Figure S.6: Gradient norm of each parameter of pre-trained model.

### D.7 Quantitative measures of gradient heterogeneity

**Gini coefficient.** In Table S.7, we provide the Gini coefficient of the normalized gradients.

Gini coefficient is a measure of statistical dispersion intended to represent the inequality of a distribution, which ranges from 0 to 1 and the higher value indicates more heterogeneity.

Given a set of values $\{x_1, x_2, \ldots, x_n\}$ sorted in non-decreasing order, the Gini coefficient is defined as:

$$G = \frac{\sum_{i=1}^{n} \sum_{j=1}^{n} |x_i - x_j|}{2n^2 \bar{x}},$$

where $\bar{x}$ is the mean of the values.

**Layer-wise gradient norm ratio.** In Table S.8, we present the ratio of the gradient norm for each layer, computed as:

$$\frac{G_l}{\sum_{l'} G_{l'}},$$

where $G_l$ represents the sum of the normalized full-batch gradient norms of the parameters in layer $l$. Since all layers contain the same number of parameters, this comparison is valid.

| Model (Dataset) | Gini coefficient |
|---|---|
| RoBERTa-Base (CB) | $0.932 \pm 0.006$ |
| RoBERTa-Base (RTE) | $0.944 \pm 0.005$ |
| RoBERTa-Base (WiC) | $0.931 \pm 0.004$ |
| RoBERTa-Base (BoolQ) | $0.944 \pm 0.001$ |
| RoBERTa-Base (CoLA) | $0.954 \pm 0.003$ |
| RoBERTa-Base (MRPC) | $0.951 \pm 0.001$ |
| RoBERTa-Base (SST-2) | $0.930 \pm 0.032$ |
| ResNet-18 (Flowers102) | $0.407 \pm 0.013$ |
| ResNet-18 (Aircraft) | $0.433 \pm 0.005$ |
| ViT-Base (Flowers102) | $0.539 \pm 0.004$ |
| ViT-Base (Aircraft) | $0.598 \pm 0.009$ |

Table S.7: Gini coefficient of normalized gradients. $\pm$ represents standard deviation.

| Layer | 1 | 2 | 3 | 4 | 5 | 6 | 7 | 8 | 9 | 10 | 11 | 12 |
|---|---|---|---|---|---|---|---|---|---|---|---|---|
| RoBERTa-Base (CB) | $0.021 \pm 0.001$ | $0.022 \pm 0.001$ | $0.027 \pm 0.002$ | $0.031 \pm 0.002$ | $0.036 \pm 0.002$ | $0.045 \pm 0.002$ | $0.054 \pm 0.002$ | $0.060 \pm 0.003$ | $0.070 \pm 0.004$ | $0.092 \pm 0.005$ | $0.156 \pm 0.015$ | $0.387 \pm 0.027$ |
| RoBERTa-Base (RTE) | $0.023 \pm 0.003$ | $0.024 \pm 0.003$ | $0.028 \pm 0.003$ | $0.030 \pm 0.003$ | $0.034 \pm 0.002$ | $0.042 \pm 0.002$ | $0.051 \pm 0.004$ | $0.058 \pm 0.003$ | $0.068 \pm 0.003$ | $0.093 \pm 0.008$ | $0.163 \pm 0.014$ | $0.387 \pm 0.023$ |
| RoBERTa-Base (WiC) | $0.047 \pm 0.014$ | $0.042 \pm 0.010$ | $0.041 \pm 0.005$ | $0.040 \pm 0.003$ | $0.036 \pm 0.002$ | $0.040 \pm 0.003$ | $0.049 \pm 0.004$ | $0.055 \pm 0.004$ | $0.063 \pm 0.003$ | $0.086 \pm 0.006$ | $0.145 \pm 0.009$ | $0.355 \pm 0.035$ |
| RoBERTa-Base (BoolQ) | $0.023 \pm 0.001$ | $0.024 \pm 0.001$ | $0.028 \pm 0.001$ | $0.031 \pm 0.002$ | $0.034 \pm 0.002$ | $0.043 \pm 0.002$ | $0.055 \pm 0.003$ | $0.062 \pm 0.004$ | $0.073 \pm 0.004$ | $0.098 \pm 0.007$ | $0.157 \pm 0.010$ | $0.370 \pm 0.034$ |
| RoBERTa-Base (CoLA) | $0.017 \pm 0.001$ | $0.018 \pm 0.001$ | $0.023 \pm 0.003$ | $0.025 \pm 0.002$ | $0.029 \pm 0.002$ | $0.037 \pm 0.003$ | $0.042 \pm 0.002$ | $0.048 \pm 0.002$ | $0.058 \pm 0.003$ | $0.083 \pm 0.006$ | $0.169 \pm 0.013$ | $0.451 \pm 0.027$ |
| RoBERTa-Base (MRPC) | $0.019 \pm 0.002$ | $0.020 \pm 0.002$ | $0.024 \pm 0.002$ | $0.028 \pm 0.002$ | $0.032 \pm 0.002$ | $0.040 \pm 0.002$ | $0.049 \pm 0.003$ | $0.057 \pm 0.004$ | $0.067 \pm 0.004$ | $0.089 \pm 0.007$ | $0.155 \pm 0.010$ | $0.421 \pm 0.037$ |
| RoBERTa-Base (SST-2) | $0.025 \pm 0.010$ | $0.026 \pm 0.010$ | $0.032 \pm 0.012$ | $0.036 \pm 0.012$ | $0.040 \pm 0.013$ | $0.046 \pm 0.012$ | $0.054 \pm 0.014$ | $0.061 \pm 0.014$ | $0.070 \pm 0.009$ | $0.087 \pm 0.008$ | $0.148 \pm 0.022$ | $0.373 \pm 0.086$ |
| ViT-Base (Flowers102) | $0.093 \pm 0.004$ | $0.065 \pm 0.002$ | $0.073 \pm 0.002$ | $0.071 \pm 0.004$ | $0.069 \pm 0.003$ | $0.071 \pm 0.005$ | $0.075 \pm 0.005$ | $0.079 \pm 0.003$ | $0.083 \pm 0.005$ | $0.094 \pm 0.002$ | $0.105 \pm 0.005$ | $0.122 \pm 0.004$ |
| ViT-Base (Aircraft) | $0.083 \pm 0.005$ | $0.058 \pm 0.003$ | $0.067 \pm 0.003$ | $0.063 \pm 0.003$ | $0.058 \pm 0.002$ | $0.063 \pm 0.003$ | $0.068 \pm 0.001$ | $0.073 \pm 0.002$ | $0.077 \pm 0.003$ | $0.090 \pm 0.001$ | $0.119 \pm 0.005$ | $0.181 \pm 0.011$ |

Table S.8: Layer-wise ratio of gradient norms in Transformers. $\pm$ represents standard deviation.

### D.8 Train curves

We show the training curves on different datasets from that in the main text. On the CB dataset, the final train loss is similar among all optimizers, but the convergence speed of SGD is slower than other optimizers. This is consistent with our analysis suggesting that SGD can converge more slowly on RoBERTa under large gradient heterogeneity.

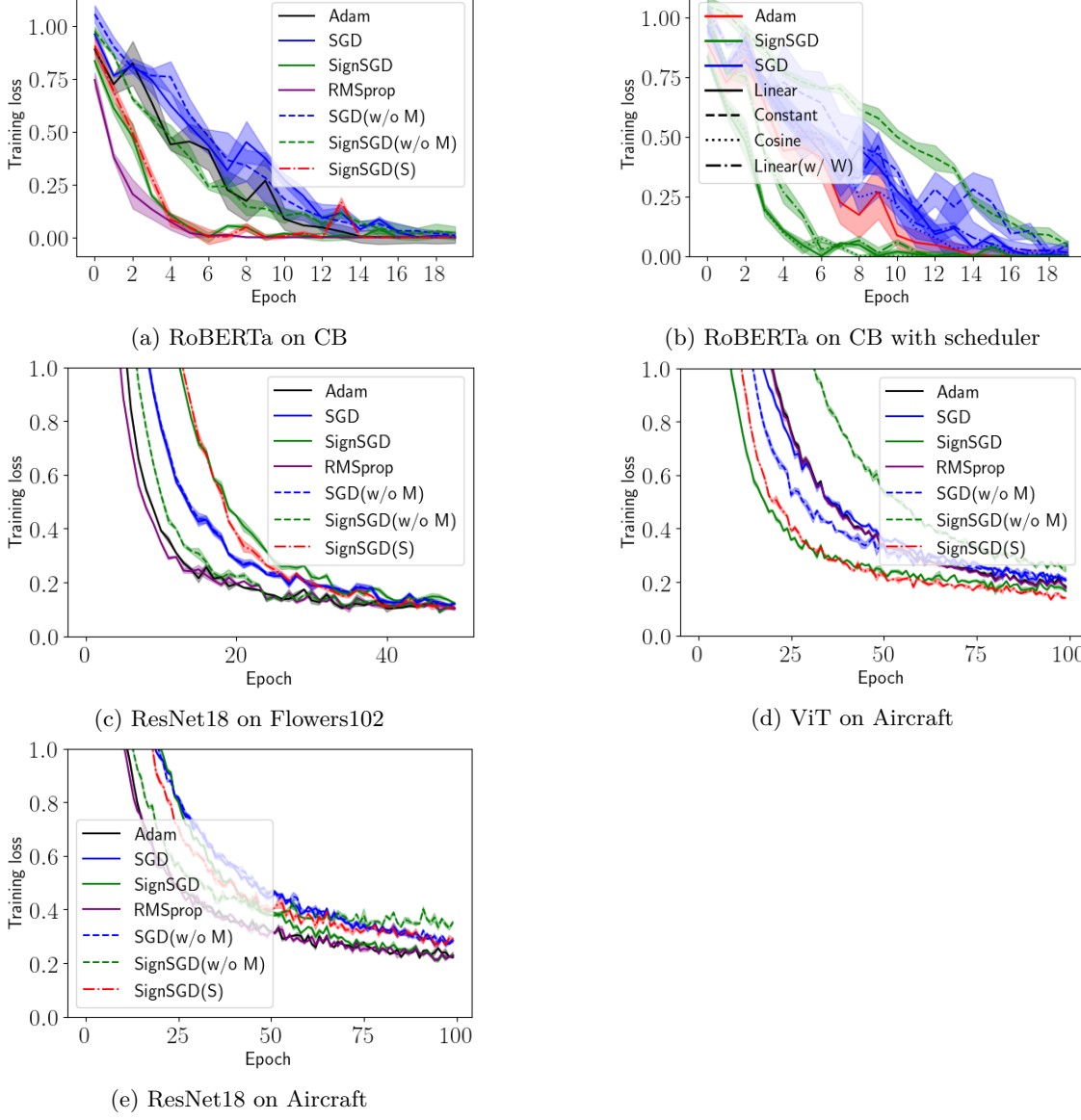

Figure S.7: Training curve with different optimizers. w/ W indicates "with warmup".

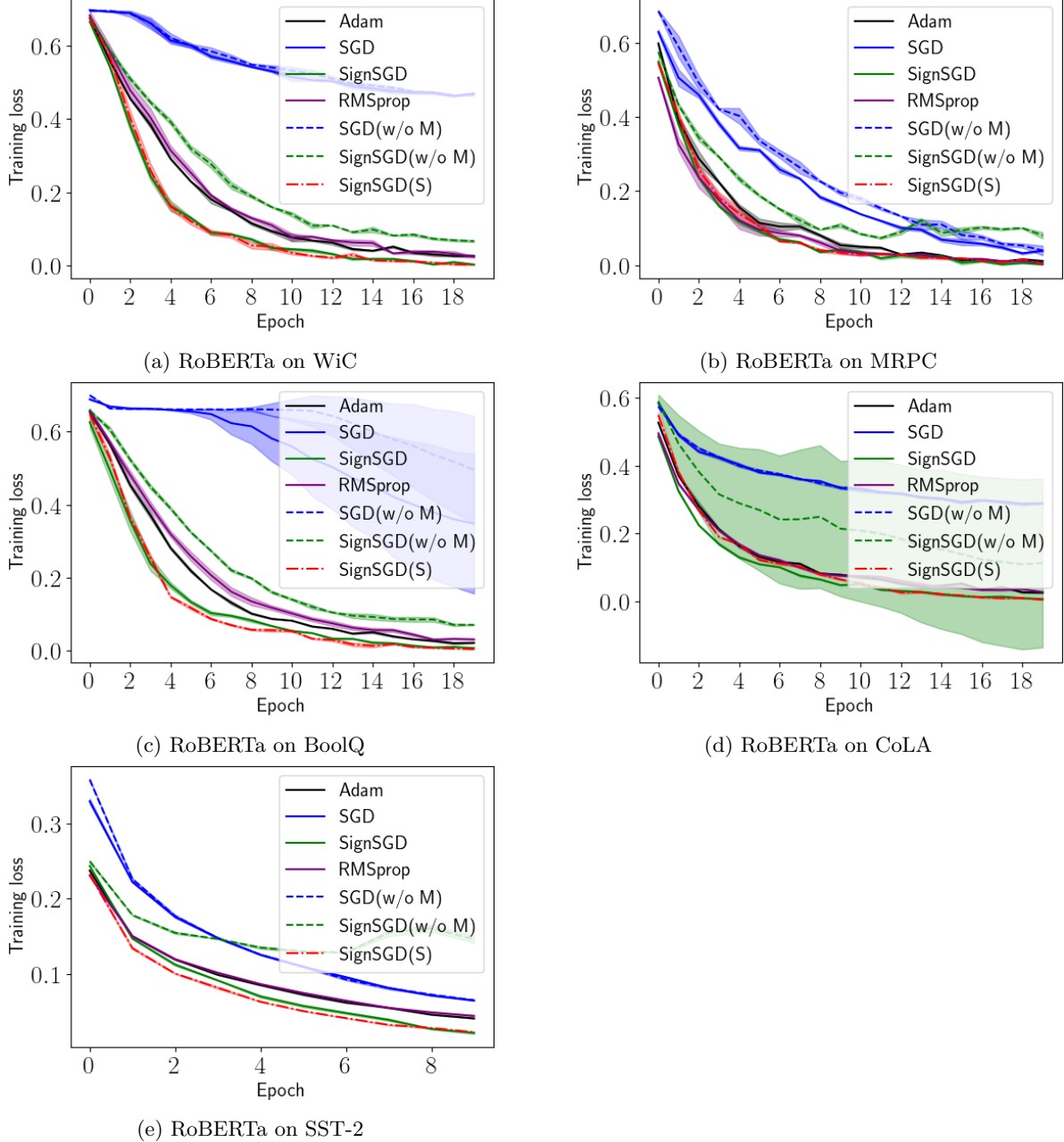

(a) RoBERTa on WiC

(b) RoBERTa on MRPC

(c) RoBERTa on BoolQ

(d) RoBERTa on CoLA

(e) RoBERTa on SST-2

Figure S.8: Training curve with different optimizers.

### D.9 Test results

Table S.9: Test results corresponding to the training curves shown in Figures 4 and S.7. We report the accuracy and its standard deviation.

| Model | Dataset | Adam | RMSprop | SGD | SignSGD | SGD(w/o M) | SignSGD(w/o M) |
|---|---|---|---|---|---|---|---|
| ViT-Base | Flowers102 | $95.06 \pm 0.34$ | $95.15 \pm 0.41$ | $94.22 \pm 0.54$ | $94.01 \pm 0.98$ | $94.49 \pm 0.62$ | $92.45 \pm 1.35$ |
| | Aircraft | $74.28 \pm 0.59$ | $74.86 \pm 0.87$ | $71.33 \pm 0.27$ | $73.96 \pm 0.73$ | $55.25 \pm 0.67$ | $75.21 \pm 0.88$ |
| ResNet18 | Flowers102 | $93.33 \pm 0.62$ | $93.27 \pm 0.71$ | $93.40 \pm 0.47$ | $94.43 \pm 0.54$ | $93.03 \pm 0.62$ | $93.10 \pm 0.37$ |
| | Aircraft | $71.95 \pm 0.69$ | $70.53 \pm 0.42$ | $72.66 \pm 0.71$ | $72.01 \pm 0.40$ | $72.16 \pm 0.41$ | $70.87 \pm 0.35$ |
| RoBERTa-Base | CB | $76.43 \pm 7.41$ | $84.29 \pm 4.96$ | $78.21 \pm 6.36$ | $83.21 \pm 2.71$ | $71.79 \pm 12.46$ | $77.86 \pm 2.99$ |
| | RTE | $75.88 \pm 1.56$ | $74.66 \pm 2.89$ | $75.31 \pm 3.12$ | $75.02 \pm 2.30$ | $73.21 \pm 1.83$ | $75.74 \pm 2.74$ |

### D.10 Effect of layer normalization

Table S.10: Gini coefficients of gradient norms for different normalization. A higher Gini coefficient indicates greater heterogeneity. "No-LN" refers to the architecture without layer normalization.

| Norm Type | Init | Dataset | Gini Coefficient |
|---|---|---|---|
| No-LN | Scratch | RTE | $0.867 \pm 0.006$ |
| Pre-LN | Scratch | RTE | $0.880 \pm 0.004$ |
| Post-LN | Scratch | RTE | $0.941 \pm 0.012$ |
| Post-LN | Pre-trained | RTE | $0.944 \pm 0.005$ |
| No-LN | Scratch | CB | $0.850 \pm 0.049$ |
| Pre-LN | Scratch | CB | $0.873 \pm 0.017$ |
| Post-LN | Scratch | CB | $0.899 \pm 0.018$ |
| Post-LN | Pre-trained | CB | $0.932 \pm 0.006$ |

### D.11 Case study: Quadratic model

Following Zhang et al. (2024a), we consider a synthetic quadratic minimization problem of the form

$$L(\boldsymbol{\theta}) = \frac{1}{2}\boldsymbol{\theta}^\top \boldsymbol{H}\boldsymbol{\theta},$$

where $\boldsymbol{H} = \mathrm{blockdiag}(\{\boldsymbol{H}_i\}_{i=1}^3)$ is a block-diagonal matrix. Each block $\boldsymbol{H}_i$ is constructed as $\boldsymbol{H}_i = \boldsymbol{Q}_i \boldsymbol{\Lambda}_i \boldsymbol{Q}_i^\top$, where $\boldsymbol{Q}_i$ is an orthogonal matrix and $\boldsymbol{\Lambda}_i$ is a diagonal matrix containing the eigenvalues of the block.

We consider two settings for the eigenvalue configurations of $\boldsymbol{\Lambda}_i$, following Zhang et al. (2024a):

- **Homogeneous (Homo):** $\{1, 99, 4998\}$, $\{2, 100, 4999\}$, $\{3, 101, 5000\}$,

- **Heterogeneous (Hetero):** $\{1, 2, 3\}$, $\{99, 100, 101\}$, $\{4998, 4999, 5000\}$,

for $i = 1, 2, 3$.

We use fixed learning rates for both SGD and sign-based optimization methods. For SGD, the learning rate is set to (Nesterov, 2013)

$$\eta = \frac{2}{\lambda_{\min} + \lambda_{\max}},$$

where $\lambda_{\min} = 1$ and $\lambda_{\max} = 5000$ denote the smallest and largest eigenvalues of $\boldsymbol{H}$, respectively. For SignSGD, we adopt the theoretically optimal learning rate derived from our analysis, as detailed in Appendix D.11.1.

Figure S.9 shows the evolution of the $\ell_2$ norm of the gradient during optimization, and Table S.11 reports the weighted Hessian complexities $\Lambda_G$ and $\Lambda_P$ defined in Definition 4.4. In this quadratic setting, $\Lambda_P$ can be computed exactly, whereas $\Lambda_G$ involves a supremum over the fine-tuning region $\mathcal{R}_{\mathrm{FT}}$ and cannot be evaluated in closed form. We therefore approximate $\Lambda_G$ by $\sup_{t \in \{0,...,T\}} \sum_{b=1}^B \frac{\|[\nabla L(\boldsymbol{\theta}_t)]_b\|_2^2}{\|\nabla L(\boldsymbol{\theta}_t)\|_2^2} \|[\nabla^2 L(\boldsymbol{\theta}_t)]_b\|_2$[1].

For SGD, the values of $\Lambda_G$ are nearly identical in the Homo and Hetero settings, and the corresponding gradient norm trajectories exhibit similar behavior. In contrast, for SignSGD, $\Lambda_P$ is substantially larger in the Homo setting than in the Hetero setting, which is reflected in a slower decay of the gradient norm (except in the early stage) and a larger number of iterations required to reach a small norm. These observations align with our theoretical prediction that the iteration complexity of gradient-based and sign-based methods is characterized by $\Lambda_G$ and $\Lambda_P$, respectively.

Moreover, the pronounced gap between $\Lambda_G$ and $\Lambda_P$ in the Hetero setting reflects strong gradient heterogeneity and gradient–Hessian correlation, explaining why sign-based methods are more effective in the Hetero setting. Furthermore, the optimization advantage of Adam over SGD in the heterogeneous quadratic setting reported by Zhang et al. (2024a) is also observed here when comparing SignSGD with SGD.

Table S.11: Values of the weighted Hessian complexities $\Lambda_G$ and $\Lambda_P$ in the quadratic model.

|  | Homo | Hetero |
|---|---|---|
| $\Lambda_G$ | 4999.9997 | 4999.9997 |
| $\Lambda_P$ | 4999.0000 | 1701.3333 |

---

[1]While this computation restricts the supremum to the optimization trajectory and is therefore formally smaller than the supremum over $\boldsymbol{\theta} \in \mathcal{R}_{\mathrm{FT}}$, the resulting value is nearly maximal over $\boldsymbol{\theta} \in \mathbb{R}^P$, indicating that this approximation is inconsequential.

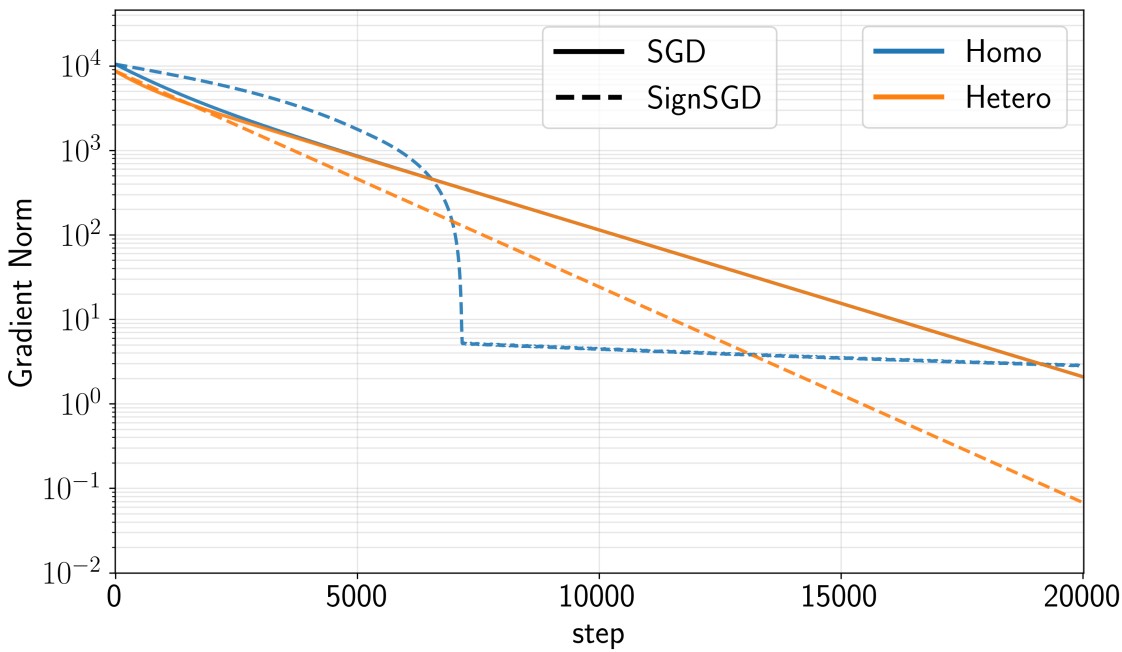

Figure S.9: Evolution of the $\ell_2$ norm of the gradient in the quadratic model.

### D.11.1 Optimal learning rate and upper bounds

**SignSGD.** We derive the optimal learning rate for SignSGD in the quadratic setting. For the quadratic model, we have $\delta_D = \rho_H = 0$. Therefore, from Eq. (10), we obtain

$$L(\boldsymbol{\theta}_{t+1}^{\mathrm{Sign}}) - L(\boldsymbol{\theta}_t^{\mathrm{Sign}}) \le -\eta_t \|\nabla L(\boldsymbol{\theta}_t^{\mathrm{Sign}})\|_1 + \frac{\eta_t^2}{2}\Lambda_P P + \frac{\eta_t^2}{2}\delta_D P + \eta_t^3 \frac{\rho_H}{6} P^{3/2}$$

$$= -\eta_t \|\nabla L(\boldsymbol{\theta}_t^{\mathrm{Sign}})\|_1 + \frac{\eta_t^2}{2}\Lambda_P P.$$

The right-hand side is minimized when

$$\eta_t = \frac{\|\nabla L(\boldsymbol{\theta}_t^{\mathrm{Sign}})\|_1}{P\Lambda_P},$$

which is the learning rate used in our experiments. This scaling is consistent with steepest descent with respect to the $\ell_\infty$-norm (Kelner et al., 2014; Carlson et al., 2015; Balles et al., 2020).

Using this learning rate, we can derive an upper bound on the iteration complexity for the quadratic model by following the same argument as in the general setting:

$$\mathcal{T}_\varepsilon(\{\boldsymbol{\theta}_t^{\mathrm{Sign}}\}_{t=0}^\infty, L, \|\cdot\|_1) \le \frac{2\big(L(\boldsymbol{\theta}_0) - L_*\big)}{P\varepsilon^2}\Lambda_P. \tag{19}$$

**SGD.** Analogously to the SignSGD case, starting from Eq. (9), we obtain the following inequality:

$$L(\boldsymbol{\theta}_{t+1}^{\mathrm{Grad}}) - L(\boldsymbol{\theta}_t^{\mathrm{Grad}}) \le -\eta_t \|\nabla L(\boldsymbol{\theta}_t^{\mathrm{Grad}})\|_2^2 + \frac{\eta_t^2}{2}\Lambda_G\|\nabla L(\boldsymbol{\theta}_t^{\mathrm{Grad}})\|_2^2 + \frac{\eta_t^2}{2}\delta_D\|\nabla L(\boldsymbol{\theta}_t^{\mathrm{Grad}})\|_2^2 + \eta_t^3\frac{\rho_H}{6}\|\nabla L(\boldsymbol{\theta}_t^{\mathrm{Grad}})\|_2^3$$

$$= -\eta_t \|\nabla L(\boldsymbol{\theta}_t^{\mathrm{Grad}})\|_2^2 + \frac{\eta_t^2}{2}\Lambda_G\|\nabla L(\boldsymbol{\theta}_t^{\mathrm{Grad}})\|_2^2,$$

where we use $\delta_D = \rho_H = 0$ for the quadratic model. The right-hand side is minimized when

$$\eta_t = \frac{1}{\Lambda_G}. \tag{20}$$

Using this learning rate, we derive an upper bound on the iteration complexity for the quadratic model by following the same argument as in the general setting:

$$\mathcal{T}_\varepsilon(\{\boldsymbol{\theta}_t^{\text{Grad}}\}_{t=0}^\infty, L, \|\cdot\|_2) \leq \frac{2\big(L(\boldsymbol{\theta}_0) - L_*\big)}{P\varepsilon^2}\Lambda_G.$$

This bound has the same form as Eq. (19), with $\Lambda_G$ replacing $\Lambda_P$.

From Table S.11, we observe that $\Lambda_G$ is approximately equal to $\lambda_{\max}$. Therefore, comparing Eq. (20) with the classical optimal learning rate for quadratic objectives,

$$\eta = \frac{2}{\lambda_{\min} + \lambda_{\max}}, \tag{21}$$

we obtain

$$\frac{1}{\Lambda_G} \leq \frac{2}{\lambda_{\min} + \lambda_{\max}}.$$

Empirically, we observed faster convergence when using the larger learning rate given by Eq. (21). Consequently, we adopt this learning rate in our experiments.

### D.12  Applicability beyond fine-tuning settings

To test generalization beyond fine-tuning, we trained nanoGPT from scratch on the Shakespeare dataset. Adam outperformed SGD, and SignSGD remained competitive. We also found that gradient heterogeneity in nanoGPT lies between that of ViT/ResNet and RoBERTa. Despite the different setup, the results align with our analysis.

Table S.12: Training loss for nanoGPT trained from scratch on the Shakespeare dataset. "Min" denotes the lowest observed loss during training, and "Last" denotes the final loss at the end of training.

| Optimizer | Min | Last |
|-----------|-----|------|
| Adam | $0.658 \pm 0.009$ | $0.687 \pm 0.019$ |
| SGD | $0.928 \pm 0.120$ | $0.964 \pm 0.122$ |
| SignSGD | $0.791 \pm 0.011$ | $0.820 \pm 0.017$ |

Table S.13: Gini coefficient of gradient norms for nanoGPT on the Shakespeare dataset. A higher Gini coefficient indicates greater gradient heterogeneity.

| Model (Dataset) | Gini Coefficient |
|-----------------|------------------|
| nanoGPT (Shakespeare) | $0.609 \pm 0.004$ |

## E  Discussion on momentum in SignSGD

The impact of the momentum term used in Adam has not been considered in the analysis so far. However, in sample-wise training, the presence of a momentum term significantly affects the updates of the linear head, particularly for the bias term.

**Model.**  The model $\boldsymbol{f}$ comprises a pre-trained feature extractor $\boldsymbol{\phi}(\cdot) : \mathcal{X} \to \mathbb{R}^h$ and a linear head with weight $\boldsymbol{V} \in \mathbb{R}^{C \times h}$ and bias $\boldsymbol{b} \in \mathbb{R}^C$. The output is given by $\boldsymbol{f}(\boldsymbol{x}) = \boldsymbol{V}\boldsymbol{\phi}(\boldsymbol{x}) + \boldsymbol{b}$.

**Proposition E.1** (SignSGD without momentum)**.** *Let $\Delta^S\theta$ and $\Delta^F\theta$ denote the one-epoch updates of a parameter $\theta$ during sample-wise and full-batch training, respectively. For a linear head trained using the cross-entropy loss and SignSGD with a learning rate $\eta$, the updates are as follows:*

*For the bias term $b_k$:*

$$\Delta^S b_k = -\frac{\eta}{N} \sum_{i=1}^{N} (1 - 2 \cdot \mathbb{1}[y^{(i)} = k]), \quad \Delta^F b_k = -\eta \operatorname{sign}\left( \sum_{i=1}^{N} \delta_{p_k}^{(i)} \right),$$

*and for the weight matrix $V_{k,l}$:*

$$\Delta^S V_{k,l} = -\frac{\eta}{N} \left( \sum_{y^{(i)} \neq k} s_l^{(i)} - \sum_{y^{(i)} = k} s_l^{(i)} \right), \quad \Delta^F V_{k,l} = -\eta \operatorname{sign}\left( \sum_{i=1}^{N} \boldsymbol{\phi}(\boldsymbol{x}^{(i)})_l \delta_{p_k}^{(i)} \right),$$

*where $\delta_{p_k}^{(i)} := \boldsymbol{\sigma}_{\mathrm{SM}}(\boldsymbol{f}(\boldsymbol{x}^{(i)}))_k - \mathbb{1}[k = y^{(i)}]$ represents the prediction error for the $i$-th sample and class $k$ and $s_l^{(i)} := \operatorname{sign}\left( \boldsymbol{\phi}(\boldsymbol{x}^{(i)})_l \right)$ is the sign of the $l$-th element of the feature embedding $\boldsymbol{\phi}(\boldsymbol{x}^{(i)})_l$.*

**Sign-alignment causes large updates.** In full-batch training, the updates $\Delta^F b_k$ and $\Delta^F V_{k,l}$ depend on the model predictions. Because the signs of these updates vary across epochs, these updates remain small. In contrast, in sample-wise training, update signs can align across epochs, resulting in disproportionately large updates. This effect is particularly pronounced for the bias term $\Delta^S b_k$, which is independent of model predictions and grows with the number of classes. Similarly, the sign of $\Delta^S V_{k,l}$, which depends on the feature extractor output $\boldsymbol{\phi}(\boldsymbol{x}^{(i)})$, may align across epochs.

**Momentum resolves the issue.** Excessively large updates can cause training instability and incorrect predictions. Although the proposition specifically addresses sample-wise updates, similar challenges can arise in batch training. Momentum, which estimates the full-batch gradient using exponential moving averages, effectively mitigates this problem.

### E.1 Experimental results

We show the norm of the linear head for different datasets, models, and optimizers. The results indicate that when the number of classes is large, the bias term of the linear head exhibits a larger norm with SignSGD without momentum compared to other optimizers. In contrast, the weight norm does not necessarily increase under the same conditions, even with SignSGD without momentum. This observation aligns with the theoretical analysis in Proposition E.1, which suggests that a large number of classes leads to an increase in the bias term norm, while the weight norm is influenced by the sign of the feature extractor outputs.

### E.2 Proof of Proposition E.1

*Proof.* The partial derivative of the bias and the weight matrix with the cross-entropy loss is given by:

$$
\begin{aligned}
\frac{\partial \ell(\boldsymbol{f}(\boldsymbol{x}^{(i)}, y^{(i)}))}{\partial b_k} &= \frac{\partial \ell(\boldsymbol{f}(\boldsymbol{x}^{(i)}, y^{(i)}))}{\partial \boldsymbol{f}(\boldsymbol{x}^{(i)})} \frac{\partial \boldsymbol{f}(\boldsymbol{x}^{(i)})}{\partial b_k} \\
&= \frac{\partial \ell(\boldsymbol{f}(\boldsymbol{x}^{(i)}, y^{(i)}))}{\partial \boldsymbol{f}(\boldsymbol{x}^{(i)})} \frac{\partial \boldsymbol{V} \boldsymbol{\phi}(\boldsymbol{x}^{(i)}) + \boldsymbol{b}}{\partial b_k} \\
&= (\boldsymbol{\sigma}_{\mathrm{SM}}(\boldsymbol{f}(\boldsymbol{x}^{(i)})) - \boldsymbol{e}^{(y^{(i)})})^\top \boldsymbol{e}^{(k)} \\
&= \boldsymbol{\sigma}_{\mathrm{SM}}(\boldsymbol{f}(\boldsymbol{x}^{(i)}))_k - \mathbb{1}[k = y^{(i)}] \\
\frac{\partial \ell(\boldsymbol{f}(\boldsymbol{x}^{(i)}, y^{(i)}))}{\partial V_{k,l}} &= \frac{\partial \ell(\boldsymbol{f}(\boldsymbol{x}^{(i)}, y^{(i)}))}{\partial \boldsymbol{f}(\boldsymbol{x}^{(i)})} \frac{\partial \boldsymbol{V} \boldsymbol{\phi}(\boldsymbol{x}^{(i)}) + \boldsymbol{b}}{\partial V_{k,l}} \\
&= (\boldsymbol{\sigma}_{\mathrm{SM}}(\boldsymbol{f}(\boldsymbol{x}^{(i)})) - \boldsymbol{e}^{(y^{(i)})})^\top \boldsymbol{\phi}(\boldsymbol{x}^{(i)})_l \boldsymbol{e}^{(k)} \\
&= \boldsymbol{\phi}(\boldsymbol{x}^{(i)})_l (\boldsymbol{\sigma}_{\mathrm{SM}}(\boldsymbol{f}(\boldsymbol{x}^{(i)}))_k - \mathbb{1}[k = y^{(i)}])
\end{aligned}
$$

The one-epoch updates of the bias and the weight matrix with the sample-wise training are given by:

$$\Delta^S b_k = -\frac{\eta}{N} \sum_{i=1}^{N} \operatorname{sign}\left( \frac{\partial \ell(\boldsymbol{f}(\boldsymbol{x}^{(i)}, y^{(i)}))}{\partial b_k} \right)$$

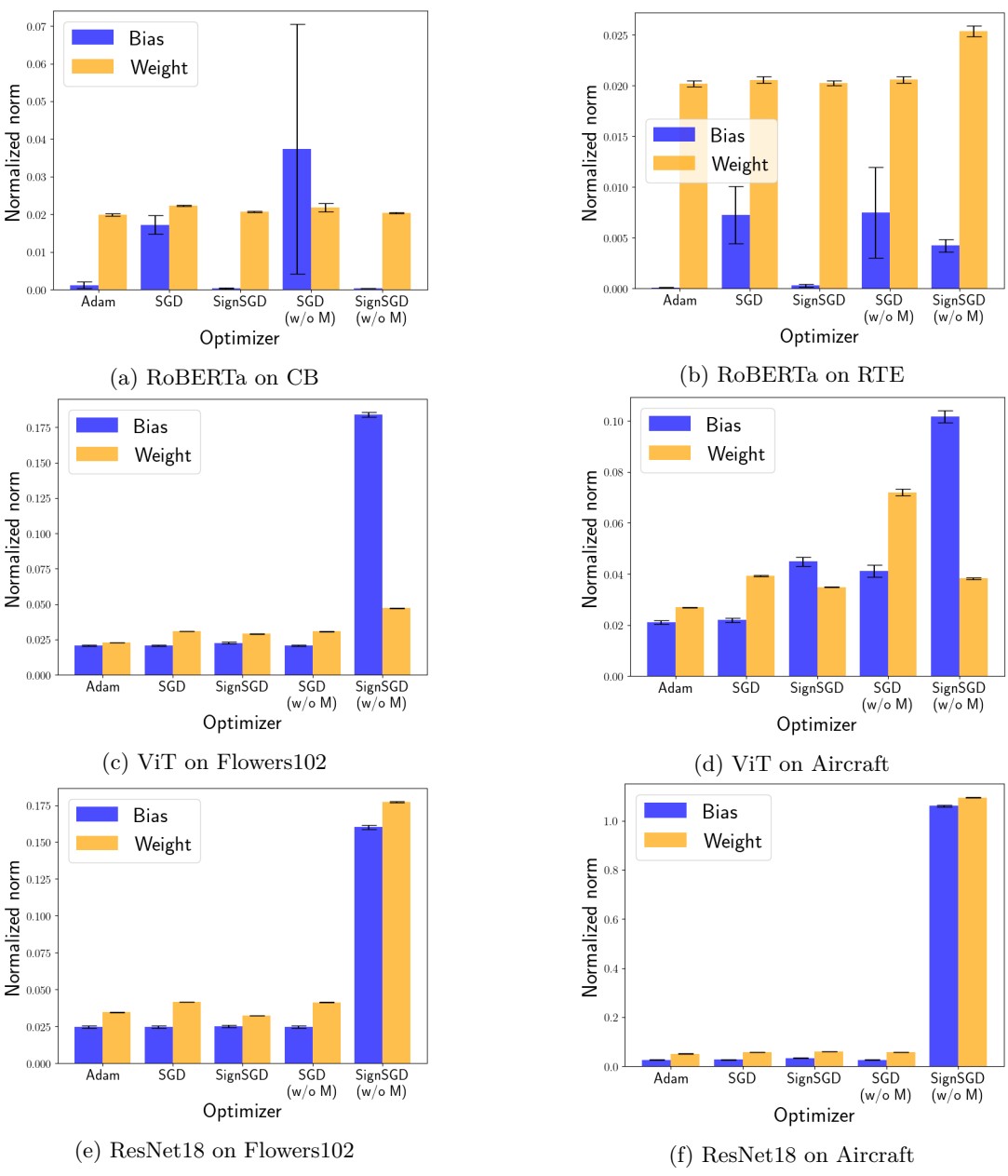

(a) RoBERTa on CB

(b) RoBERTa on RTE

(c) ViT on Flowers102

(d) ViT on Aircraft

(e) ResNet18 on Flowers102

(f) ResNet18 on Aircraft

Figure S.10: Norm of the linear head.

$$= -\frac{\eta}{N} \sum_{i=1}^{N} \text{sign}\left(\boldsymbol{\sigma}_{\text{SM}}(\boldsymbol{f}(\boldsymbol{x}^{(i)}))_k - \mathbb{1}[k = y^{(i)}]\right)$$

$$= -\frac{\eta}{N} \sum_{i=1}^{N} (1 - 2 \cdot \mathbb{1}[y^{(i)} = k])$$

and

$$\Delta^{\text{S}} V_{k,l} = -\frac{\eta}{N} \sum_{i=1}^{N} \text{sign}\left(\frac{\partial \ell(\boldsymbol{f}(\boldsymbol{x}^{(i)}, y^{(i)}))}{\partial V_{k,l}}\right)$$

$$= -\frac{\eta}{N} \sum_{i=1}^{N} \text{sign}\left(\boldsymbol{\phi}(\boldsymbol{x}^{(i)})_l (\boldsymbol{\sigma}_{\text{SM}}(\boldsymbol{f}(\boldsymbol{x}^{(i)}))_k - \mathbb{1}[k = y^{(i)}])\right)$$

$$= -\frac{\eta}{N} \sum_{i=1}^{N} \text{sign}\left(\boldsymbol{\phi}(\boldsymbol{x}^{(i)})_l\right) \text{sign}\left(\boldsymbol{\sigma}_{\text{SM}}(\boldsymbol{f}(\boldsymbol{x}^{(i)}))_k - \mathbb{1}[k = y^{(i)}]\right)$$

$$= -\frac{\eta}{N} \left(\sum_{y^{(i)} \neq k} \text{sign}\left(\boldsymbol{\phi}(\boldsymbol{x}^{(i)})_l\right) - \sum_{y^{(i)} = k} \text{sign}\left(\boldsymbol{\phi}(\boldsymbol{x}^{(i)})_l\right)\right)$$

The one-epoch updates of the bias and the weight matrix with the full-batch training are given by:

$$\Delta^{\text{F}} b_k = -\eta \, \text{sign}\left(\frac{1}{N} \sum_{i=1}^{N} \frac{\partial \ell(\boldsymbol{f}(\boldsymbol{x}^{(i)}, y^{(i)}))}{\partial b_k}\right)$$

$$= -\eta \, \text{sign}\left(\frac{1}{N} \sum_{i=1}^{N} \left(\boldsymbol{\sigma}_{\text{SM}}(\boldsymbol{f}(\boldsymbol{x}^{(i)}))_k - \mathbb{1}[k = y^{(i)}]\right)\right)$$

$$= -\eta \, \text{sign}\left(\sum_{i=1}^{N} \delta_{p_k}^{(i)}\right)$$

and

$$\Delta^{\text{F}} V_{k,l} = -\eta \, \text{sign}\left(\frac{1}{N} \sum_{i=1}^{N} \frac{\partial \ell(\boldsymbol{f}(\boldsymbol{x}^{(i)}, y^{(i)}))}{\partial V_{k,l}}\right)$$

$$= -\eta \, \text{sign}\left(\frac{1}{N} \sum_{i=1}^{N} \boldsymbol{\phi}(\boldsymbol{x}^{(i)})_l (\boldsymbol{\sigma}_{\text{SM}}(\boldsymbol{f}(\boldsymbol{x}^{(i)}))_k - \mathbb{1}[k = y^{(i)}])\right)$$

$$= -\eta \, \text{sign}\left(\sum_{i=1}^{N} \boldsymbol{\phi}(\boldsymbol{x}^{(i)})_l \delta_{p_k}^{(i)}\right).$$

$\square$

## F   More discussion on Transformers

In this section, we provide additional discussion on the gradient heterogeneity in Transformers, focusing on the self-attention mechanism.

**Additional notation.** The $k$-th standard basis vector is denoted by $\boldsymbol{e}^{(k)}$ with $\boldsymbol{e}_l^{(k)} = \delta_{kl}$, where $\delta_{kl}$ is the Kronecker delta. Function $\text{vec}(\cdot)$ denotes row-wise vectorization. Frobenius norm and the Kronecker product is denoted by $\|\cdot\|_F$ and $\otimes$, respectively.

### F.1   Transformer architecture

The Transformer architecture (Vaswani, 2017) relies on the self-attention mechanism, which assigns importance to each token in the input sequence.

For an input sequence of $n$ tokens, each of dimension $d$, represented by $\boldsymbol{X} \in \mathbb{R}^{n \times d}$, single-head self-attention is defined as:

$$\text{SA}(\boldsymbol{X}) \coloneqq \boldsymbol{\sigma}_{\text{SM}} \left( \frac{\boldsymbol{X}\boldsymbol{W}_Q (\boldsymbol{X}\boldsymbol{W}_K)^\top}{\sqrt{d_k}} \right) \boldsymbol{X}\boldsymbol{W}_V,$$

where $\boldsymbol{W}_Q, \boldsymbol{W}_K \in \mathbb{R}^{d \times d_k}$ and $\boldsymbol{W}_V \in \mathbb{R}^{d \times d_v}$ are learnable projection matrices for queries, keys, and values, respectively. Multi-head attention concatenates the outputs of parallel single-head self-attention mechanisms and applies a linear transformation, followed by a feed-forward network.

### F.2   Gradient of self-attention mechanism

We analyze the gradients in self-attention, focusing on the value and query/key weight matrices. Using Lemma A.2 from Noci et al. (2022), the Frobenius norms of these gradients are:

$$\begin{aligned}
\|\frac{\partial \, \text{SA}(\boldsymbol{X})}{\partial \boldsymbol{W}_V}\|_F &= \|\boldsymbol{P}\boldsymbol{X} \otimes \boldsymbol{I}_{d_v}\|_F \\
&\leq \underbrace{\sqrt{d_v}\|\boldsymbol{P}\|_F\|\boldsymbol{X}\|_F}_{=:\mathcal{U}_V},
\end{aligned} \tag{22}$$

$$\begin{aligned}
&\|\frac{\partial \, \text{SA}(\boldsymbol{X})}{\partial \boldsymbol{W}_Q}\|_F \\
=&\|(\boldsymbol{I}_n \otimes \boldsymbol{W}_V \boldsymbol{X}^\top) \frac{\partial \boldsymbol{P}}{\partial \boldsymbol{M}} \frac{\boldsymbol{X} \otimes \boldsymbol{X}\boldsymbol{W}_K}{\sqrt{d_k}}\|_F \\
\leq&\underbrace{\sqrt{n}\|\boldsymbol{W}_V \boldsymbol{X}^\top\|_F\|\frac{\partial \boldsymbol{P}}{\partial \boldsymbol{M}}\|_F \frac{\|\boldsymbol{X}\|_F\|\boldsymbol{X}\boldsymbol{W}_K\|_F}{\sqrt{d_k}}}_{=:\mathcal{U}_Q},
\end{aligned} \tag{23}$$

where $\boldsymbol{M} \coloneqq \boldsymbol{X}\boldsymbol{W}_Q\boldsymbol{W}_K^\top\boldsymbol{X}^\top / \sqrt{d_k}$, $\boldsymbol{P} \coloneqq \boldsymbol{\sigma}_{\text{SM}}(\boldsymbol{M})$, and $\mathcal{U}_V$ and $\mathcal{U}_Q$ represent the upper bounds for the gradients of the value and query weight matrices, respectively. The derivation of the gradient for the key weight matrix is omitted, as it is analogous to that of the query weight matrix.

Focusing on the attention matrix $\boldsymbol{P}$, we derive the following result.

**Proposition F.1** (Gradients and attention matrices). *In Transformers, one-hot attention matrices uniquely maximize the upper bound of the Frobenius norm of the gradient with respect to the value weight matrix $\mathcal{U}_V$ and uniquely minimize that with respect to the query weight matrix $\mathcal{U}_Q$, as follows:*

$$\arg\max_{\boldsymbol{P}} \mathcal{U}_V = \arg\min_{\boldsymbol{P}} \mathcal{U}_Q = \mathcal{P}_{one\text{-}hot},$$

*where*

$$\mathcal{P}_{one\text{-}hot} \coloneqq \{\boldsymbol{P} \mid \forall i, \ \exists k_i \ s.t. \ \boldsymbol{P}_{i,:} = \boldsymbol{e}^{(k_i)}\}$$

*is the set of one-hot matrices.*

The proof of the proposition is provided in Appendix F.4. The statement about the query weight matrix also applies to the key weight matrix due to their analogous gradients. The proposition demonstrates that the gradients of the value and query/key weight matrices exhibit opposing behaviors with respect to one-hot attention matrices: the gradient of the value weight matrix is maximized, while those of the query/key weight matrices are minimized.

Previous studies (Noci et al., 2022; Wang et al., 2021) observed that the gradient of the value weight matrix is typically larger than those of the query/key weight matrices, consistent with our experimental findings in Section 5.2. Together with Proposition F.1, these results suggest that attention matrices close to one-hot amplify gradient heterogeneity in the self-attention mechanism.

### F.3 Uniformity of the attention matrix

In Figure S.11, we compare the attention matrices of pre-trained RoBERTa and ViT. The attention matrix of ViT is more uniform than that of RoBERTa, reflecting the differences between NLP and vision tasks. In NLP, the use of special tokens and stronger interrelations between input tokens lead to less uniform attention, with only a few tokens receiving attention (Clark, 2019). Conversely, vision tasks, which prioritize holistic information (Torralba, 2003; Rabinovich et al., 2007; Shotton et al., 2009), produce more uniform attention matrices, where all tokens are attended to. This observation aligns with Hyeon-Woo et al. (2023), who also reported uniform attention matrices in ViT. Notably, more uniform attention matrices are farther from one-hot matrices, indicating reduced dominance by individual tokens.

Combined with the analysis in Appendix F.2, which shows that attention matrices closer to one-hot matrices amplify gradient heterogeneity, this suggests that gradient heterogeneity in the self-attention mechanism is more pronounced in NLP tasks than in vision tasks.

### F.4 Proof of Proposition F.1

*Proof of $\mathcal{U}_V$.* As defined in Eq.(22), the upper bound of the gradient is given by:

$$\mathcal{U}_V = \sqrt{d_v}\|\boldsymbol{P}\|_F\|\boldsymbol{X}\|_F.$$

We observe that:

$$\begin{aligned}
\arg\max_{\boldsymbol{P}} \mathcal{U}_V &= \arg\max_{\boldsymbol{P}} \|\boldsymbol{P}\|_F \\
&= \arg\max_{\boldsymbol{P}} \|\boldsymbol{P}\|_F^2 \\
&= \arg\max_{\boldsymbol{P}} \sum_{i=1}^{n} \|\boldsymbol{P}_{i,:}\|_2^2.
\end{aligned}$$

Since the rows of the attention matrix are independent, we focus on the $i$-th row. The $i$-th row of the attention matrix satisfies the following constraints:

$$1 \le j \le n, \quad P_{i,j} \ge 0, \quad \sum_{j=1}^{n} P_{i,j} = 1.$$

We define the Lagrangian function as:

$$\mathcal{L}_V = -\sum_{j=1}^{n} P_{i,j}^2 - \sum_{j=1}^{n} \mu_j P_{i,j} + \lambda(\sum_{j=1}^{n} P_{i,j} - 1),$$

where $\lambda$ and $\mu_j$ are the Lagrange multipliers. To minimize the Lagrangian function, the solution must satisfy the following KKT conditions:

$$\frac{\partial \mathcal{L}_V}{\partial P_{i,j}} = -2P_{i,j} - \mu_j + \lambda = 0, \quad 1 \le j \le n, \tag{24}$$

$$\sum_{j=1}^{n} P_{i,j} - 1 = 0, \tag{25}$$

$$P_{i,j} \geq 0, \quad 1 \leq j \leq n, \tag{26}$$

$$\mu_j \geq 0, \quad 1 \leq j \leq n, \tag{27}$$

$$\mu_j P_{i,j} = 0, \quad 1 \leq j \leq n. \tag{28}$$

From Equations (25) and (26), it follows that $P_{i,j} > 0$ for some $j$. Let $k$ $(1 \leq k \leq n)$ denote the number of non-zero elements in $\boldsymbol{P}_{i,:}$, and suppose $P_{i,j_l} > 0$ for $1 \leq l \leq k$. From Equation (28), we have $\mu_{j_l} = 0$, and thus, from Equation (24), we deduce that $P_{i,j_l} = \frac{\lambda}{2}$ for $1 \leq l \leq k$. Using Equation (25), we get $\sum_{l=1}^{k} \frac{\lambda}{2} = 1$, which gives $\lambda = 2/k$. For $j \notin \{j_l \mid 1 \leq l \leq k\}$, we have $P_{i,j} = 0$ and $\mu_j = \lambda = 2/k$, satisfying Eq.(27).

With $k$ non-zero elements of $\boldsymbol{P}_{i,:}$, the value of the Lagrangian function becomes $-\sum_{j=1}^{n} P_{i,j}^2 = -\sum_{l=1}^{k} (\frac{\lambda}{2})^2 = -\frac{\lambda^2}{4} k = -\frac{1}{k}$. The minimum value of the Lagrangian function is achieved if and only if $k = 1$, which implies $\boldsymbol{P}_{i,:} = \boldsymbol{e}^{(k_i)}$ for some $k_i$. Therefore, we conclude:

$$\arg\max_{\boldsymbol{P}} \mathcal{U}_V = \{\boldsymbol{P} \mid \forall i, \ \exists k_i \ s.t. \ \boldsymbol{P}_{i,:} = \boldsymbol{e}^{(k_i)}\}.$$

$\square$

*Proof of $\mathcal{U}_Q$.* As defined in Eq.(23), the upper bound of the gradient is given by:

$$\mathcal{U}_Q = \sqrt{n} \|\boldsymbol{W}_V \boldsymbol{X}^\top\|_F \|\frac{\partial \boldsymbol{P}}{\partial \boldsymbol{M}}\|_F \frac{\|\boldsymbol{X}\|_F \|\boldsymbol{X}\boldsymbol{W}_K\|_F}{\sqrt{d_k}}.$$

The partial derivative is expressed as:

$$\frac{\partial \boldsymbol{P}}{\partial \boldsymbol{M}} = \frac{\partial \boldsymbol{\sigma}_{\text{SM}}(\boldsymbol{M})}{\partial \boldsymbol{M}}$$

$$= \text{blockdiag}(\{\frac{\partial \boldsymbol{\sigma}_{\text{SM}}(\boldsymbol{M}_{i,:})}{\partial \boldsymbol{M}_{i,:}}\}_{i=1}^{n})$$

$$= \text{blockdiag}(\{\text{diag}(\boldsymbol{P}_{i,:}) - \boldsymbol{P}_{i,:}\boldsymbol{P}_{i,:}^\top\}_{i=1}^{n}).$$

Considering the attention matrix $\boldsymbol{P}$, we obtain:

$$\arg\min_{\boldsymbol{P}} \mathcal{U}_Q = \arg\min_{\boldsymbol{P}} \|\frac{\partial \boldsymbol{P}}{\partial \boldsymbol{M}}\|_F$$

$$= \arg\min_{\boldsymbol{P}} \sum_{i=1}^{n} \|\text{diag}(\boldsymbol{P}_{i,:}) - \boldsymbol{P}_{i,:}\boldsymbol{P}_{i,:}^\top\|_F^2.$$

As in the proof of $\mathcal{U}_V$, we focus on the value of the $i$-th row:

$$\|\text{diag}(\boldsymbol{P}_{i,:}) - \boldsymbol{P}_{i,:}\boldsymbol{P}_{i,:}^\top\|_F^2 = \sum_{j=1}^{n} (P_{i,j} - P_{i,j}^2)^2 + \sum_{j \neq l} P_{i,j}^2 P_{i,l}^2,$$

subject to the constraints $1 \leq j \leq n$, $P_{i,j} \geq 0$, $\sum_{j=1}^{n} P_{i,j} = 1$. Since both the first term and the second term are non-negative, the minimum value is attained if and only if both terms are 0. This condition is satisfied if $\boldsymbol{P}_{i,:}$ is a one-hot vector. Conversely, if $\boldsymbol{P}_{i,:}$ is not a one-hot vector, the second term becomes positive, and the minimum value cannot be attained. Thus, we have shown that the minimum value of the objective function is achieved if and only if $\boldsymbol{P}_{i,:}$ is a one-hot vector. Therefore:

$$\arg\min_{\boldsymbol{P}} \mathcal{U}_Q = \{\boldsymbol{P} \mid \forall i, \ \exists k_i \ s.t. \ \boldsymbol{P}_{i,:} = \boldsymbol{e}^{(k_i)}\}.$$

$\square$

### F.5 Experimental results

**Heatmap of attention matrices.** In Figure S.11, we show the attention matrices computed from pre-trained models. These matrices are calculated for a randomly sampled sequ ence from the training data and are averaged across all heads.

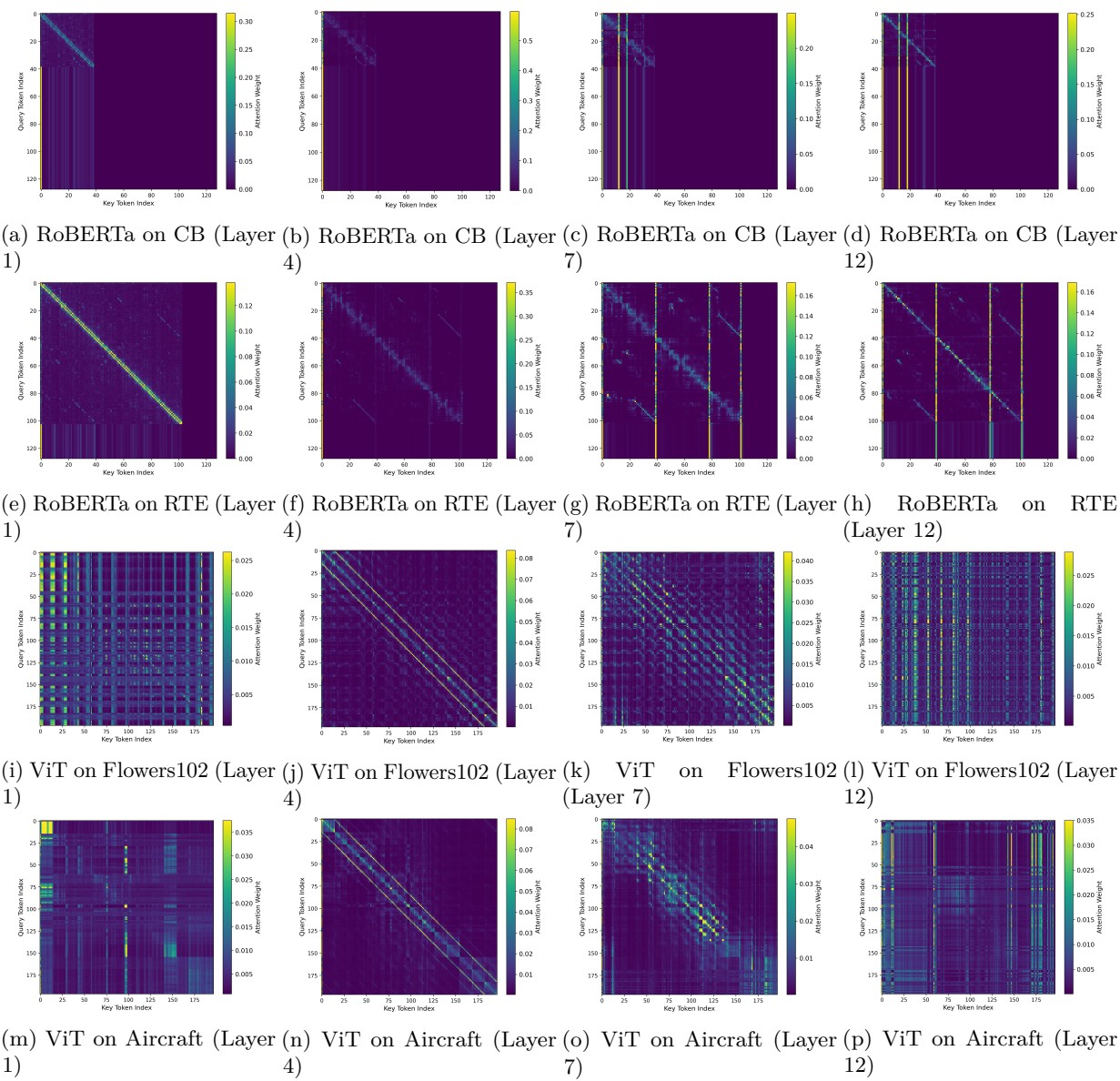

(a) RoBERTa on CB (Layer 1)  (b) RoBERTa on CB (Layer 4)  (c) RoBERTa on CB (Layer 7)  (d) RoBERTa on CB (Layer 12)

(e) RoBERTa on RTE (Layer 1)  (f) RoBERTa on RTE (Layer 4)  (g) RoBERTa on RTE (Layer 7)  (h) RoBERTa on RTE (Layer 12)

(i) ViT on Flowers102 (Layer 1)  (j) ViT on Flowers102 (Layer 4)  (k) ViT on Flowers102 (Layer 7)  (l) ViT on Flowers102 (Layer 12)

(m) ViT on Aircraft (Layer 1)  (n) ViT on Aircraft (Layer 4)  (o) ViT on Aircraft (Layer 7)  (p) ViT on Aircraft (Layer 12)

Figure S.11: Attention matrices of the pre-trained RoBERTa and ViT.

**Gradient and entropy of attention matrices.** In Figure S.12 (a) and (c), we show the ratio of the mean entropy relative to the maximum entropy of the attention matrix for each layer of the Transformer model. Error bars indicate the standard deviation. Specifically, we plot:

$$\frac{1}{HNS} \sum_{h=1}^{H} \sum_{i=1}^{N} \sum_{s=1}^{S} \left( \sum_{j=1}^{S} A_{s,j}^{(i,h,l)} \log(A_{s,j}^{(i,h,l)}) / \log(S) \right),$$

for each layer $l$, where $H$ is the number of heads, $S$ is the sequence length, and $\boldsymbol{A}^{(i,h,l)} \in \mathbb{R}^{S \times S}$ is the attention matrix of the $h$-th head in the $l$-th layer for sample $\boldsymbol{x}^{(i)}$.

In Figure S.12 (b) and (d), we show the ratio of the mean gradient norm relative to the sum of the gradient norms of the attention matrix for each layer. Specifically, we plot:

$$\frac{G_p^{(l)}}{G_Q^{(l)} + G_K^{(l)} + G_V^{(l)}},$$

for each layer $l$ and $p \in \{Q, K, V\}$, where $G_Q^{(l)}$, $G_K^{(l)}$, and $G_V^{(l)}$ are the full-batch gradient norms of the query, key, and value weight matrices in the $l$-th layer of the Transformer model, respectively.

The results show that the entropy of the attention matrix is higher in RoBERTa than in ViT, and the gradient norm of the attention matrix is more heterogeneous in RoBERTa than in ViT. This observation is consistent with the theoretical analysis in Appendix F.3.

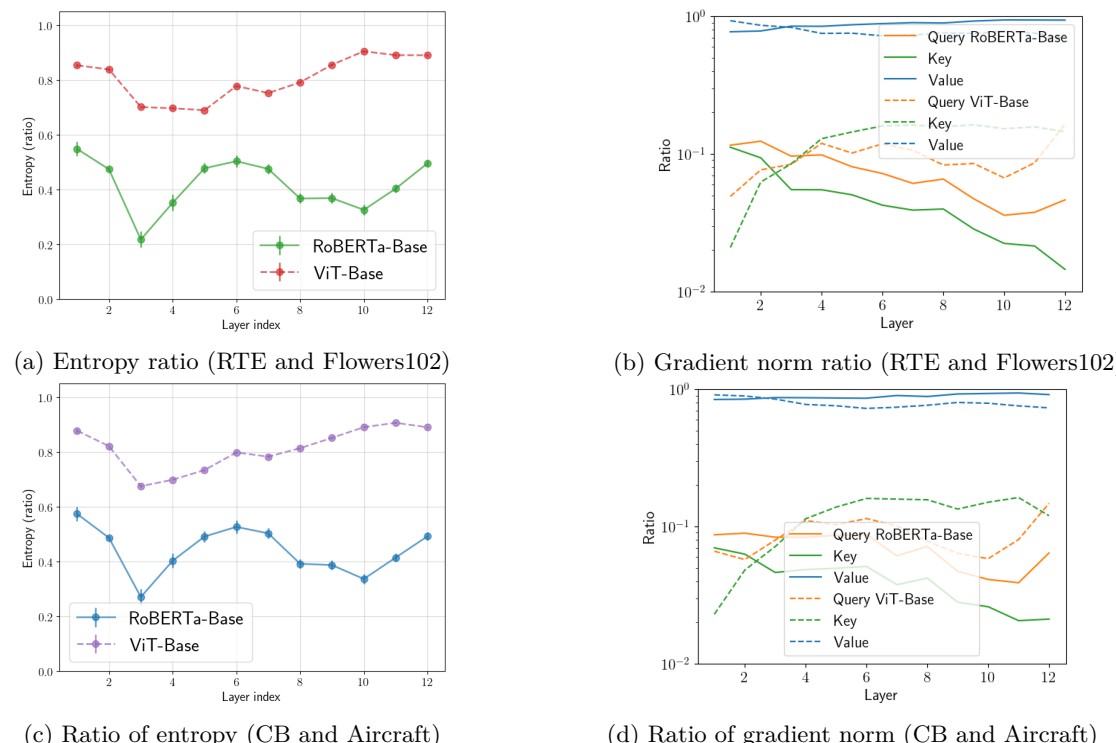

(a) Entropy ratio (RTE and Flowers102)  (b) Gradient norm ratio (RTE and Flowers102)

(c) Ratio of entropy (CB and Aircraft)  (d) Ratio of gradient norm (CB and Aircraft)

Figure S.12: Comparison of entropy and gradient norms in attention matrices for RoBERTa and ViT. (a) and (c): the ratio of entropy relative to the maximum possible entropy. (b) and (d): the ratio of the gradient norm for self-attention parameters relative to the total gradient norm.

# G  Iteration complexity of SignSGD under $l_2$ norm

If we evaluate both SGD and SignSGD using the $l_2$ norm, we obtain the following upper bound for SignSGD as a direct consequence of Theorem 4.6:

$$\mathcal{T}_\varepsilon(\{\boldsymbol{\theta}_t^{\text{Sign}}\}_{t=0}^\infty, L, \|\cdot\|_2) \leq \frac{6(L(\boldsymbol{\theta}_0) - L_*)}{\varepsilon^2 \zeta_0} \Lambda_P.$$

Compared with the original bound under the $l_1$ norm, this bound is larger than that of SGD by a factor of $P$ because SGD and SignSGD are steepest descent methods with respect to the $l_2$ and $l_\infty$ norms, respectively (as discussed in Section 3.2). Thus, evaluating both methods under the $l_2$ norm favors SGD, which is why we evaluate SignSGD under its natural metric ($l_1$ norm).

*Proof.* From Eq. (11),

$$L(\boldsymbol{\theta}_T^{\text{Sign}}) - L(\boldsymbol{\theta}_0) \leq -\frac{\zeta_0}{6} \sum_{t=0}^{T-1} \min\left(\frac{\|\nabla L(\boldsymbol{\theta}_t^{\text{Sign}})\|_1}{P\Lambda_P}, \sqrt{\frac{\|\nabla L(\boldsymbol{\theta}_t^{\text{Sign}})\|_1}{\rho_H P^{3/2}}}\right) \|\nabla L(\boldsymbol{\theta}_t^{\text{Sign}})\|_1$$

$$\leq -\frac{\zeta_0}{6} \sum_{t=0}^{T-1} \min\left(\frac{\|\nabla L(\boldsymbol{\theta}_t^{\text{Sign}})\|_2}{P\Lambda_P}, \sqrt{\frac{\|\nabla L(\boldsymbol{\theta}_t^{\text{Sign}})\|_2}{\rho_H P^{3/2}}}\right) \|\nabla L(\boldsymbol{\theta}_t^{\text{Sign}})\|_2$$

Assume that $\|\nabla L(\boldsymbol{\theta}_t^{\text{Sign}})\|_2 \geq \sqrt{P}\varepsilon$ holds for all $0 \leq t < T$. Then, we have

$$L(\boldsymbol{\theta}_T^{\text{Sign}}) - L(\boldsymbol{\theta}_0) \leq -\frac{T\varepsilon\zeta_0}{6} \min\left(\frac{\varepsilon}{\Lambda_P}, \sqrt{\frac{\varepsilon}{\rho_H}}\right)$$

$$= -\frac{T\varepsilon^2\zeta_0}{6\Lambda_P} \quad \left(\text{From } \varepsilon < \frac{\Lambda_P^2}{\rho_H\sqrt{P}} < \frac{\Lambda_P^2}{\rho_H}\right).$$

Therefore, we have:

$$T \leq \frac{6(L(\boldsymbol{\theta}_0) - L(\boldsymbol{\theta}_T^{\text{Sign}}))}{\varepsilon^2 \zeta_0} \Lambda_P$$

$$\leq \frac{6(L(\boldsymbol{\theta}_0) - L_*)}{\varepsilon^2 \zeta_0} \Lambda_P.$$

This means:

$$\mathcal{T}_\varepsilon(\{\boldsymbol{\theta}_t^{\text{Sign}}\}_{t=0}^\infty, L, \|\cdot\|_2) \leq \frac{6(L(\boldsymbol{\theta}_0) - L_*)}{\varepsilon^2 \zeta_0} \Lambda_P.$$

$\square$

# H    Block-wise normalized gradient descent

Block-wise normalization is a common technique for stabilizing SGD in heterogeneous models. Adam-mini reduces optimizer state by sharing learning rates within Hessian-informed blocks (Zhang et al., 2024b), and Zhao et al. (2025) study layer-wise learning-rate adaptation for sign-based optimizers. More recently, column-wise gradient normalization has been shown to substantially narrow the Adam–SGD gap in LLM pretraining (Glentis et al., 2025). This appendix extends our deterministic analysis to the abstract block-normalized update in Eq. (30) and explains why normalization reduces sensitivity to gradient heterogeneity.

**Block-normalized update.**    For $\boldsymbol{\theta} \in \mathcal{R}_{\mathrm{FT}}$ with $\nabla L(\boldsymbol{\theta}) \neq \mathbf{0}$, define the block-normalized gradient $\mathcal{N}(\nabla L(\boldsymbol{\theta}))$ block-wise by

$$[\mathcal{N}(\nabla L(\boldsymbol{\theta}))]_b := \begin{cases} \dfrac{[\nabla L(\boldsymbol{\theta})]_b}{\|[\nabla L(\boldsymbol{\theta})]_b\|_2} & \text{if } \|[\nabla L(\boldsymbol{\theta})]_b\|_2 > 0, \\ \mathbf{0} & \text{otherwise,} \end{cases} \qquad b = 1, \ldots, B. \tag{29}$$

The *block-normalized sequence* is

$$\boldsymbol{\theta}_{t+1}^{\mathrm{Norm}} = \boldsymbol{\theta}_t^{\mathrm{Norm}} - \eta_t \mathcal{N}(\nabla L(\boldsymbol{\theta}_t^{\mathrm{Norm}})). \tag{30}$$

When each block corresponds to a single parameter coordinate ($B = P$), Eq. (30) reduces to SignSGD. When a weight matrix is further partitioned into columns (or rows), Eq. (29) recovers column-wise (or row-wise) gradient normalization as in SCALE (Glentis et al., 2025). Methods such as Adam-mini and layer-wise sign adaptation use coarser or alternative block partitions (Zhang et al., 2024b; Zhao et al., 2025); our result applies to any fixed partition once $\Lambda_{\mathrm{BN}}$ and $\delta_D$ are defined with respect to that partition.

**Complexity measure.**    Block normalization equalizes the update magnitude across blocks, so the natural stationarity criterion is the average block gradient norm

$$\|\nabla L(\boldsymbol{\theta})\|_{\mathrm{blk}} := \frac{1}{B} \sum_{b=1}^{B} \|[\nabla L(\boldsymbol{\theta})]_b\|_2.$$

This norm coincides with the dual norm of the block-wise update direction in Eq. (30).    We write $\mathcal{T}_\varepsilon(\{\boldsymbol{\theta}_t\}, L, \|\cdot\|_{\mathrm{blk}})$ for the iteration complexity with stopping condition $\|\nabla L(\boldsymbol{\theta}_t)\|_{\mathrm{blk}} \leq \varepsilon$.

**Block-uniform curvature measure.**    Because $\|[\mathcal{N}(\nabla L(\boldsymbol{\theta}))]_b\|_2 \in \{0, 1\}$, each active block contributes equally to the local curvature term, independent of $\|[\nabla L(\boldsymbol{\theta})]_b\|_2$. We therefore define

$$\Lambda_{\mathrm{BN}} := \sup_{\boldsymbol{\theta} \in \mathcal{R}_{\mathrm{FT}}} \sum_{b=1}^{B} \|[\nabla^2 L(\boldsymbol{\theta})]_b\|_2.$$

Unlike $\Lambda_G$, $\Lambda_{\mathrm{BN}}$ does not weight Hessian blocks by squared gradient norms and is therefore insensitive to gradient heterogeneity across blocks. When all blocks have comparable dimension, $\Lambda_{\mathrm{BN}}$ is on the order of $B\Lambda_P$, whereas $\Lambda_G$ can be much larger under gradient–Hessian correlation (Section 4.3).

**Theorem H.1** (Deterministic block-normalized sequence). *Assume $\delta_D < \Lambda_{\mathrm{BN}}/3$. For the block-normalized sequence* (30), *suppose that $\varepsilon < \Lambda_{\mathrm{BN}}^2/(\rho_H \sqrt{B})$ and that*

$$\eta_t = \zeta_t \min\left( \frac{\sum_{b=1}^{B} \|[\nabla L(\boldsymbol{\theta}_t^{Norm})]_b\|_2}{\Lambda_{\mathrm{BN}}}, \sqrt{\frac{\sum_{b=1}^{B} \|[\nabla L(\boldsymbol{\theta}_t^{Norm})]_b\|_2}{\rho_H B^{3/2}}} \right), \qquad \zeta_t \in [\zeta_0, 1],$$

*we have*

$$\mathcal{T}_\varepsilon(\{\boldsymbol{\theta}_t^{Norm}\}_{t=0}^\infty, L, \|\cdot\|_{\mathrm{blk}}) \leq \frac{6(L(\boldsymbol{\theta}_0) - L_*)}{B\varepsilon^2 \zeta_0} \Lambda_{\mathrm{BN}}.$$

**Interpretation.** Theorem H.1 parallels Theorem 4.6: SGD is controlled by $\Lambda_G$, SignSGD by $\Lambda_P$, and block-normalized SGD by $\Lambda_{\mathrm{BN}}$. Since $\Lambda_{\mathrm{BN}}$ does not amplify blocks with disproportionately large gradients, block normalization can reduce iteration complexity relative to vanilla SGD when gradient heterogeneity is large, even though it does not introduce coordinate-wise adaptivity like Adam. This provides a theoretical explanation for why block-wise normalization methods (Zhang et al., 2024b; Zhao et al., 2025; Glentis et al., 2025) can close part of the Adam–SGD gap within our block-heterogeneity framework.

**Finer block partitions and the block-diagonal Hessian assumption.** Theorem H.1 relies on Assumption 4.3, which approximates the Hessian as block-diagonal with respect to the same partition used for normalization. This assumption is most natural when blocks correspond to coarse modules such as layers or attention/feed-forward submodules, as in our main analysis. When blocks are refined—for example, to matrix rows or columns as in column-wise normalization (Glentis et al., 2025)—coupling between sub-blocks inside a layer is treated as off-diagonal mass in the chosen partition. In general, refining the partition therefore tends to increase the approximation error $\delta_D$, even though finer normalization can better equalize gradient scales within a layer. Theorem H.1 remains valid whenever $\delta_D < \Lambda_{\mathrm{BN}}/3$, but the effective bound can deteriorate as $\delta_D$ grows. Quantifying the trade-off between finer normalization (smaller gradient-heterogeneity bias) and larger $\delta_D$ (weaker block-diagonal Hessian approximation) for a given partition is an interesting direction for future work.

*Proof.* Write $G_t := \sum_{b=1}^{B} \|[\nabla L(\boldsymbol{\theta}_t^{\mathrm{Norm}})]_b\|_2 = B\|\nabla L(\boldsymbol{\theta}_t^{\mathrm{Norm}})\|_{\mathrm{blk}}$ and $\Delta_t := \mathcal{N}(\nabla L(\boldsymbol{\theta}_t^{\mathrm{Norm}}))$. Using Lemma A.1 as in the sign-based proof (Appendix A.3),

$$
L(\boldsymbol{\theta}_{t+1}^{\mathrm{Norm}}) - L(\boldsymbol{\theta}_t^{\mathrm{Norm}})
$$

$$
\leq \nabla L(\boldsymbol{\theta}_t^{\mathrm{Norm}})^\top (\boldsymbol{\theta}_{t+1}^{\mathrm{Norm}} - \boldsymbol{\theta}_t^{\mathrm{Norm}}) + \frac{1}{2}(\boldsymbol{\theta}_{t+1}^{\mathrm{Norm}} - \boldsymbol{\theta}_t^{\mathrm{Norm}})^\top \nabla^2 L(\boldsymbol{\theta}_t^{\mathrm{Norm}})(\boldsymbol{\theta}_{t+1}^{\mathrm{Norm}} - \boldsymbol{\theta}_t^{\mathrm{Norm}}) + \frac{\rho_H}{6}\|\boldsymbol{\theta}_{t+1}^{\mathrm{Norm}} - \boldsymbol{\theta}_t^{\mathrm{Norm}}\|_2^3
$$

$$
= -G_t \eta_t + \frac{\eta_t^2}{2}\Delta_t^\top \nabla^2 L(\boldsymbol{\theta}_t^{\mathrm{Norm}})\Delta_t + \frac{\eta_t^3 \rho_H}{6}\|\Delta_t\|_2^3
$$

$$
= -G_t \eta_t + \frac{\eta_t^2}{2}\Delta_t^\top \nabla^2 L_D(\boldsymbol{\theta}_t^{\mathrm{Norm}})\Delta_t + \frac{\eta_t^2}{2}\Delta_t^\top(\nabla^2 L(\boldsymbol{\theta}_t^{\mathrm{Norm}}) - \nabla^2 L_D(\boldsymbol{\theta}_t^{\mathrm{Norm}}))\Delta_t + \frac{\eta_t^3 \rho_H}{6}\|\Delta_t\|_2^3
$$

$$
\leq -G_t \eta_t + \frac{\eta_t^2}{2}\sum_{b=1}^{B}\|[\nabla^2 L(\boldsymbol{\theta}_t^{\mathrm{Norm}})]_b\|_2 + \frac{\eta_t^2}{2}\delta_D\|\Delta_t\|_2^2 + \frac{\eta_t^3 \rho_H}{6}B^{3/2}
$$

$$
\leq -G_t \eta_t + \frac{\eta_t^2}{2}\Lambda_{\mathrm{BN}} + \frac{\eta_t^2}{2}\delta_D B + \frac{\eta_t^3 \rho_H}{6}B^{3/2},
$$

where we used $\|[\Delta_t]_b\|_2 \leq 1$ and $\|\Delta_t\|_2^2 \leq B$. With $\eta_t \leq \min(G_t/\Lambda_{\mathrm{BN}}, \sqrt{G_t/(\rho_H B^{3/2})})$ and $\delta_D < \Lambda_{\mathrm{BN}}/3$, we obtain

$$
L(\boldsymbol{\theta}_{t+1}^{\mathrm{Norm}}) - L(\boldsymbol{\theta}_t^{\mathrm{Norm}}) \leq -\frac{\eta_t}{6}G_t.
$$

Taking a telescoping sum and using $\eta_t \geq \zeta_0 \min(G_t/\Lambda_{\mathrm{BN}}, \sqrt{G_t/(\rho_H B^{3/2})})$ yields

$$
L(\boldsymbol{\theta}_T^{\mathrm{Norm}}) - L(\boldsymbol{\theta}_0) \leq -\frac{\zeta_0}{6}\sum_{t=0}^{T-1}\min\left(\frac{G_t^2}{\Lambda_{\mathrm{BN}}}, \sqrt{\frac{G_t^3}{\rho_H B^{3/2}}}\right).
$$

If $\|\nabla L(\boldsymbol{\theta}_t^{\mathrm{Norm}})\|_{\mathrm{blk}} \geq \varepsilon$ for all $0 \leq t < T$, then $G_t \geq B\varepsilon$ and, using $\varepsilon < \Lambda_{\mathrm{BN}}^2/(\rho_H \sqrt{B})$,

$$
L(\boldsymbol{\theta}_T^{\mathrm{Norm}}) - L(\boldsymbol{\theta}_0) \leq -\frac{TB^2\varepsilon^2\zeta_0}{6\Lambda_{\mathrm{BN}}}.
$$

Therefore,

$$
T \leq \frac{6(L(\boldsymbol{\theta}_0) - L(\boldsymbol{\theta}_T^{\mathrm{Norm}}))}{B\varepsilon^2\zeta_0}\Lambda_{\mathrm{BN}} \leq \frac{6(L(\boldsymbol{\theta}_0) - L_*)}{B\varepsilon^2\zeta_0}\Lambda_{\mathrm{BN}},
$$

which proves the claim. □

# I Additional background and related work

## I.1 Steepest descent

Beyond the descent direction, steepest descent can be formulated as a full update obtained by minimizing a local smoothness-based upper bound of the objective (Bernstein & Newhouse, 2024).

We say that $L$ is $L_p$-smooth if its gradient is Lipschitz continuous with respect to the $\ell_p$ norm, that is,

$$\|\nabla L(\boldsymbol{\theta}) - \nabla L(\boldsymbol{\theta}')\|_q \leq L_p \|\boldsymbol{\theta} - \boldsymbol{\theta}'\|_p,$$

where $\|\cdot\|_q$ is the dual norm of $\|\cdot\|_p$ For an $L_p$-smooth function in a deterministic setting, the steepest descent update is given by

$$\boldsymbol{\theta}_{t+1} \in \operatorname*{arg\,min}_{\boldsymbol{\theta} \in \mathbb{R}^P} \left( \langle \nabla L(\boldsymbol{\theta}_t), \boldsymbol{\theta} - \boldsymbol{\theta}_t \rangle + \frac{L_p}{2} \|\boldsymbol{\theta} - \boldsymbol{\theta}_t\|_p^2 \right).$$

For the $\ell_2$ norm, this reduces to the standard gradient descent update

$$\boldsymbol{\theta}_{t+1} = \boldsymbol{\theta}_t - \frac{1}{L_2} \nabla L(\boldsymbol{\theta}_t),$$

whereas for the $\ell_\infty$ norm, a closed-form solution is given by

$$\boldsymbol{\theta}_{t+1} = \boldsymbol{\theta}_t - \frac{\|\nabla L(\boldsymbol{\theta}_t)\|_1}{L_\infty} \operatorname{sign}(\nabla L(\boldsymbol{\theta}_t)),$$

which corresponds to a scaled sign-based update.

## I.2 Extended related work

**Transformer architecture and layer normalization.** The original Transformer architecture (Vaswani, 2017), referred to as Post-LN, applies layer normalization after the residual connection. In contrast, the Pre-LN architecture places layer normalization before the residual connection. Wang et al. (2019b) demonstrated that Post-LN Transformers are difficult to train when the number of layers is large, a finding later theoretically confirmed by Xiong et al. (2020) using mean field theory. Other architectures such as Reformer (He et al., 2021) were also introduced. Shi et al. (2022) showed that a large standard deviation in layer normalization leads to rank collapse in Post-LN Transformers. Furthermore, Wu et al. (2024) observed that sparse masked attention mitigates rank collapse in the absence of layer normalization and that layer normalization induces equilibria ranging from rank one to full rank.

**Attention sparsity.** Sparse attention mechanisms have been proposed to reduce the computational costs of Transformers. For example, ETC (Ainslie et al., 2020) introduces efficient sparse attention, and Zaheer et al. (2020) proposed BigBird, which they theoretically demonstrated to be as expressive as full attention. These sparse attention mechanisms are widely used in language models with large context windows, such as Longformer (Beltagy et al., 2020) and Mistral 7B (Jiang et al., 2023). In NLP, Clark (2019) found that attention of pre-trained BERT focuses on specific tokens. In vision, Hyeon-Woo et al. (2023) showed that while uniform attention is challenging to learn with the softmax function, ViT successfully learns uniform attention, which is key to its success. Additionally, Zhai et al. (2023) suggested that low attention entropy contributes to training instability in Transformers, a phenomenon they termed *entropy collapse*. Furthermore, Bao et al. (2024) demonstrated that a small eigenspectrum variance of query and key matrices leads to localized attention and mitigates both rank and entropy collapse.

# J Notation

Table S.14 shows our notations.

Table S.14: Table of notations.

| Variable | Definition |
|---|---|
| $a_k$ | $k$-th element of vector $\boldsymbol{a}$ |
| $\boldsymbol{A}_{k,:}, \boldsymbol{A}_{:,j}, A_{k,j}$ | $k$-th row, $j$-th column, and $(k,j)$-th element of matrix $\boldsymbol{A}$ |
| $[\boldsymbol{A}]_b, [\boldsymbol{a}]_b$ | $b$-th block of matrix $\boldsymbol{A}$ and vector $\boldsymbol{a}$ |
| $B$ | number of blocks in parameters |
| $\boldsymbol{1}_a$ | all-ones vector of size $a$ |
| $\boldsymbol{I}_a$ | identity matrix of size $a \times a$ |
| $\mathrm{vec}(\cdot), \mathrm{blockdiag}(\cdot)$ | row-wise vectorization, block diagonal matrix |
| $\otimes$ | Kronecker product |
| $C, N$ | number of classes and training samples |
| $P, P_b$ | dimensions of model parameters, and $b$-th block of parameters |
| $\mathcal{X}$ | sample space |
| $\boldsymbol{\theta}$ | model parameter |
| $\boldsymbol{f}(\cdot), \boldsymbol{\phi}(\cdot)$ | model and feature extractor |
| $\boldsymbol{V}, \boldsymbol{b}$ | weight matrix and bias of the linear head |
| $h, d$ | dimensions of features and tokens |
| $\boldsymbol{x}^{(i)}, y^{(i)}$ | $i$-th training sample and label |
| $L(\cdot)$ | training loss |
| $\widehat{L}(\cdot)$ | mini-batch loss |
| $\eta_t$ | learning rate at iteration $t$ |
| $\ell(\cdot, \cdot)$ | cross entropy loss function |
| $\boldsymbol{\sigma}_{\mathrm{SM}}(\cdot), \mathrm{sign}(\cdot)$ | softmax and sign function |
| $\mathcal{R}_{\mathrm{FT}}$ | parameter region of fine-tuning |
| $L_* = L(\boldsymbol{\theta}_*)$ | local minimum of training loss |
| $\rho_H$ | Lipschitz constant of the Hessian matrix |
| $L_D$ | block-diagonal approximation of the Hessian matrix |
| $\delta_D$ | upper bound of the approximation of $L_D$ |
| $\sigma_2, \sigma_3$ | constants in the upper bound of the gradient error |
| $\mathrm{SA}(\cdot)$ | single-head self-attention |
| $\boldsymbol{W}_Q, \boldsymbol{W}_K, \boldsymbol{W}_V$ | query, key, and value weight matrix |
| $d_k, d_v$ | dimensions of key/query and value |

