# OpenReview forum: "Gradient Heterogeneity Complements Hessian Heterogeneity in Transformer Optimization"
_TMLR — Decision pending for TMLR_

### Review · Reviewer_ToUG · 2026-05-29

**Summary Of Contributions:**

The paper argues that the Adam–SGD performance gap in Transformer fine-tuning arises jointly from gradient heterogeneity and Hessian heterogeneity. Sign-based methods like SignSGD are shown to be less sensitive to this heterogeneity. The paper provides iteration-complexity upper bounds, connects the analysis to layer normalization placement, and validates claims on NLP and vision tasks.

Understanding why Adam-like optimizers work better than SGD for Transformers is a valuable direction especially through the introduction of   $\Lambda_G$ and $\Lambda_P$ and iteration complexity bounds provides a useful way to compare why gradient-based and sign-based methods behave differently.

**Additional Comments:**

N/A

**Audience:**

Yes

**Audience Explanation:**

The Adam vs SGD gap in Transformer training is a widely relevant and actively studied problem, and TMLR's audience includes researchers working on optimization, deep learning theory, and large language models who would care about this.

**Claims And Evidence:**

Yes

**Claims Explanation:**

The motivation and method are clear, paper is well-written, making it easy to follow and The empirical claims hold up well. The theory is mostly sound but has two loose ends which is described below

- The proof of Theorems 4.6 and 4.8 seems mostly correct, but a few assumptions need clearer justification. The theorem uses a learning rate depending on $\|\nabla L(\theta_t)\|$, while the experiments use gradient clipping for SGD, so this connection should be explained. The condition $\delta_D < \min(\Lambda_G,\Lambda_P)/3$ is also not empirically checked, even though the sign-based bound needs $\delta_D < \Lambda_P/3$. The paper should also discuss whether $\epsilon < \Lambda_G^2/(\rho_H\sqrt{P})$ is realistic for large models, and make the telescoping step clearer.

- The key mechanism depends on $\Lambda_G$ and $\Lambda_P$, but the paper does not compute these for RoBERTa, ViT, or ResNet. It only shows high gradient heterogeneity and gradient-Hessian correlation, then argues this implies $\Lambda_G \gg \Lambda_P$. As a result, the link between theory and SGD’s failure remains qualitative. Reporting actual $\Lambda_G / \Lambda_P$ values for the real models would make the claim much stronger.
- The stochastic bound in Theorem 4.8 relies on Eq. (4), but Figures 2 and S.5 only show cross-coordinate correlation between gradient size and noise. They do not directly verify the required per-coordinate variance bound. This would require sampling many mini-batches at the same $\theta$, so the current support is suggestive rather than confirmatory.



- Gradient–Hessian correlation varies across model types; ResNet case is weak. in particular, why the correlation is stronger for Transformers than CNNs, given that the theory applies to general parameter-block models.
- The Post-LN explanation is fine, but the wording is too informal. The paper says $J_{\text{LN}}$ “proportionally scales the entire Jacobian,” but $J_{\text{LN}}$ depends on input magnitudes and can vary across tokens and layers. To make this claim rigorous, the authors should show that this variation is larger or compounds more in Post-LN than in Pre-LN. Table 2 gives useful empirical support, but the theoretical argument should be strengthened or stated more cautiously.
-  The paper reports Gini coefficients over parameter-level normalized gradients, but the theory defines gradient heterogeneity using block-level gradient norms $\|[\nabla L(\theta)]_b\|_2$. These are not the same. The authors should clarify how the empirical Gini measure connects to the theoretical quantities, especially $\Lambda_G$.

**Requested Changes:**

The paper would be much stronger if the authors added:

- More careful Adam-specific framing, or a direct extension of the theory to Adam/RMSProp
- The positioning of this work with the literature is very weak.  Authors are suggested to compare and contrast  in the "optimization challenges .." section in the related work section with the literature especially Zhang et al. (2024a)
- How often is Assumption 4.7 violated in practice, especially for coordinates where $\|\nabla L(\theta_i)\|$ is close to zero?
- A common-norm complexity comparison, so SGD and SignSGD are evaluated under the same stationarity criterion.

- Empirical estimates of $\Lambda_G$, $\Lambda_P$, and $\Lambda_G / \Lambda_P$ across models and datasets.

---

> ### Author Response · Authors · 2026-06-19
> **Author Response**
>
> Thank you for your positive assessment of our work. We appreciate your recognition of the roles of gradient and Hessian heterogeneity in explaining the Adam–SGD gap, as well as our theoretical and empirical contributions. We are also grateful for your view that understanding optimizer behavior in Transformer training is an important research direction. Below, we address each concern in turn.
>
> ---
> > W1: The proof of Theorems 4.6 and 4.8 seems mostly correct, but a few assumptions need clearer justification. The theorem uses a learning rate depending on
> $|\nabla L(\theta_{t})|$, while the experiments use gradient clipping for SGD, so this connection should be explained. The condition $\delta_D < \min(\Lambda_{G}, \Lambda_{P})/3$ is also not empirically checked, even though the sign-based bound needs $\Lambda_{P}/3$. The paper should also discuss whether $\varepsilon<\Lambda_{G}^2/(\rho_{H}\sqrt{P})$ is realistic for large models, and make the telescoping step clearer.
>
> We thank the reviewer for these helpful suggestions. We addressed each point as follows.
>
> * We clarify that the gradient-dependent factor in the theorem controls the global update scale, while tuned learning rates, schedules, and clipping play this role in practical experiments. We added an explicit explanation in the *Training* paragraph of the experimental setup. Gradient clipping controls the global update scale and is closely related to normalized gradient descent up to a constant factor, whereas Adam-like and sign-based methods additionally perform coordinate-wise reweighting.
> * Regarding the condition $\delta_D < \min(\Lambda_G,\Lambda_P)/3$, we agree that the original manuscript did not provide empirical information about the relevant quantities. While directly estimating the near-block-diagonal approximation error $\delta_D$ remains challenging, we added empirical estimates of $\widehat{\Lambda}_G$ and $\widehat{\Lambda}_P$ in the appendix section *Empirical estimates of weighted Hessian quantities*. These results provide the empirical scale of the main curvature quantities appearing in the condition and strengthen the connection between the theory and the experiments.
> * We view the condition $\varepsilon < \Lambda_G^2/(\rho_H\sqrt{P})$ as a technical assumption that restricts the analysis to sufficiently small stationarity levels. Our goal is to characterize the iteration complexity in the regime where \(\varepsilon\) is small enough for the Hessian-heterogeneity effects captured by our bounds to become relevant, rather than to impose a practical constraint on optimization.
> * We made the telescoping argument more explicit in the proofs of Theorems 4.6 and 4.9 by clearly stating the lower bound on $\eta_t$, the event used in the stochastic setting, and how the stationarity condition yields the final iteration-complexity bound.
>
> ---
> > W3: The stochastic bound in Theorem 4.8 relies on Eq. (4), but Figures 2 and S.5 only show cross-coordinate correlation between gradient size and noise. They do not directly verify the required per-coordinate variance bound. This would require sampling many mini-batches at the same $\theta$, so the current support is suggestive rather than confirmatory.
>
> We substantially revised the stochastic analysis rather than claiming that Figures 2 and S.5 directly verify the per-coordinate variance bound. In the "Stochastic setting" subsection, the main SignSGD bound is now stated in Theorem 4.9 under a new "Sign reliability" assumption (Assumption 4.8) based on a weighted sign mismatch rate $q<1/2$, which is the quantity directly needed in the proof.
>
> We also added a new appendix section, "Empirical sign reliability," where we estimate this weighted sign mismatch rate from coordinate-gradient data. In Appendix A, we further show that when Eq.(4) holds with $\tau_2=0$, Chebyshev's inequality implies sign reliability with $q=\sigma_2$; the resulting explicit bound is given as a sufficient specialization corollary, not as the main stochastic result. Thus, Figures 2 and S.5 are now treated as qualitative motivation for the gradient-error relationship, not as direct empirical verification of the stronger coordinate-wise condition.

---

> ### Author Response · Authors · 2026-06-19
> **Author Response (Part II)**
>
> > W4: Gradient–Hessian correlation varies across model types; ResNet case is weak. in particular, why the correlation is stronger for Transformers than CNNs, given that the theory applies to general parameter-block models.
>
> **We believe this is because heterogeneity is much smaller in ResNets**. Transformer models contain functionally diverse blocks, such as attention, MLP, and normalization layers, whose gradient norms and Hessian spectra can differ substantially across blocks. In contrast, ResNets are composed mainly of repeated convolutional blocks with more homogeneous structure and scale. As a result, both gradient heterogeneity and Hessian heterogeneity are less pronounced, making the Gradient--Hessian correlation more difficult to observe. This interpretation is also consistent with the smaller Adam--SGD gap observed in ResNets compared with Transformers.
>
> ---
> > W5: The Post-LN explanation is fine, but the wording is too informal. The paper says $J_{LN}$ “proportionally scales the entire Jacobian,” but $J_{LN}$ depends on input magnitudes and can vary across tokens and layers. To make this claim rigorous, the authors should show that this variation is larger or compounds more in Post-LN than in Pre-LN. Table 2 gives useful empirical support, but the theoretical argument should be strengthened or stated more cautiously.
>
> **We revised the wording in the "Greater gradient heterogeneity in Post-LN" paragraph to make the claim more cautious.** Specifically, we changed the statement that $J_{\mathrm{LN}}$ "proportionally scales the entire Jacobian" to the following:
>
> ```
> Equation (8) shows that the Jacobian of layer normalization, $\bm{J}_{\text{LN}}$, depends on the input, causing variations in its scale across tokens and layers. From Eqs.(6) and (7), we observe that in Post-LN, $\bm{J}_{\text{LN}}$ appears outside the residual branch and is multiplied with broad components of the layer Jacobian. This multiplicative placement can compound input-dependent scale variation more directly than in Pre-LN, where the identity path bypasses the normalization Jacobian."
> ```
>
> This revision avoids claiming a uniform proportional scaling effect and instead states the more limited mechanism supported by the Jacobian expressions and Table 2.
>
> ---
> > W6: The paper reports Gini coefficients over parameter-level normalized gradients, but the theory defines gradient heterogeneity using block-level gradient norms. These are not the same. The authors should clarify how the empirical Gini measure connects to the theoretical quantities, especially $\Lambda_{G}$
>
> To connect the experiments more directly to the theoretical quantities, **we added the appendix section "Empirical estimates of weighted Hessian quantities,"** where we compute $\hat{\Lambda}_G$, $\hat{\Lambda}_P$, and $\hat{\Lambda}_G/\widehat{\Lambda}_P$ from block-wise Hessian and gradient statistics. Thus, the revised paper uses the Gini coefficient as qualitative evidence of gradient heterogeneity, while the new $\widehat{\Lambda}$ estimates provide the direct empirical connection to the quantities appearing in the complexity bounds.

---

> ### Author Response · Authors · 2026-06-19
> **Author Response (Part III)**
>
> > Q1: More careful Adam-specific framing, or a direct extension of the theory to Adam/RMSProp
>
> **We revised the framing to make clear that our theory analyzes a sign-based proxy for Adam-like behavior, not Adam itself**. Specifically, we changed statements such as
>
> > "Adam's coordinate-wise normalization makes its update directions depend mainly on gradient signs, so Adam can be interpreted as a soft variant of SignSGD"
>
> to
> > "Adam's coordinate-wise normalization makes its update directions depend strongly on gradient signs, which motivates our use of SignSGD as an analytically tractable proxy for Adam-like behavior."
>
> We also changed
> > "we provide a theoretical explanation of Adam's advantage"
>
> to
> > "we focus on a sign-based proxy for Adam-like behavior and analyze how heterogeneity across parameter blocks affects gradient-based and sign-based optimization differently."
>
> These changes avoid claiming a direct Adam theorem while keeping the connection to adaptive optimizers explicit.
>
> ---
>
> > Q2: The positioning of this work with the literature is very weak. Authors are suggested to compare and contrast in the "optimization challenges .." section in the related work section with the literature especially Zhang et al. (2024a)
>
> **We strengthened the comparison with Zhang et al. (2024a) in the "Optimization challenges in Transformers" paragraph and in the main results**. Specifically, we added:
>
> > "Our work complements these studies by introducing gradient heterogeneity and theoretically explaining how it interacts with Hessian heterogeneity to affect optimization."
>
> We also added a paragraph in the main results:
>
> > "Zhang et al. (2024a) show that the Adam--SGD gap arises from Hessian heterogeneity. This finding is consistent with our theoretical results (Theorem 4.6), and our analysis further explains this gap by taking gradient heterogeneity into account."
>
> Thus, our contribution is not to replace their Hessian-heterogeneity explanation, but to explain how gradient heterogeneity interacts with Hessian heterogeneity to affect SGD and sign-based methods differently.
>
> ---
> > Q3: How often is Assumption 4.7 violated in practice, especially for coordinates where $|\nabla L(\theta_{i})|$ is close to zero?
>
> We address this by substantially revising the stochastic analysis in the "Stochastic setting" subsection and the corresponding proof. **Rather than relying only on the per-coordinate relative-variance condition, the main SignSGD bound is now stated in Theorem 4.9 under a weighted sign mismatch rate $q<1/2$ (Assumption 4.8)**, which is the quantity directly needed in the SignSGD proof.
>
> We also added a new appendix section, "Empirical sign reliability," where we compute this quantity from existing coordinate-gradient data. The empirical mean mismatch rates are below $1/2$ in all analyzed settings, although some settings are close to the boundary. This avoids making a strong claim that every near-zero coordinate satisfies a uniform relative-noise bound. In Appendix A, we show that when Eq.~\eqref{eq:error_2} holds with $\tau_2=0$, Chebyshev's inequality implies sign reliability with $q=\sigma_2$; the resulting explicit bound is given as a sufficient specialization corollary of Theorem 4.9.
>
>
> ---
> > Q4: A common-norm complexity comparison, so SGD and SignSGD are evaluated under the same stationarity criterion.
>
> **We added a common-$\ell_2$ comparison for SignSGD in the appendix section "Iteration complexity of SignSGD under $\ell_2$ norm"** and explicitly refer to this section immediately after Theorem 4.6 in the main text. Our main comparison still uses the natural dual norms for each steepest-descent direction: $\ell_2$ stationarity for gradient descent and $\ell_1$ stationarity for sign descent. However, we agree that a common criterion is useful for calibration, so the revision includes the SignSGD bound under the same $\ell_2$ stationarity criterion and explains that this comparison is less favorable to SignSGD because $\ell_\infty$-steepest descent is not naturally measured by $\ell_2$ stationarity.

---

> ### Author Response · Authors · 2026-06-19
> **Author Response (Part IV)**
>
> > W2: The key mechanism depends on $\Lambda_{G}$ and $\Lambda_{P}$, but the paper does not compute these for RoBERTa, ViT, or ResNet. It only shows high gradient heterogeneity and gradient-Hessian correlation, then argues this implies $\Lambda_{G} >> \Lambda_{P}$. As a result, the link between theory and SGD’s failure remains qualitative. Reporting actual $\Lambda_{G} / \Lambda_{P}$values for the real models would make the claim much stronger.
>
> > Q5: Empirical estimates of $\Lambda_{G}$, $\Lambda_{P}$, and $\Lambda_{G} / \Lambda_{P}$ across models and datasets.
>
> Following your request, **we added a new appendix section, "Empirical estimates of weighted Hessian quantities,"** reporting empirical estimates of $\widehat{\Lambda}_G$, $\widehat{\Lambda}_P$, and their ratio using existing block-wise Hessian and gradient statistics. The results show a substantial separation between $\widehat{\Lambda}_G$ and $\widehat{\Lambda}_P$ for RoBERTa, particularly in the pre-trained models. For example,
>
> $
> \text{RoBERTa-Base (CB, Pre-trained): }
> \widehat{\Lambda}_G = 40.9,;
> \widehat{\Lambda}_P = 6.14,;
> \widehat{\Lambda}_G/\widehat{\Lambda}_P = 21.0,
> $
>
> $
> \text{RoBERTa-Base (RTE, Pre-trained): }
> \widehat{\Lambda}_G = 52.6,;
> \widehat{\Lambda}_P = 2.28,;
> \widehat{\Lambda}_G/\widehat{\Lambda}_P = 39.9.
> $
>
> In contrast, the separation is much smaller for ResNet18 and ViT, which also exhibit a much smaller performance gap between SGD and Adam. Since $\widehat{\Lambda}_G$ emphasizes Hessian blocks according to the gradient mass while $\widehat{\Lambda}_P$ depends only on parameter dimensions, a large gap between the two quantities indicates that high-curvature blocks also receive disproportionately large gradients. Therefore, these results provide direct evidence that gradient heterogeneity and Hessian heterogeneity occur simultaneously in RoBERTa, consistent with the mechanism predicted by our theory.

---

### Review · Reviewer_RcLK · 2026-06-10

**Summary Of Contributions:**

This paper aims to explain why Adam outperforms SGD in training Transformers. It argues that, beyond Hessian heterogeneity, which was proposed in a previous paper, gradient heterogeneity—the variation in gradient norms across parameter blocks—also plays an important role. The paper identifies cases where gradient and Hessian heterogeneity jointly affect stationarity guarantees. Theoretically, the authors use a norm-based framework to derive deterministic and stochastic iteration complexity upper bounds (Theorems 4.6 and 4.8). Empirically, they show that Transformers exhibit much stronger gradient heterogeneity than ResNets; in these settings, SGD has a clearly worse training loss trajectory, while signSGD performs close to Adam.

**Audience:**

Yes

**Audience Explanation:**

The optimization properties of Transformer training have recently become a topic of interest.

**Claims And Evidence:**

Yes

**Claims Explanation:**

This paper presents numerical evidence suggesting gradient heterogeneity in Transformer training.

**Requested Changes:**

1. Can the authors provide some explanation of the definition of the weighted Hessian norms (Definition 4.4)? They seem to serve as proxies for Lipschitz constants in (nearly) block-wise Hessian scenarios. Why are two different quantities needed?

2. The Hessian Lipschitz condition in Assumption 4.2 is common in the analysis of Newton-type methods or accelerated first-order methods. Since this paper studies first-order methods without acceleration, namely SGD and signSGD, I am curious why the Hessian Lipschitz condition is imposed instead of a gradient Lipschitz condition. Could the authors discuss this point or cite some related references around Assumption 4.2? Similar comments apply to the third-order moment bound imposed on the gradient noise, Assumption 4.7, as this condition differs from the commonly used bounded-variance assumption. I recommend that the authors include a paragraph discussing how the analysis is carried out, particularly for the inequality at the end of page 19, and explain the role played by the Hessian Lipschitz condition in that derivation.

3. There appears to be a mismatch between the theory and the experiments. In the theoretical complexity comparison, the weighted Hessian norm is treated as an important constant, but the authors do not provide numerical results showing its behavior. Since the main argument of the paper is that both gradient and Hessian heterogeneity affect the complexity, it may not be sufficient to show only gradient heterogeneity. One also needs evidence that gradient and Hessian heterogeneity can occur simultaneously.

---

> ### Author Response · Authors · 2026-06-19
> **Author Response**
>
> Thank you for your positive assessment of our work. We appreciate your summary highlighting the role of gradient heterogeneity, alongside Hessian heterogeneity, in explaining Adam’s advantage for Transformer training. Below, we address each of your requested changes.
>
> ---
> > Q1: Can the authors provide some explanation of the definition of the weighted Hessian norms (Definition 4.4)? They seem to serve as proxies for Lipschitz constants in (nearly) block-wise Hessian scenarios. Why are two different quantities needed?
>
> Following your suggestion, **we added the following explanation after Definition 4.4**:
>
> ```
> This definition is introduced to bound the local curvature term in the one-step descent bound, $\Delta_t^\top \nabla^2 L(\theta_t)\Delta_t$, where $\Delta_t$ denotes the update direction.
> The effective curvature depends not only on the Hessian blocks but also on how the update direction distributes its magnitude across blocks.
> For the gradient-based sequence, $\Delta_t=\nabla L(\theta_t)$, and the local curvature term is bounded by $\Lambda_G$.
> For the sign-based sequence, $\Delta_t=\sign(\nabla L(\theta_t))$, whose magnitude is independent of the gradient norm, and the corresponding curvature term is bounded by $\Lambda_P$.
> Thus, $\Lambda_G$ and $\Lambda_P$ characterize the effective curvature encountered by gradient-based and sign-based updates, respectively.
> ```
>
> In addition, **we expanded the proof intuition subsection in Appendix A** to further explain how the two quantities arise from different update directions. In particular, $\Lambda_G$ weights Hessian blocks according to the gradient mass, whereas $\Lambda_P$ weights them according to parameter dimensions. Therefore, the two quantities are not redundant; they correspond to different block-weighting schemes in the descent bound and yield distinct complexity constants for gradient-based and sign-based methods, respectively.

---

> ### Author Response · Authors · 2026-06-19
> **Author Response (Part II)**
>
> > Q2: The Hessian Lipschitz condition in Assumption 4.2 is common in the analysis of Newton-type methods or accelerated first-order methods. Since this paper studies first-order methods without acceleration, namely SGD and signSGD, I am curious why the Hessian Lipschitz condition is imposed instead of a gradient Lipschitz condition. Could the authors discuss this point or cite some related references around Assumption 4.2? Similar comments apply to the third-order moment bound imposed on the gradient noise, Assumption 4.7, as this condition differs from the commonly used bounded-variance assumption. I recommend that the authors include a paragraph discussing how the analysis is carried out, particularly for the inequality at the end of page 19, and explain the role played by the Hessian Lipschitz condition in that derivation.
>
> Following your suggestion, **we added the following explanation** after Assumption 4.2:
>
> ```
> We assume Lipschitz continuity of the Hessian matrix, a standard assumption in nonconvex optimization analysis (Nesterov, 2013). Although the algorithms studied in this study are first-order methods, this assumption is used only in the analysis to obtain a second-order descent bound with a controlled third-order remainder. This enables us to capture the effect of Hessian heterogeneity in the analysis.
> ```
>
> **We also added the following explanation** after Assumption 4.7:
>
> ```
> Eq. (3) is introduced because our analysis relies on a second-order descent bound under the Hessian-Lipschitz assumption (Assumption 4.2), which contains a third-order remainder term (Lemma A.1). To control this term in the stochastic setting, we require a bound on the third-order moment of the gradient noise norm.
> ```
>
> Our analysis relies on a second-order descent bound under the Hessian-Lipschitz assumption. Although the algorithms studied in this paper are first-order methods, we use this assumption only as an analytical tool to keep the local curvature term explicit, allowing the convergence analysis to capture the effect of Hessian heterogeneity.
>
> This also explains why Assumption 4.7 contains a third-order moment condition. Since the Hessian-Lipschitz descent bound contains a third-order remainder term, controlling its stochastic counterpart requires a bound on the third-order moment of the gradient noise norm.
>
> Regarding the inequality at the end of page 19, it is derived directly from Lemma A.1. In the original paper, this connection was obscured because the lemma was incorrectly referenced as a theorem and the statement of the lemma appeared after the inequality. We corrected both issues in the revision.
>
> In addition, **we added the following paragraph immediately before Lemma A.1** to clarify the role of the lemma in the overall analysis:
>
> ```
> The following lemma is the starting point of our convergence analysis. Under the Hessian-Lipschitz assumption, it provides a second-order descent bound with a controlled third-order remainder. This form keeps the second-order term explicit, allowing us to relate the convergence behavior to Hessian heterogeneity.
> ```

---

> ### Author Response · Authors · 2026-06-19
> **Author Response (Part III)**
>
> > Q3: There appears to be a mismatch between the theory and the experiments. In the theoretical complexity comparison, the weighted Hessian norm is treated as an important constant, but the authors do not provide numerical results showing its behavior. Since the main argument of the paper is that both gradient and Hessian heterogeneity affect the complexity, it may not be sufficient to show only gradient heterogeneity. One also needs evidence that gradient and Hessian heterogeneity can occur simultaneously.
>
> Following your suggestion, **we added a new experiment in Appendix D.1 that estimates the empirical counterparts of the theoretical quantities**, $\widehat{\Lambda}_G$ and $\widehat{\Lambda}_P$, from block-wise Hessian norms and gradient statistics.
>
> The results show a substantial separation between $\widehat{\Lambda}_G$ and $\widehat{\Lambda}_P$ for RoBERTa, particularly in the pre-trained models. For example,
>
> $
> \text{RoBERTa-Base (CB, Pre-trained): }
> \widehat{\Lambda}_G = 40.9,;
> \widehat{\Lambda}_P = 6.14,;
> \widehat{\Lambda}_G/\widehat{\Lambda}_P = 21.0,
> $
>
> $
> \text{RoBERTa-Base (RTE, Pre-trained): }
> \widehat{\Lambda}_G = 52.6,;
> \widehat{\Lambda}_P = 2.28,;
> \widehat{\Lambda}_G/\widehat{\Lambda}_P = 39.9.
> $
>
> In contrast, the separation is much smaller for ResNet18 and ViT, which also exhibit a much smaller performance gap between SGD and Adam. Since $\widehat{\Lambda}_G$ emphasizes Hessian blocks according to the gradient mass while $\widehat{\Lambda}_P$ depends only on parameter dimensions, a large gap between the two quantities indicates that high-curvature blocks also receive disproportionately large gradients. Therefore, these results provide direct evidence that gradient heterogeneity and Hessian heterogeneity occur simultaneously in RoBERTa, consistent with the mechanism predicted by our theory.

---

### Review · Reviewer_qgxM · 2026-06-15

**Summary Of Contributions:**

This paper aims to understand the optimization dynamics difference between SGD & Adam through the lens of Hessian heterogeneity; While this was previously explored in prior work (e.g. [1]), the paper brings up an additional factor to consider: the gradients across the different Hessian blocks are also different, and thus when measuring Hessian block heterogeneity we ought to take this into account with a different definition (see Definition 4.4). The authors argue SGD is sensitive to this gradient heterogeneity while SignGD is not (Theorem 4.6). While SignGD is not Adam, it's been argued before that the dynamics are sufficiently similar for analysis [1]. An extension to the stochastic case is done in Theorem 4.8. The paper argues that Post-LN induces this greater gradient heterogeneity (Section 4.7) and give some empirical evidence for this (Table 2), which justifies this gradient-heterogeneity-dependent analysis.

Strengths:
- The idea of weighing different Hessian blocks' norms by their relative gradient diversity is creative and well-motivated through both theoretical & empirical analysis.
- The paper is clearly written and easy to read.

Weaknesses:
- The argument for the gradient heterogeneity strongly influencing SGD is not supported by enough evidence (the theory is an upper bound, not a lower bound; There is no ablation for the effect of layer normalization on the SGD-Adam gap with the same network but only that difference in architecture).
- The stochastic theory requires interpolation.

[1] [Frederik Kunstner, Jacques Chen, Jonathan Wilder Lavington, and Mark Schmidt (2023) "Noise Is Not the Main Factor Behind the Gap Between SGD and Adam on Transformers, but Sign Descent Might Be." arXiv preprint arXiv:2304.13960. https://arxiv.org/abs/2304.13960](https://arxiv.org/abs/2304.13960)

**Audience:**

Yes

**Audience Explanation:**

I think the paper is quite interesting as this a topic that has received a lot of attention in the optimization community, as in [1], [3], [4].

[1] [Frederik Kunstner, Jacques Chen, Jonathan Wilder Lavington, and Mark Schmidt (2023) "Noise Is Not the Main Factor Behind the Gap Between SGD and Adam on Transformers, but Sign Descent Might Be." arXiv preprint arXiv:2304.13960. https://arxiv.org/abs/2304.13960](https://arxiv.org/abs/2304.13960)
[3] [Weronika Ormaniec, Felix Dangel, and Sidak Pal Singh (2024) "What Does It Mean to Be a Transformer? Insights from a Theoretical Hessian Analysis." arXiv preprint arXiv:2410.10986. https://arxiv.org/abs/2410.10986](https://arxiv.org/abs/2410.10986)
[4] [Yushun Zhang, Congliang Chen, Tian Ding, Ziniu Li, Ruoyu Sun, and Zhi-Quan Luo (2024) "Why Transformers Need Adam: A Hessian Perspective." arXiv preprint arXiv:2402.16788. https://arxiv.org/abs/2402.16788](https://arxiv.org/abs/2402.16788)

**Broader Impact Concerns:**

N/A.

**Claims And Evidence:**

No

**Claims Explanation:**

Not all of them.

1. I am not 100% convinced that this effect on SGD is major or very well-pronounced compared to the baseline Hessian heterogeneity. While the paper shows Pre-LN increases the gradient heterogeneity, the change in Gini coefficient is overall small. I don't think we have the training curves needed to prove that removing Post-LN actually closes the Adam-SGD performance gap.
2. The stochastic proofs assume gradient noise variance shrinks proportionally to the full-batch gradient (Eqs. 3 & 4), which requires interpolation (if the full gradient is zero, all stochastic gradients have to be zero). This condition is quite strong, and not realistic at all in standard (noisy) pretraining or fine-tuning.
3. I think some missing baselines would be very interesting: does column- or row-normalization close this gap? Prior work (e.g. [2]) shows that just adding one of these bridges a significant part of the gap between SGD and Adam. Can the theoretical analysis be extended to this setting?
4. We already know about the difference between SignGD & GD as steepest descent on different norms, and we know how to set the learning rate that maximizes descent based on that; What is the use of Section 4.6? It basically just shows we should have minimized the bound to maximize descent, but that's already present in a lot of prior work.

[2] [Athanasios Glentis, Jiaxiang Li, Andi Han, and Mingyi Hong (2025) "Memory-Efficient LLM Pretraining via Minimalist Optimizer Design." arXiv preprint arXiv:2506.16659. https://arxiv.org/abs/2506.16659](https://arxiv.org/abs/2506.16659)

**Requested Changes:**

Please address my concerns in the soundness section.

---

> ### Author Response · Authors · 2026-06-19
> **Author Response**
>
> Thank you for the careful reading and constructive feedback. We are particularly encouraged that you found our gradient-weighted notion of Hessian heterogeneity to be creative and well motivated, and that the overall presentation was clear and easy to follow.
>
> Below, we address each concern in turn.
>
> ---
> > Q1: I am not 100% convinced that this effect on SGD is major or very well-pronounced compared to the baseline Hessian heterogeneity. While the paper shows Pre-LN increases the gradient heterogeneity, the change in Gini coefficient is overall small. I don't think we have the training curves needed to prove that removing Post-LN actually closes the Adam-SGD performance gap.
>
> We would like to clarify that the primary goal of our paper is not to argue that gradient heterogeneity is more important than Hessian heterogeneity, nor that it alone explains the Adam--SGD performance gap. Rather, **our goal is to theoretically show that gradient heterogeneity provides an additional source of the gap beyond the Hessian heterogeneity identified in prior work**.
>
> Our intention in the original Section "Optimization of Transformers" was not to establish a causal link between LayerNorm placement and the Adam--SGD performance gap. Instead, the purpose of this section is to investigate potential sources of gradient heterogeneity in Transformers. Specifically, we analyze how the placement of LayerNorm affects the distribution of gradient magnitudes across parameter blocks.
>
> To make this objective clearer, **we renamed the section to "Gradient heterogeneity in Transformers" and revised the discussion accordingly**. The revised text positions the LayerNorm analysis as a diagnostic study of gradient heterogeneity rather than evidence that modifying LayerNorm alone closes the Adam--SGD gap.
>
> ---
> > Q2: The stochastic proofs assume gradient noise variance shrinks proportionally to the full-batch gradient (Eqs. 3 & 4), which requires interpolation (if the full gradient is zero, all stochastic gradients have to be zero). This condition is quite strong, and not realistic at all in standard (noisy) pretraining or fine-tuning.
>
> To address this concern, **we revised the stochastic analysis** by introducing additive noise floors:
>
> $\mathbb{E}[|\nabla \widehat{L}_i-\nabla L_i|^2]
> \le
> \sigma_2 |\nabla L_i|^2 + \tau_2$
> and
> $\mathbb{E}[|\nabla \widehat{L}-\nabla L|_2^3]
> \le
> \sigma_3 |\nabla L|_2^3 + \tau_3.$
>
> These assumptions no longer require the stochastic gradient noise to vanish when the full-batch gradient is zero. The resulting complexity bound now includes a noise-induced threshold $\varepsilon_{\mathrm{noise}}$, which captures the fact that optimization cannot be guaranteed below a certain accuracy level in the presence of persistent gradient noise.
>
> In addition, based on comments from another reviewer, we revised the stochastic analysis of the sign-based sequence by introducing an explicit sign-reliability assumption (Assumption 4.8). This separates the analysis of sign errors from the variance assumptions and makes the assumptions required by SignSGD more transparent. The main sign-based bound is now stated in Theorem 4.9; when Eq. (4) holds with $\tau_2=0$, we also provide a sufficient specialization corollary in Appendix A.
>
> Importantly, while the stochastic assumptions and bounds have been generalized, the roles of the heterogeneity measures $\Lambda_G$ and $\Lambda_P$ in the complexity analysis remain unchanged.
>
> ---
> > Q3: I think some missing baselines would be very interesting: does column- or row-normalization close this gap? Prior work (e.g. [2]) shows that just adding one of these bridges a significant part of the gap between SGD and Adam. Can the theoretical analysis be extended to this setting?
>
> Following your comment, **we extended our theoretical analysis to block-wise normalized gradient methods and added a new appendix section, "Block-wise Normalized Gradient Descent" (Appendix H). We also cite the work of Glentis et al. (2025)** suggested in your review, which shows that column-wise normalization substantially narrows the Adam--SGD gap in LLM pretraining.
>
> The resulting complexity bound is characterized by a new curvature measure, $\Lambda_{\mathrm{BN}}$, which is less sensitive to gradient heterogeneity across blocks. This suggests a theoretical explanation for why block-wise normalization methods, including column-wise and row-wise normalization, can reduce the sensitivity of SGD to gradient heterogeneity and thereby close part of the Adam--SGD gap.

---

> > ### Author Response · Authors · 2026-06-19
> > **Author Response (continued)**
> >
> > > Q4: We already know about the difference between SignGD & GD as steepest descent on different norms, and we know how to set the learning rate that maximizes descent based on that; What is the use of Section 4.6? It basically just shows we should have minimized the bound to maximize descent, but that's already present in a lot of prior work.
> >
> > Based on your comment, **we revised Section 4.6 from "Implication for learning rates of SignSGD" to "Consistency with existing views of SignSGD"**.
> >
> > We agree that the steepest-descent interpretation of SignSGD and the corresponding learning-rate scaling are well established in prior work. Our intention in Section 4.6 was not to claim a novel learning-rate rule or a new geometric interpretation. Rather, the purpose of this section is to show that the learning-rate dependence appearing in our complexity bound is consistent with existing perspectives, including (i) the interpretation of SignSGD as steepest descent with respect to the $\ell_\infty$ norm and (ii) the optimal step-size scaling recovered in our quadratic analysis.
> >
> > We acknowledge that the original wording could be interpreted as presenting this observation as a contribution. We therefore rewrote Section 4.6 to explicitly position it as a connection between our heterogeneity-based analysis and these existing interpretations.